# Context is Key: A Benchmark for Forecasting with Essential Textual Information

## Abstract

Forecasting is a critical task in decision making across various domains. While numerical data provides a foundation, it often lacks crucial context necessary for accurate predictions. Human forecasters frequently rely on additional information, such as background knowledge or constraints, which can be efficiently communicated through natural language. However, the ability of existing forecasting models to effectively integrate this textual information remains an open question. To address this, we introduce "Context is Key" (CiK), a time series forecasting benchmark that pairs numerical data with diverse types of carefully crafted textual context, requiring models to integrate both modalities. We evaluate a range of approaches, including statistical models, time series foundation models, and LLM-based forecasters, and propose a simple yet effective LLM prompting method that outperforms all other tested methods on our benchmark. Our experiments highlight the importance of incorporating contextual information, demonstrate surprising performance when using LLM-based forecasting models, and also reveal some of their critical shortcomings. By presenting this benchmark, we aim to advance multimodal forecasting, promoting models that are both accurate and accessible to decision-makers with varied technical expertise. The benchmark can be visualized at `https://anon-forecast.github.io/benchmark_report_dev/`.

## 1 Introduction

The estimation of future conditions is the foundation of decision making (Hyndman & Athanasopoulos, 2018) and intelligence (Wang, 2019). Articulated as time-series forecasting, this problem pervades many domains of science and commerce. Accurate forecasting relies on several crucial decisions up to the practitioner (Hyndman & Athanasopoulos, 2018), in particular on: 1. *Model selection*: Choosing the appropriate forecasting model for a given problem, and 2. *Incorporating prior information*: Determining what relevant information should be included in the model and how to effectively integrate it. This involves decisions about statistical priors, inductive biases in the model architecture, and other forms of domain knowledge integration. Traditionally, these decisions have heavily relied on expert knowledge and manual intervention. However, recent advancements in machine learning have shown particular promise in democratizing time-series forecasting by automating both model selection and the integration of prior information.

In the wake of the foundation model paradigm shift (Bommasani et al., 2021), several works (e.g., Liang et al. (2024); Chen et al. (2023); Lim & Zohren (2021)) have addressed automatic model selection by learning flexible, adaptable models that can be applied across various problem scenarios. Unfortunately, when compared to traditional statistical methods, current approaches provide debatable performance improvements while requiring significantly more resources (Garza & Mergenthaler-Canseco, 2024). Moreover, these models typically cast inputs and outputs as purely numerical time series, which leaves no room for the context that human experts typically rely on to focus their modelling efforts.

An alternative class of recent approaches (Jin et al., 2024; Liu et al., 2024c; Requeima et al., 2024) adapt large language models (LLMs) for forecasting and leverage natural language as an intuitive interface to integrate side information. These methods overcome a significant limitation of traditional forecasting techniques by eliminating the need to manually encode priors or design specialized models. They further hold the promise to capture a broader range of prior knowledge and context, potentially leading to more comprehensive and accurate forecasts. Unfortunately, there are as of yet no systematic evaluations of these models' abilities to jointly leverage historical observations and natural language for forecasting. While several benchmarks for context-aided forecasting have been

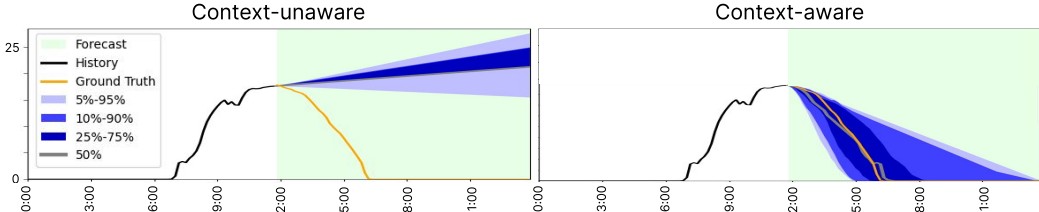

**Figure 1:** *An example task from the proposed Context is Key (CiK) benchmark with forecasts produced by a context-aware model. **Left:** Using the numerical history alone leads to poor forecasts, as nothing indicates a reversion to zero. **Right:** Awareness of the context significantly improves the forecasts as it reveals that no power will be produced during the night (via deductive reasoning) and enables estimating the peak hour of production.*

recently released (Zhang et al., 2023; Liu et al., 2024a; Xu et al., 2024; Emami et al., 2024; Merrill et al., 2024), their contexts are not guaranteed to be useful for improving performance. Hence, it is still an open question as to what extent existing methods can improve their predictions by leveraging crucially-relevant information provided in textual form.

To this end, we propose the Context Is Key (CiK, pronounced *kick*) benchmark of forecasting tasks with numerical input-output pairs and *essential textual context*. The benchmark is designed to assess a forecaster's ability to utilize both the numerical data and key information contained within the accompanying text, as the accuracy of the forecasts relies heavily on effectively leveraging both; see Fig. 1 for an example where context is imperative to forecast accuracy.

Our contributions are:

- We carefully design 71 forecasting tasks (Sec. 3) spanning 7 domains, which cover various kinds of contextual information (Sec. 3.2), and in addition to basic natural language-processing and time-series analysis, require various capabilities (Sec. 3.3).
- We introduce the Region of Interest CRPS metric (RCRPS) to evaluate context-aided forecasting performance (Sec. 4), which prioritizes context-sensitive windows in the prediction and accounts for constraint satisfaction.
- We evaluate various methods on CiK (Sec. 5), including statistical models, time series foundation models using only numerical data, and LLM-based forecasters capable of incorporating context. We introduce *Direct Prompt*, a simple prompting method that achieves the best results on CiK. Our analysis explores key factors such as the impact of context conditioning, prompting techniques, model capabilities, and discusses failure modes of models.

## 2 PROBLEM SETTING

**Context-Aided Forecasting** This work addresses the problem of *context-aided forecasting*, where the goal is to produce statistical forecasts by incorporating relevant side information (i.e., context). Let $\mathbf{X}_H = [X_1, \ldots, X_t]$ represent a series of random variables corresponding to historical observations, where each $X_\tau \in \mathcal{X} \subseteq \mathbb{R}$, and let $\mathbf{X}_F = [X_{t+1}, \ldots, X_T]$ represent future observations. In the classical statistical forecasting problem, the objective is to estimate the joint distribution of the future observations given the historical data:

$$P(\mathbf{X}_F \mid \mathbf{X}_H).$$

We further assume access to *context*, denoted $\mathbf{C} \in \mathcal{C}$, which is additional data of arbitrary nature ($\mathcal{C}$) that contains information relevant for predicting $\mathbf{X}_F$ and complementary to the history $\mathbf{X}_H$. The task then becomes estimating the distribution:

$$P(\mathbf{X}_F \mid \mathbf{X}_H, \mathbf{C}).$$

Crucially, we restrict our focus to *relevant context*, which we define as context that does not degrade the prediction of future time steps. Formally, for $\mathbf{x}_F \sim \mathbf{X}_F \mid \mathbf{X}_H, \mathbf{C}$, given some loss function $\mathcal{L}$ assessing a predictive distribution over $\mathbf{X}_F$ against a realization $\mathbf{x}_F$, $\mathcal{L} : P(\mathbf{X}_F) \times \mathbf{x}_F \rightarrow \mathbb{R}$, we are

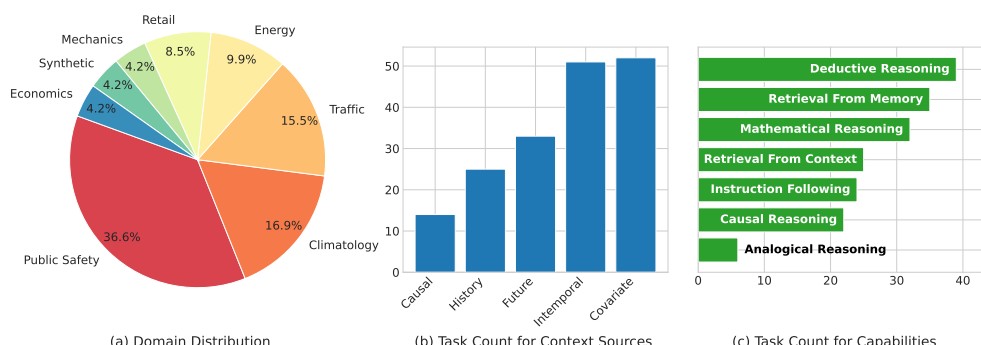

(a) Domain Distribution      (b) Task Count for Context Sources      (c) Task Count for Capabilities

**Figure 2:** *Overview: The tasks in the CiK benchmark rely on real-world numerical data, from 7 domains, as well as synthetic data (left), coupled with natural language context capturing up to 5 different aspects of the dynamical process (center), and require up to 7 non-trivial capabilities to unlock accurate forecasts (right).*

interested in systems where, in expectation, forecasts that leverage context perform better:[1]

$$\mathbb{E}_{\mathbf{x}_F} \mathcal{L}(P(\mathbf{X}_F \mid \mathbf{X}_H, \mathbf{C}), \mathbf{x}_F) \leq \mathbb{E}_{\mathbf{x}_F} \mathcal{L}(P(\mathbf{X}_F \mid \mathbf{X}_H), \mathbf{x}_F).$$

Furthermore, although the nature of the context $\mathcal{C}$ can vary widely, we specifically concentrate on *context communicated through natural language*.

## 3   CONTEXT IS KEY: A NATURAL LANGUAGE CONTEXT-AIDED FORECASTING BENCHMARK

We present the *Context is Key* (CiK) benchmark, a collection of probabilistic forecasting tasks that cannot be solved without integrating natural language contextual information with numerical data. CiK consists of 71 distinct tasks spanning seven application domains (Sec. 3.1) and that can be instantiated in different ways, e.g., by changing target time series or by selecting different time windows. These tasks encompass diverse types of contextual information (e.g., past events and known causal relationships; Sec. 3.2), and are designed such that various capabilities (e.g., causal reasoning; Sec. 3.3) are required to fully leverage the context and *unlock* accurate forecasts (see Fig. 2 for an overview). One key particularity of CiK is that all tasks are carefully designed to ensure quality, avoiding reliance on automation (e.g., via LLMs) or crowdsourcing (see Appendix A.2 for details). An example task is illustrated in Fig. 1 and others are given in Appendix B. The complete set of tasks can be explored at `https://anon-forecast.github.io/benchmark_report_dev/` and the source code is available at `https://anonymous.4open.science/r/context-is-key-forecasting-E391`.

### 3.1   DOMAINS AND NUMERICAL DATA SOURCES

The vast majority (95%) of tasks in CiK draw numerical data from 2,644 real-world time series acquired from public sources. These series cover a range of domains: Climatology (solar irradiance and cloud coverage (Sengupta et al., 2018)); Economics (unemployment rates across states and counties (U.S. Bureau of Labor Statistics, 2024)); Energy (electricity consumption and production (Godahewa et al., 2021)); Mechanics (experimental properties of physical systems (Gamella et al., 2024)); Public Safety (fire department intervention counts (Ville de Montréal, 2020)); Transportation (highway segment occupancy rates and average speeds (Chen et al., 2001)); and Retail (cash withdrawals from various ATMs (Godahewa et al., 2021)). The remaining 5% of tasks use simulated data from dynamical systems crafted specifically for the tasks. Overall, the time series in CiK exhibit diverse sampling frequencies, with observations ranging from every 10 minutes to monthly intervals. Additional details on data sources can be found in Appendix A.1.

**Memorization mitigation:** Using publicly available real-world data introduces the risk that pretrained LLMs and time-series foundation models may have memorized portions of the data,

---

[1]Using the negative log-probability as the loss function would make this statement equivalent to: the entropy of $P(\mathbf{X}_F \mid \mathbf{X}_H, \mathbf{C})$ must be lower than the cross entropy of $P(\mathbf{X}_F \mid \mathbf{X}_H, \mathbf{C})$ and $P(\mathbf{X}_F \mid \mathbf{X}_H)$.

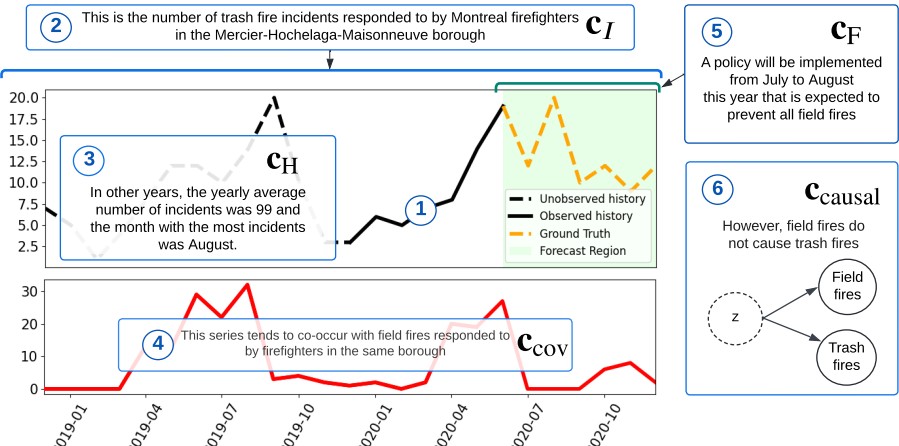

**Figure 3:** *Illustration of a CiK task annotated with types of natural language context:* ① *The short numerical history is misleading, suggesting an increasing trend. However, contextual information compensates and enables accurate forecasts:* ② *The intemporal information ($c_I$) reveals the nature of the series, implying a seasonal pattern with greater prevalence in the summer months due to weather.* ③ *The historical information ($c_H$) complements the short history by providing high-level statistics on past values.* ④ *The covariate information ($c_{cov}$) reveals an association with another quantity: field fires, reinforcing potential seasonal behavior.* ⑤ *In addition, the future information ($c_F$) reveals a future effort to reduce field fires. Could this impact future values of the target series?* ⑥ *No, the causal information ($c_{causal}$) provides the answer.*

potentially inflating evaluation performance. To mitigate this, we employ several strategies. First, we prioritize live data sources that are continuously updated, such as Chen et al. (2001) and Ville de Montréal (2020), ensuring the data is collected after the training cut-off dates of the models being evaluated. Second, where applicable, we use derived series that are not directly available in the raw data, such as converting an incident log into time series (Ville de Montréal, 2020). Finally, as a last resort, we apply minor transformations, such as adding noise or shifting timestamps, but use these sparingly to avoid misalignment between common-sense knowledge (e.g., holiday dates) and the numerical data. We provide details on the mitigation methods used in Appendix A.1.

## 3.2 NATURAL LANGUAGE CONTEXT

For each task in the benchmark, we jointly sample numerical data from one of the series described in Sec. 3.1 and then *manually* craft the natural language context necessary to unlock accurate forecasts. In some cases, this context is purely descriptive, providing information about the general nature of the target variable and its historical behavior, as seen in the task illustrated in Fig. 1. In other cases, the raw numerical data is adjusted to reflect the influence of the context. For example, in one task based on data from Godahewa et al. (2021), an ATM is expected to be inaccessible during a specific period, leading to zero withdrawals (visualized in Appendix B.3). In another task, electricity demand is projected to surge due to an incoming weather event (visualized in Appendix B.2). In such cases, we modify the series to incorporate patterns included in the context.

Overall, we include diverse forms of natural language context, capturing various aspects of the process underlying the time series and revealing complementary knowledge that could be provided by a human expert or an external information source. The types of context are described below and exemplified in the task illustrated in Fig. 3. Several additional examples are given in Appendix B.

**Intemporal information ($c_I$)** Information about the process that remains invariant in time. For example, a description of the process and the nature of the target variable, as in Fig. 3 (point ②), patterns that cannot be inferred from the available numerical data (e.g., long-period seasonalities), or constraints on values (e.g., positivity).

**Historical information ($c_H$)** Information about the past behavior of the series that is not reflected in the available numerical history. For example, statistics on past values of the series, as in Fig. 3 (point ③), or an explanation for spurious patterns to be disregarded at inference (e.g., periodic anomalies due to sensor maintenance).

**Covariate information ($c_{cov}$)** Information about any additional variables that are statistically associated with the variable of interest and that may help prediction. For instance, a variable correlated with the target values (as in Fig. 3 point ④).

**Future information ($c_F$)** Information relevant to the future behavior of the time series. For example, a scenario to be simulated (as in Fig. 3 point ⑤) or expected events along with any entailed constraints (e.g., an inventory shortage restricting future sales amounts).

**Causal information ($c_{causal}$)** Information about causal relationships between covariates and the target variable. For example, if the covariates are known to cause or are confounded with the target variable (as in Fig. 3 point ⑥).

Finally, we note that, in contrast with the work of Zhang et al. (2023); Merrill et al. (2024); Liu et al. (2024a); Emami et al. (2024) which rely on LLM-created context or scraped news articles, all contextual information and data transformations in the CiK benchmark are manually crafted, using the procedure described in Appendix A.2, to ensure both quality and relevance. The quality of the natural language context in CiK is further demonstrated in Appendix A.3.

## 3.3 MODEL CAPABILITIES

In addition to forecasting and natural language understanding, all tasks are designed such that fully utilizing the contextual information requires a range of capabilities, including **instruction following**, various forms of **reasoning**, and **retrieval**.

For example, to solve the task in Fig. 3 , the model could *retrieve from memory* that Montreal experiences snowfall and cold weather during the winter months. It could then infer that trash fires are less likely to occur during this period through *deductive reasoning*. This chain of thought reveals a seasonal pattern that is not apparent in the short numerical history. Additionally, through *causal reasoning*, it is apparent that, despite a strong association between field fires and trash fires, the intervention described in ⑤ is unlikely to reduce the frequency of the latter. Failure to recognize this distinction would lead to inaccurate forecasts.

A list of all capabilities with definitions is available in Appendix A.6 and the capabilities required to solve each task are documented at `https://anon-forecast.github.io/benchmark_report_dev`. The distributions of tasks per capability and context type are shown in Fig. 2, while the distribution of lengths of the numerical historical data, prediction horizons and natural language context are provided in Appendix A.7. Multiple example tasks from CiK are given in Appendix B, along with an explanation of their sources of natural language context and the capabilities required to solve them.

## 4 REGION OF INTEREST CONTINUOUS RANKED PROBABILITY SCORE

Alongside the tasks, we introduce the Region of Interest CRPS (RCRPS), a novel proper scoring rule designed specifically for context-aided probabilistic forecasting. This new scoring rule is an extension of the Continuous Ranked Probability Score (CRPS; Gneiting & Raftery (2007)), a proper scoring rule that provides a comprehensive assessment of forecast quality by evaluating the entire predictive distribution rather than focusing solely on summary statistics. Importantly, since it is based on the CRPS, the RCRPS can be calculated using only samples from the predictive distribution, and so can be used even in cases where closed-form distributions are unavailable. The RCRPS extends the CRPS via two key components: a *region of interest* and a measure of *constraint satisfaction*. This allows assessing both forecast accuracy and the integration of contextual information.

**Region of interest (RoI):** The score reweighs a strict subset of time steps, denoted by $\mathcal{I} \subseteq [t+1, \ldots, T]$, whose values are heavily informed by the context. For example, in the ATM task described in Sec. 3.2 (visualized in Appendix B.3), this would correspond to the time intervals during which the ATM is expected to be unavailable. In other tasks, such as those in Figs. 1 and 3, where the context informs the value of all future time points, we set the RoI to an empty set, essentially weighting all time steps equally (for readability, we report the definition of RCRPS for this special case in Appendix E).

**Constraint satisfaction:** The score penalizes violations of constraints, whether explicitly or implicitly included in the context, by measuring a task-specific function, denoted by $v_C$, whose value is zero for any trajectory that satisfies the constraints and $> 0$ for any trajectory that violates them. Concrete examples are given in Appendix E.4. For tasks whose context does not imply constraints, we use $v_C(\cdot) \equiv 0$.

Given an inferred forecast distribution $\widetilde{\mathbf{X}}_F$ and a ground truth $\mathbf{x}_F$, the scoring rule is defined as:

$$\text{RCRPS}(\widetilde{\mathbf{X}}_F, \mathbf{x}_F) := \alpha \cdot \left[ \frac{1}{2|\mathcal{I}|} \cdot \sum_{i \in \mathcal{I}} \text{CRPS}\left( \widetilde{X}_i, x_i \right) + \frac{1}{2|\neg\mathcal{I}|} \cdot \sum_{i \in \neg\mathcal{I}} \text{CRPS}\left( \widetilde{X}_i, x_i \right) + \beta \cdot \text{CRPS}\left( v_{\mathbf{C}}(\widetilde{\mathbf{X}}_F), 0 \right) \right],$$

where the terms respectively account for CRPS inside the RoI, CRPS outside of the RoI, and constraint satisfaction. We note that the last term is inspired by the threshold-weighted CRPS of Gneiting & Ranjan (2011) and that it vanishes when all constraints are satisfied. The $\alpha$ term is a task-dependent scaling factor that is used to ensure that score values for tasks with numerical data of various scales can be aggregated; its calculation is described in Appendix E.1. Finally, $\beta$ is a scaling factor that controls the impact of constraint violation on the score; we use $\beta = 10$ in our experiments. For additional details and discussion on the RCRPS properness, we refer the reader to Appendix E.

## 5 EXPERIMENTS AND RESULTS

In this section, we define our evaluation protocol (Sec. 5.1) and outline the baseline models evaluated on CiK (Sec. 5.2). We then present results on the benchmark (Sec. 5.3), along with an analysis of factors affecting model performance. Finally, we look at areas for improvement by analyzing forecasting errors (Sec. 5.4) and inference cost (Sec. 5.4).

### 5.1 EVALUATION PROTOCOL

Each task in CiK has many unique specifications, i.e. *instances* arising from the various time series and windows in the associated numerical data, as well as minor variations in natural language context. In order to make the evaluation reproducible and affordable, we deterministically select 5 instances of each task for evaluation. For each instance, we generate 25 independent samples from each model for evaluation. Since many of the tasks in the benchmark share similarities due to having been created from the same data sources or using variants of the same context, we identify these clusters of similar tasks, and design a weighting scheme such that each cluster has equal total weight in our aggregate score (see Appendix A.4 for more details). Finally, to prevent the aggregate scores from being dominated by rare instances where some models give forecasts which are orders of magnitudes away from the ground truth, we introduce an upper bound of 5 to the RCPRS value for each instance, which intuitively represents the value a forecast would get if the distance between the forecast and the ground-truth was 5 times bigger than the range of the ground-truth of the instance.

### 5.2 BASELINES

We evaluate a wide variety of models ranging from methods based on language models to state-of-the-art numerical time series foundation models and classical statistical forecasting methods. Since CiK is meant to be an evaluation benchmark and hence does not have a corresponding training set, we only directly evaluate models that support zero-shot inference (such as LLMs and time series foundation models), and those which can be fit directly to the few historical data points of each task instance evaluated, such as traditional statistical models. We outline these methods below and refer the reader to Appendix D for additional details.

**LLM-based Forecasters:** We consider two prompt-based approaches: LLM-processes (LLMP; Requeima et al. (2024)) and a simple approach which we propose, called "Direct Prompt", where we instruct the model to directly output a forecast as a structured output, rather than prompting it multiple times as in (Requeima et al., 2024) (described in detail in Appendix D.1). For each of these, we evaluate a variety of LLMs with diverse architectures and sizes, such as GPT-4o (Achiam et al., 2023), Mixtral-8x7B (Jiang et al., 2024)), Qwen-2.5-7B (Yang et al., 2024), Llama-3-8B (Dubey et al., 2024), Llama-3.1-405B (Dubey et al., 2024). [2] Next, we evaluate multimodal forecasting models, UniTime (Liu et al., 2024c) and Time-LLM (ETTh1) Jin et al. (2024) each trained according to their respective authors' guidelines (detailed in Appendix D.3). For all of these approaches, inference is performed zero-shot on the benchmark and we compare their performance with and without the natural language context.

**Quantitative Forecasting Models:** To contrast the performance of LLM-based forecasters, we also evaluate a number of models that are only capable of processing numerical data (no natural language). This includes exponential smoothing (Gardner Jr., 1985), ETS (Hyndman et al., 2008), and ARIMA (Box et al., 2015), three simple, but time-tested statistical approaches, as well as four

---

[2]For LLMP, we do not consider Llama-3.1-405b and GPT models as LLMP requires loading model weights into memory, which is infeasible due to resource limitations and confidentiality, respectively.

**Table 1:** *Results on the CiK benchmark. Starting from the left, the first column shows the RCRPS averaged over all tasks. The second column shows the rank of each method w.r.t. other baselines, averaged over all tasks. The remaining columns show the average RCRPS stratified by model capabilities (Sec. 3.3). All averages are weighted according to the scheme described in Sec. 5.1 and accompanied by standard errors. Lower is better and the best averages are in bold. An asterisk (\*) denotes models that do not use natural language context.*

| Model | Average RCRPS | Average Rank | Instruction Following | Retrieval | | Reasoning | | | |
|---|---|---|---|---|---|---|---|---|---|
| | | | | From Context | From Memory | Deductive | Analogical | Mathematical | Causal |
| **Direct Prompt (ours)** | | | | | | | | | |
| Llama-3.1-405B-Inst | **0.159 ± 0.008** | **4.677 ± 0.205** | **0.140 ± 0.013** | 0.109 ± 0.002 | **0.191 ± 0.006** | 0.133 ± 0.001 | **0.167 ± 0.008** | 0.316 ± 0.028 | 0.376 ± 0.039 |
| Llama-3-70B-Inst | 0.518 ± 0.030 | 10.878 ± 0.205 | 0.504 ± 0.038 | 0.371 ± 0.071 | 0.523 ± 0.048 | 0.461 ± 0.048 | 0.694 ± 0.117 | 0.573 ± 0.044 | 0.643 ± 0.049 |
| Llama-3-8B-Inst | 1.647 ± 0.069 | 15.884 ± 0.182 | 1.604 ± 0.131 | 0.199 ± 0.010 | 1.568 ± 0.067 | 2.133 ± 0.082 | 1.555 ± 0.008 | 1.589 ± 0.177 | 1.840 ± 0.238 |
| Mixtral-8x7B-Inst | 1.061 ± 0.058 | 14.035 ± 0.253 | 0.857 ± 0.077 | 0.296 ± 0.049 | 1.077 ± 0.078 | 1.352 ± 0.117 | 1.145 ± 0.144 | 1.000 ± 0.086 | 1.096 ± 0.106 |
| GPT-4o | 0.276 ± 0.010 | **4.596 ± 0.155** | 0.180 ± 0.004 | **0.087 ± 0.003** | 0.519 ± 0.029 | **0.113 ± 0.006** | 0.447 ± 0.029 | 0.590 ± 0.033 | 0.769 ± 0.046 |
| GPT-4o-mini | 0.353 ± 0.022 | 9.394 ± 0.192 | 0.296 ± 0.043 | 0.419 ± 0.014 | 0.471 ± 0.012 | 0.218 ± 0.005 | 1.024 ± 0.033 | 0.475 ± 0.080 | 0.578 ± 0.112 |
| Qwen-2.5-7B-Inst | 0.292 ± 0.032 | 10.802 ± 0.815 | 0.353 ± 0.062 | 0.141 ± 0.021 | 0.307 ± 0.019 | 0.206 ± 0.016 | 0.248 ± 0.032 | 0.399 ± 0.053 | 0.471 ± 0.073 |
| **LLMP** | | | | | | | | | |
| Llama-3-70B-Inst | 0.550 ± 0.013 | 8.443 ± 0.214 | 0.645 ± 0.018 | 0.284 ± 0.015 | 0.392 ± 0.014 | 0.519 ± 0.026 | 0.312 ± 0.019 | 0.453 ± 0.020 | 0.495 ± 0.028 |
| Llama-3-70B | 0.237 ± 0.006 | 6.875 ± 0.272 | 0.310 ± 0.011 | 0.126 ± 0.009 | 0.217 ± 0.007 | 0.134 ± 0.003 | 0.241 ± 0.019 | **0.294 ± 0.008** | **0.329 ± 0.010** |
| Llama-3-8B-Inst | 0.484 ± 0.010 | 9.935 ± 0.178 | 0.345 ± 0.002 | 0.138 ± 0.004 | 0.910 ± 0.030 | 0.242 ± 0.008 | 1.278 ± 0.069 | 0.617 ± 0.022 | 0.787 ± 0.030 |
| Llama-3-8B | 0.313 ± 0.023 | 9.966 ± 0.347 | 0.404 ± 0.043 | 0.124 ± 0.003 | 0.280 ± 0.026 | 0.179 ± 0.014 | 0.267 ± 0.015 | 0.530 ± 0.084 | 0.661 ± 0.117 |
| Mixtral-8x7B-Inst | 0.264 ± 0.004 | 8.898 ± 0.276 | 0.344 ± 0.004 | 0.127 ± 0.003 | 0.224 ± 0.005 | 0.179 ± 0.010 | 0.173 ± 0.009 | 0.348 ± 0.005 | 0.405 ± 0.007 |
| Mixtral-8x7B | 0.262 ± 0.008 | 9.013 ± 0.225 | 0.348 ± 0.012 | 0.146 ± 0.022 | 0.230 ± 0.016 | 0.153 ± 0.002 | 0.230 ± 0.041 | 0.354 ± 0.007 | 0.414 ± 0.009 |
| **Multimodal Models** | | | | | | | | | |
| UniTime | 0.371 ± 0.002 | 14.132 ± 0.109 | 0.271 ± 0.003 | 0.179 ± 0.001 | 0.318 ± 0.001 | 0.510 ± 0.003 | 0.333 ± 0.001 | 0.332 ± 0.001 | 0.384 ± 0.001 |
| Time-LLM (ETTh1) | 0.476 ± 0.001 | 17.443 ± 0.089 | 0.448 ± 0.002 | 0.192 ± 0.000 | 0.373 ± 0.000 | 0.538 ± 0.003 | 0.397 ± 0.001 | 0.382 ± 0.001 | 0.440 ± 0.001 |
| **TS Foundation Models\*** | | | | | | | | | |
| Lag-Llama | 0.329 ± 0.004 | 13.770 ± 0.245 | 0.355 ± 0.007 | 0.181 ± 0.003 | 0.324 ± 0.003 | 0.272 ± 0.006 | 0.342 ± 0.006 | 0.386 ± 0.009 | 0.449 ± 0.012 |
| Chronos | 0.326 ± 0.002 | 12.548 ± 0.156 | 0.385 ± 0.002 | 0.138 ± 0.002 | 0.288 ± 0.002 | 0.249 ± 0.002 | 0.295 ± 0.003 | 0.362 ± 0.003 | 0.417 ± 0.004 |
| TimeGEN | 0.354 ± 0.000 | 15.026 ± 0.107 | 0.402 ± 0.000 | 0.176 ± 0.000 | 0.308 ± 0.000 | 0.279 ± 0.000 | 0.324 ± 0.000 | 0.377 ± 0.000 | 0.431 ± 0.000 |
| Moirai | 0.520 ± 0.006 | 13.038 ± 0.273 | 0.414 ± 0.004 | 0.155 ± 0.004 | 0.260 ± 0.003 | 0.751 ± 0.015 | 0.276 ± 0.008 | 0.337 ± 0.000 | 0.397 ± 0.010 |
| **Statistical Models\*** | | | | | | | | | |
| ARIMA | 0.480 ± 0.006 | 12.925 ± 0.189 | 0.399 ± 0.006 | 0.160 ± 0.002 | 0.517 ± 0.012 | 0.522 ± 0.013 | 0.706 ± 0.026 | 0.354 ± 0.007 | 0.403 ± 0.010 |
| ETS | 0.522 ± 0.009 | 15.031 ± 0.212 | 0.407 ± 0.009 | 0.228 ± 0.010 | 0.682 ± 0.018 | 0.571 ± 0.019 | 0.855 ± 0.035 | 0.453 ± 0.012 | 0.479 ± 0.015 |
| Exp-Smoothing | 0.603 ± 0.013 | 15.689 ± 0.146 | 0.571 ± 0.021 | 0.334 ± 0.013 | 0.743 ± 0.018 | 0.557 ± 0.019 | 0.899 ± 0.035 | 0.673 ± 0.038 | 0.782 ± 0.053 |

state-of-the-art time series foundation models: Lag-Llama (Rasul et al., 2023), Chronos (Ansari et al., 2024) [3] , Moirai (Woo et al., 2024), and TimeGEN (Garza et al., 2023). We note that exponential smoothing, ETS, and ARIMA are fitted to each task instance's numerical history, while the foundation models are evaluated zero-shot.

## 5.3 RESULTS ON THE BENCHMARK

Our main results are shown in Tab. 1. At a high level, we observe that the best-performing baselines combine pretrained LLMs with prompting strategies like Direct Prompt and LLMP, with a bias toward the largest models. In terms of RCRPS, Llama-3.1-405B-Inst (Direct Prompt) significantly outperforms all of its counterparts. As can be seen in Fig. 4, it achieves this only with context. GPT-4o (Direct Prompt) performs worse with respect to RCRPS, but compares favorably in terms of average rank, taking the best average rank by a small margin. This discrepancy is due to strong failures on some of the tasks, which we discuss in Sec. 5.4. Other models like Llama-3-70B (LLMP), Mixtral-8x7B-Inst (LLMP), Mixtral-8x7B (LLMP), and Llama-3-8B (LLMP) are on par with Qwen-2.5-7B-Inst (Direct Prompt) and GPT-4o (Direct Prompt) in terms of RCRPS. Interestingly, all of these baselines outperform UniTime and Time-LLM, which also rely on LLMs (GPT-2 & LLaMA-7B). We discuss this gap in Appendix D.3. Finally, as emphasized in Fig. 5, we observe that the best-performing LLM baselines significantly outperform purely quantitative models. In what follows, we examine various aspects of these results (and refer to Appendix C for additional results).

### Explaining the performance of LLM-based approaches

The stronger performance of LLM baselines could be due to two factors: (i) properly leveraging the natural language context and (ii) being more proficient at numerical forecasting. We thus attempt to disentangle their contributions. On the one hand, Fig. 4 shows clear evidence that most baselines make use of the context to improve their forecasts. For example, Llama-3.1-405B-Inst (Direct Prompt) improves by 67.1% with context. This is reflected in the quality of the forecasts, where we observe clear improvements especially in regions

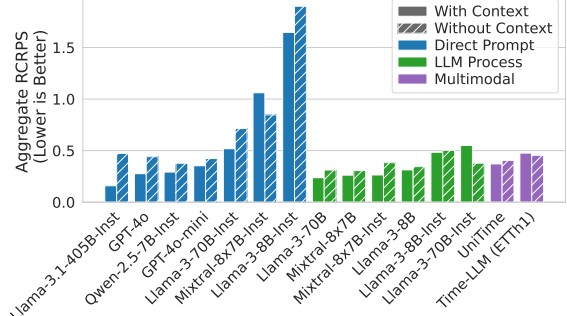

**Figure 4:** *Performance with and without context (lower is better). Full bars show performance with context; striped bars show performance without. All models improve with context, except DP Mixtral-8x7B-Inst, LLMP Llama-3-70B-Inst and Time-LLM. Llama-3.1-405B-Inst improves significantly with context, exhibiting the best aggregate RCRPS.*

---

[3]Results reported here are on Chronos-Large and Moirai-Large. Results on all versions are in App. C.3.

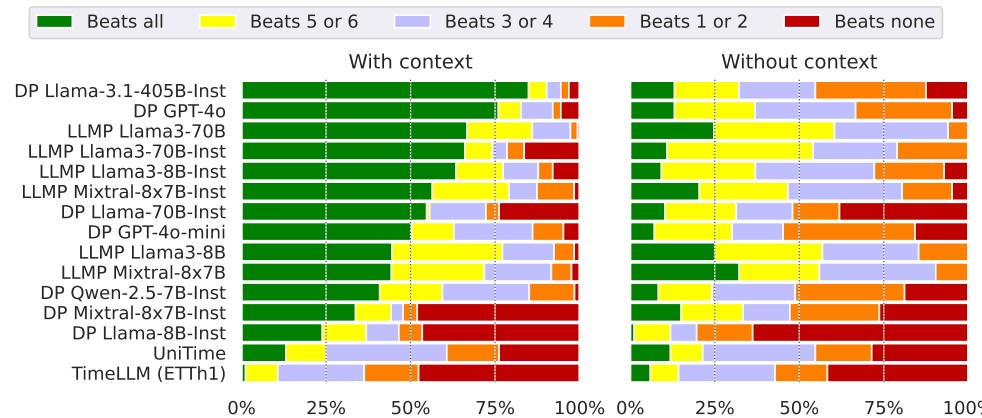

**Figure 5:** *Proportion of tasks for which LLM-based baselines outperform the 7 quantitative forecasting baselines (see Sec. 5.2). A baseline is considered to outperform another on a task if its mean RCPRS is lower on said task. Results are shown for variants that use (left) and do not use (right) the natural language context. A full green bar would indicate that the baseline is better on all tasks, whereas a full red bar would indicate that it is worse everywhere. Averages are weighted according to Sec. 5.1.*

of interest and improved satisfaction of constraints (see Appendix C.5 for examples). Other models show much slighter improvements and, in three cases, even a degradation in performance. These can be explained either by the context being ignored, or by significant failures in using context, impoverishing overall performance (see Sec. 5.4).

On the other hand, Fig. 5 (right) shows that some LLM baselines are surprisingly good forecasters when compared to quantitative forecasting models in a no-context setting. For instance, multiple Llama-3-based models used with the LLMP strategy outperform at least 5 of the quantitative baselines on the majority of tasks. This is further substantiated by results in Appendix C.3. In contrast, other baselines, including the best models Llama-3.1-405B-Inst (Direct Prompt) and GPT-4o (Direct Prompt), show much weaker numerical forecasting abilities without context, suggesting that their performance is mostly due to leveraging the context.

**Comparing the LLMP and Direct Prompting Strategies**

Clear patterns emerge when comparing these strategies. First, as shown in Fig. 5 (right), LLMP baselines exhibit stronger numerical forecasting performance without context than Direct Prompt baselines. This advantage likely stems from LLMP's closer alignment with the forecasting task: LLMP simply prompts the LLM to autoregressively predict the next value in the time series – a task well suited for non-instruction tuned LLMs. This contrasts with Direct Prompting which requires output forecasts to be structured, complicating the overall task.

This line of reasoning leads us to our second observation; as reflected in Tab. 1 and Fig. 5, instruction tuning appears to generally degrade LLMP performance, with Llama-3 models showing a twofold decrease in performance after tuning—a behavior previously observed by Gruver et al. (2024). Interestingly, instruction tuning does not degrade Mixtral-8x7B performance. Finally, while instruction tuning generally harms LLMP, it is essential for models used with the Direct Prompt strategy. Again, Direct Prompt requires forecasts to be produced in a specific structure, a skill that base models typically hone during post-training (see Appendix D.1.1 for details).

**No Baseline Excels Across All Capabilities**

Based on the results in Tab. 1, it is evident that some models possess the necessary capabilities to effectively utilize the contextual information provided. However, no single model is the best across all capabilities. Llama-3.1-405B-Inst (Direct Prompt), our overall top-performing baseline, outperforms its counterparts in only 4 out of 7 capabilities. This finding indicates that the benchmark remains unsolved, leaving significant room for advancements from the research community.

## 5.4 ERROR ANALYSIS

We find that models occasionally return forecasts that miss the ground truth by a large margin. A *significant failure* denotes a forecast that over or undershoots by at least five times the range of the ground truth; at that point, we clip the RCRPS to 5 as explained in Sec. 5.1. Despite this cap, such

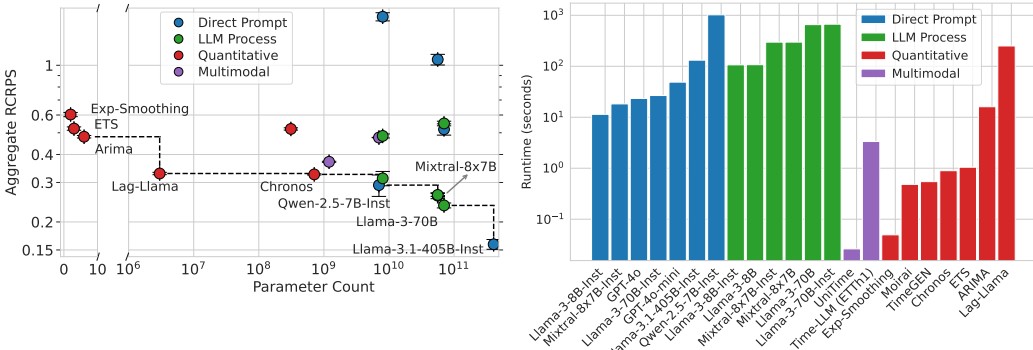

**Figure 6:** *Overview of inference costs. **(Left)** Comparison of average RCRPS (per Tab. 1), vs. the parameter count of each baseline model (lower is better for both). The GPT family, as well as TimeGEN, are left out as there is no information on them about parameter count. The dashed line illustrates the Pareto front: models above and to the right of this front are dominated. Quantitative forecasters dominate the low-parameter regime, while LLM-based methods such as DP Qwen-2.5-7B-Inst or LLMP Llama-3-70B and DP 3.1-405B-Inst offer superior performance for a higher parameter count. **(Right)** Inference time in seconds, for all baselines, averaged over all tasks. Several quantitative methods are much faster on average than LLM-based methods. However, there are significant differences in inference time between the LLM-based forecasters: for the Llama models, LLM Process takes about an order of magnitude more time to run on average than Direct Prompt.*

unpredictable behaviour impacts the results of Tab. 1: GPT-4o with Direct Prompt, while emerging as a top-performer in most tasks (as reflected in its average rank), provides significantly higher aggregate RCRPS than models ranked worse, such as Mixtral-8x7B with LLMP. As an example, Direct Prompt with GPT-4o fails significantly in a task with a context involving scientific notation (see Fig. 17; more examples can be found in Appendix C.6). Notably, while a model may generally achieve a high win rate, a few significant failures can dominate its aggregate performance, as observed for Mixtral-8x7B. We analyse this in detail in Appendix C.10. These findings underscore the need for future work to develop more robust models that can handle context effectively while avoiding significant failures.

### 5.5 INFERENCE COST

A key practical aspect for forecasting applications is the inference time of models and their associated cost. Fig. 6 (left) shows that, while Llama-3.1-405B-Instruct has the best RCRPS, it comes at the cost of a significantly higher parameter count than the quantitative forecasters. This emphasizes that, while LLMs can be powerful context-aware forecasters, they come with a steep computational cost, highlighting the need for efficient models that balance both accuracy and resource demands. Of note is also that many LLM baselines are Pareto dominated by quantitative forecasters such as Lag-Llama and Chronos. This suggests that the ability to ingest text is not enough and that a careful choice of LLM and prompting strategy is crucial for Pareto efficiency.

Fig. 6 (right) emphasizes the disparity in inference time between LLMs and quantitative models. LLMs take significantly longer to make predictions, with the most accurate LLMs having inference times that are orders of magnitude higher than their quantitative counterparts. Quantitative models, in contrast, maintain much lower inference times, making them far more efficient for practical use. The high computational demands of context-aware LLMs hinder their practical use in real-world forecasting, especially where speed and cost matter. The clear benefits of incorporating context warrants research into making them more efficient, aiming to match the cost-effectiveness of traditional models and enabling their deployment in large-scale forecasting.

## 6 RELATED WORK

We review two streams of related work: (i) work that introduce related benchmarks and datasets, and (ii) work that repurpose LLMs to obtain foundation models for context-aided forecasting.

**Benchmarks and Datasets** Merrill et al. (2024) present a benchmark designed to evaluate LLMs' ability to reason about time series, with context-aided forecasting as one assessed capability. Their approach differs from ours in several important ways. First, they focus on purely synthetic time series, which may not accurately reflect real-world dynamics, whereas our benchmark is based primarily on real-world data. Second, their evaluation is limited to point forecasting metrics, which do not measure

the quality of the full forecast distribution. In contrast, we adopt probabilistic forecasting metrics, e.g., the continuous ranked probability score (CRPS; *c.f.* Gneiting & Raftery, 2007), to assess the quality of entire forecast distributions. Other related datasets include Time-MMD (Liu et al., 2024a), which integrates text extracted from reports and web searches, TGTSF (Xu et al., 2024), which incorporates information such as weather reports and news articles, SysCaps (Emami et al., 2024), which includes LLM-generated descriptions of building energy consumption systems, TS-Insights (Zhang et al., 2023), which includes LLM-generated descriptions of trends and seasonalities, and the works of Sawhney et al. (2021); Liu et al. (2024b) who propose automated filtering methods to construct datasets of paired textual and numerical information. The key distinction between these works and ours lies in the role of textual information: while in these works, the text is not essential to generating high-quality forecasts, in our benchmark, all tasks are handcrafted to ensure that accurate forecasts *cannot be achieved* without using the provided textual information.

**Repurposing LLMs for Forecasting** A natural approach to this problem is to build forecasting methods based on LLMs. Xue & Salim (2023) showed that forecasting could be framed as a question-answering problem. Subsequently, Gruver et al. (2024) and Requeima et al. (2024) showed that zero-shot sequence completion could generate accurate forecasts and that textual side-information could be used to influence forecasts. However, their analysis is limited to illustrative examples rather than a comprehensive evaluation. Other approaches incorporate time series into pretrained LLMs (Jin et al., 2024; Liu et al., 2024c; Zhang et al., 2024) by introducing special tokens used to represent patched time series patterns; or modifying their encoders to account for time series data (Jia et al., 2024). While these methods show promising results, their evaluations primarily rely on datasets where the contextual information is not guaranteed to improve forecasts over numerical data alone. As a result, it remains unclear whether their success is driven by accurate numerical forecasting or by effectively incorporating context; this shortcoming motivates our investigation into this question.

## 7 DISCUSSION

In this work, we propose the Context is Key (CiK) benchmark: a collection of forecasting tasks that require combining historical data with critical natural language context. We evaluate a range of models on CiK, including our proposed LLM prompting method, Direct Prompt, which achieves the best performance. We analyse and discuss the failure modes of these models, and our findings underscore the critical role of contextual information in improving forecasts, while also revealing both the unexpected strengths and notable limitations of the investigated LLM-based forecasters.

**Limitations:** While our benchmark provides valuable insights into the integration of contextual information in time series forecasting, it is important to acknowledge its limitations. Our study excludes modalities other than time series data and text, and excludes multivariate time series scenarios. Although we carefully and deliberately designed the tasks to assess how well time series forecasters can integrate contextual information, our focus was on relationships between context and forecasts that are discernible to humans. Hence, our benchmark does not explicitly evaluate a models' capacity to leverage latent relationships that might elude human observation. Moreover, while tasks are designed to require certain capabilities, we do not guarantee that alternative approaches to solving them do not exist. Our collection of capabilities and context types was not intended to be exhaustive but rather to serve as tools for analyzing forecasters' performance on the benchmark. While we have taken steps to mitigate memorization concerns, as discussed in Sec. 3.1, achieving absolute certainty in this regard is challenging without strictly held-out data.

**Future work:** There are several promising avenues for future work. All tasks in the proposed benchmark are univariate forecasting tasks with textual context. Enhancements to the benchmark could include tasks that require multivariate forecasting or incorporate additional modalities, such as images, structured databases, or spatiotemporal data. Tasks that deliberately challenge context length limitations or probe specific weaknesses of language models would also be valuable additions. Methods to automate the generation of large, high-quality datasets for context-aided forecasting are also a valuable direction of investigation. Furthermore, this benchmark strongly motivates research into developing more accurate and efficient multimodal forecasting models, which it is well-positioned to support. Lastly, as models become more robust, they could be integrated into agentic systems with conversational interfaces, allowing forecasts to be augmented with human expertise and automatically retrieved facts (e.g., via search engines). Such advancements would represent a significant step toward automating and democratizing access to powerful forecasting tools.

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

# Appendix

## Table of Contents

# A  ADDITIONAL DETAILS ON THE BENCHMARK

## A.1  DATA SOURCES

We list here the domains and the respective sources of time series data we use in the various tasks in the CiK benchmark. We also show the number of tasks that use each source's data and list any memorization mitigation strategies used for each dataset.

- **Traffic** (11 tasks):
  - **Traffic occupancy rate**: We use traffic occupancy rate (%) data from the California Performance Measurement System (PeMS) (Chen et al., 2001), with frequency hourly. This dataset contains a total of 446 series.
    * As this is a live dataset (updated frequently), we use data from 2024 (i.e. data after the cutoff dates of LLMs used) and do not apply any memorization mitigation strategy.
- **Climatology** (12 tasks):
  - **Solar irradiance and cloud cover data** (9 tasks): We use solar irradiance and cloud cover data for the Americas in 2022 (Sengupta et al., 2018), with frequency either 10 minutes or hourly. We extract a subset of 45 series from this dataset for the benchmark.
    * To mitigate memorization, we shift the dates by one day ahead.
  - **Solar photovoltaic power production** (3 tasks): Time series reflecting solar power production in Alabama during 2006 (Godahewa et al., 2021), with a frequency 10 minutes. This dataset contains a total of 137 series, but our tasks only use a single aggregated series generated from them.
    * To mitigate memorization, we add gaussian noise to the data with a standard deviation of 3% of the standard deviation of the data in each respective sampled window.
- **Public Safety** (26 tasks):
  - **Fire Department Intervention Logs**: Logs of number of interventions carried out by the Montreal Fire Department due to the occurence of various kinds of incidents (such as trash fires, field fires, nautical accidents, bike accidents) (Ville de Montréal, 2020). The data was processed from a raw log and aggregated to monthly frequency. This dataset contains a total of 48 series.
    * Due to it being processed, we do not apply any special memorization mitigation strategy on top.
- **Mechanics** (3 tasks):
  - **Causal Chambers**: Experimental data collected from the wind tunnel physical system from Gamella et al. (2024), released in April 2024. We make use of the load_in, pressure_downwind, pressure_ambient and speed_in series (downsampling them to 1s frequency) to build out-of-distribution forecasting tasks where the target values can be inferred from the driver variate provided as covariate and the description of the physical system given in the context. We select a subset of 17 series from this dataset for the benchmark.
    * Since the data is released in 2024 and after the cutoff dates of the LLMs used, we do not apply any memorization mitigation technique to transform the data.
- **Economics** (3 tasks):
  - **FRED**: American unemployment data at the state and county levels, from the Federal Reserve Bank of St. Louis (U.S. Bureau of Labor Statistics, 2024), with frequency monthly. We extract a subset of 1769 series from this dataset for the benchmark.
    * As this is a live dataset (updated frequently), we use data from 2024 (i.e. data after the cutoff dates of LLMs used) and do not apply any memorization mitigation strategy.
- **Retail** (6 tasks):

- **NN5 ATM cash withdrawals**: The NN5 dataset of ATM cash withdrawals in the UK from the Monash Time Series Forecasting Repository (Godahewa et al., 2021), with frequency daily. This dataset contains a total of 111 series.
    * To mitigate memorization, we add gaussian noise to the data with a standard deviation of 3% of the standard deviation of the data in each respective sampled window.
- **Energy** (7 tasks):
  - **Electricity consumption**: Electricity usage from 2012 to 2014 from the Monash Time Series Forecasting Repository (Godahewa et al., 2021), with frequency daily. This dataset contains a total of 321 series.
    * To mitigate memorization, we add gaussian noise to the data with a standard deviation of 3% of the standard deviation of the data in each respective sampled window.
- **Synthetic Data** (3 tasks): We employ a bivariate setup where the parent variable is drawn from a categorical distribution, and the child variable is generated using a continuous linear Structural Vector Autoregressive (SVAR) model with Gaussian noise, with a lag of 3 and a noise scale of $0.1$.
  - Since this data is synthetic, we do not apply any mitigation technique on top of data to mitigate memorization. Since our models assume a timestamp, we use dates from 2025, and a frequency of daily when we input this data to our models.

Depending on the task and the context used in the task, appropriate history and prediction lengths are used in the task.

## A.2 TASK CREATION PROCESS

All tasks were manually designed, from scratch, by the authors of this work without resorting to external annotators, crowdsourcing, or LLMs. We use the following procedure to create the tasks in the benchmark.

First, we identified high-quality sources of public time series data from various application domains (listed in Appendix A.1). Special care was taken to find data sources that are continuously updated to facilitate future benchmark updates. Second, we established the categorization for sources of context (Sec. 3.2) and capabilities (Sec. 3.3) as a framework to guide the creation of new tasks and ensure their diversity. Third, team members created the tasks, each time

1. Selecting a data source
2. Implementing a time series window selection strategy (e.g., short or long history)
3. Brainstorming about context types and capabilities required to solve the forecasting problem
4. Writing a code to generate the context (e.g., calculating statistics of the series beyond the observed numerical history), and
5. Finally, if required, writing code to modify the time series data to reflect the context (e.g., introducing some spikes in future values).

Then, the tasks were peer-reviewed by a committee composed of all other authors (each with time series research experience). The creator of each task was not allowed to participate in the review. The review ensured that the text was of high quality, that it undoubtedly enabled a better forecast, and that the context source and capability tags were well-assigned. If a task was deemed of not high enough quality, it was either returned for revisions, or rejected.

The code for all tasks is available here: `https://anonymous.4open.science/r/context-is-key-forecasting-E391/`. An example task can be found here: `https://anonymous.4open.science/r/context-is-key-forecasting-E391/cik_benchmark/tasks/montreal_fire/short_history.py`, where the time series window selection occurs from L94-112 and the context generation occurs from L114-158.

After the benchmark was developed, we further assessed the quality of the context using an LLM-based critique to validate that all the tasks are high-quality context aided forecasting tasks. This procedure is detailed in Appendix A.3.

### A.3 AN LLM-BASED CRITIQUE OF THE RELEVANCE OF CONTEXT

To further assess the quality of the tasks, we build an LLM-based critique by prompting GPT-4o with the historical and future numerical data, as well as the context, and asking it to assess whether its estimation of future values would be "significantly better", "slightly better", "unchanged", or "worse" when the context is provided compared to when it is not provided. Note that this experiment was ran after the benchmark was created, as an analysis tool to further validate the quality of the tasks.

We run this critique on 5 instances of each of the 71 tasks and report results in Fig. 7. All tasks are assessed as enabling better forecasts when given context, with the majority of tasks assessed as having contexts that enable "significantly better" forecasts. The code linked to this experiment is provided at https://github.com/anon-forecast/benchmark_report_dev/blob/main/iclr_rebuttal_resources/llm_validation.py. The prompt used in the critique is below:

```
 "

You are a critic whose role is to evaluate the quality of tasks in the "context is key" time
    series forecasting benchmark.

"Context is Key" (CiK) is a time series forecasting benchmark that pairs numerical data with
    diverse types of carefully crafted textual context, requiring models to integrate both
    modalities to arrive at accurate predictions.

Here is a task to evaluate.

<history>
((history))
</history>

<context>
    <background>
        ((background))
    </background>
    <scenario>
        ((scenario))
    </scenario>
    <constraints>
        ((constraints))
    </constraints>
</context>
<future>
((future))
</future>

Assume the following two scenarios:
1) You are given only the numerical data in <history> and have no additional information
    about the nature of the time series. You must ignore the <context> section completely.

2) You are given the <context> section in addition to the numerical data in <history>.

Now, assume you had to estimate the probability distribution of the <future> values given
    the information available in each scenario. How would the quality of your estimation
    change in scenario 2 compared to scenario 1?

First show your reasoning in <reason></reason> tags, then answer in <answer></answer> tags
    with either "significantly better", "slightly better", "unchanged", "worse" (no other
    reponses are allowed).
```

### A.4 WEIGHTING SCHEME FOR TASKS

To take full advantage of the available data, we create multiple tasks using each data source, by varying the specific contextual information we provide to the models. Since we do not want our

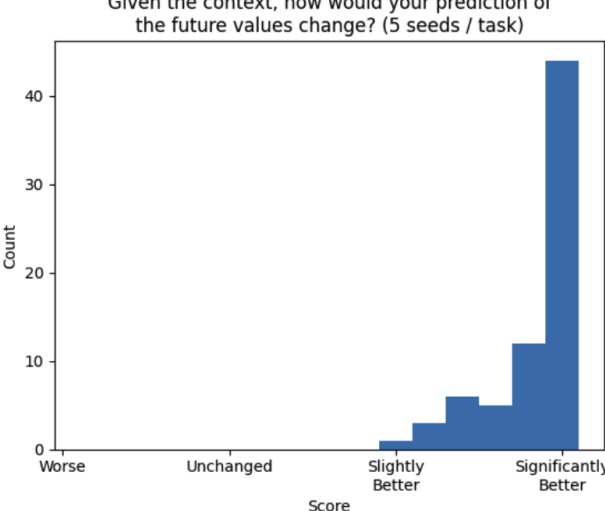

**Figure 7:** *A histogram of results from the LLM-based critique of the relevance of context. Given the historical data, the future data and the associated context of tasks, GPT-4o is asked to assess whether its predictions would be "significantly better", "slightly better", "unchanged", or "worse" (see Appendix A.3 for the details). The context in all tasks is considered as enabling better forecasts, with the majority of tasks having context that enable "significantly better" forecasts.*

aggregate results to be dominated by the few datasets for which there are a larger number of tasks, we weight the contribution of each task to the various aggregated results.

To define the weight of each task, we first group the tasks in clusters. These clusters are primarily defined based on the original data source used to create the tasks. However, when tasks are fundamentally different, due to not testing the same capabilities, we put them in different clusters despite them using the same data source. For example, for tasks created using the Solar irradiance and cloud cover data, all of which ask models to forecast the irradiance, the tasks form three distinct clusters: one for tasks asking models to do forecast with very short history (less than a day), one for tasks giving the cloud cover as covariate, and the final one for tasks where the models are given a tight upper bound on the possible irradiance. Once we define these clusters, we simply equal weight to each cluster, and equal weight to each task inside each cluster.

### A.5 Standard errors and average ranks

To get the standard errors shown in Tab. 1, we first compute the standard error for tasks using the method described in Appendix E.5. We then aggregate them according to each task weight, by assuming that errors for each are independent and thus using the formula for the variance of a weighted sum of independent variables.

To take into consideration the uncertainty we have for the scores, we compute average ranks through a simple simulation. In this simulation, we replace the RCRPS for each task and model pair by an independent Gaussian variable of mean equals to the one we measured, and of standard deviation equals to the standard error. We then draw from this distribution and compute the weighted average ranks for each model. The results shown in Tab. 1 are the mean and standard deviation measured from 10,000 repetitions of this simulation.

### A.6 Model Capabilities

We provide a detailed explanation of each model capability here. Note that tasks in the CiK benchmark need not be mutually exclusive with the model capabilities they require; tasks are tagged with one or more model capabilities.

**Instruction following** (24 Tasks): Using direct instructions available in the context. Instructions could express constraints to be satisfied, or the expected effect of an event, for example.

**Retrieval**: Retrieving facts from memory or context.

- **Retrieval from memory** (35 Tasks): Retrieving from memory facts that enable interpretation of the context, such as relevant physical constants or quantitative laws.
- **Retrieval from context** (25 Tasks): Retrieving relevant information from context and distinguishing it from irrelevant information.

**Reasoning**: Reasoning about information in context or memory.

- **Analogical Reasoning** (6 tasks): Making analogies between entities or events, for instance, applying knowledge from a past event that is similar to an upcoming one.
- **Mathematical Reasoning** (32 tasks): Performing calculations over the context, e.g. solving an equation.
- **Deductive Reasoning** (39 tasks): Inferring new facts not explicitly mentioned in the context, e.g. inferring from the context that certain values are logically impossible to occur.
- **Causal Reasoning** (22 tasks): Deriving or using causal information from the context to reason about actions (such as interventions).

### A.7 TASK LENGTHS

Fig. 8 provides an overview of the distribution of the lengths of the natural language context, numerical history and target (prediction horizon) for a set of five instances for each task in the CiK benchmark.

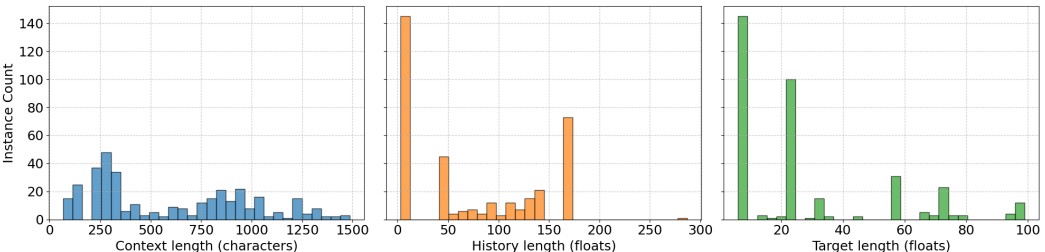

**Figure 8:** *Histograms depicting the distribution of lengths for the context, numerical history and target length of a set of five instances for each task in CiK. We measure the length of the natural language context in characters, and the numerical sequences in floats.*

## B   EXAMPLES OF TASKS FROM THE BENCHMARK

In this section, we feature multiple examples from the benchmark to exemplify exactly what a task is, what context sources represent (Sec. 3.2), and how these tasks encourage the use of capabilities (Sec. 3.3). To visualize all tasks in the benchmark, we refer the reader to `https://anon-forecast.github.io/benchmark_report_dev`.

## B.1 TASK: CONSTRAINED PREDICTIONS

> **Domain:** *Traffic*
> **Context sources:** *Future information*
> **Capabilities:** *Instruction Following*
>
> **Context:** *"Suppose that in the forecast, the values are bounded above by 11.88, the values are bounded below by 7.06."*

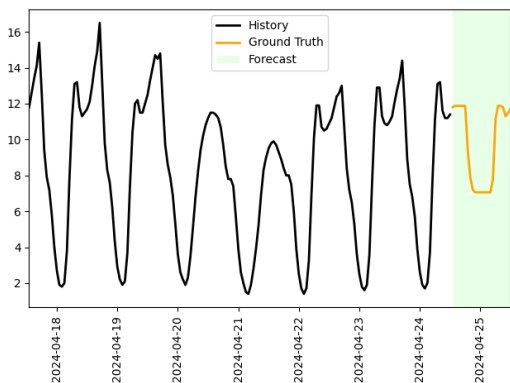

This task, which we refer to as "Bounded Prediction Constraint Based On Prediction Quantiles", is a forecasting task where we modify the forecast horizon (in green in the plot) by bounding one or both of its extremes according to its unmodified ground truth's quantile values. We verbalize these bounds in the context, and the model is expected to interpret and respect them.

Since we draw this series from the PeMS dataset (Chen et al., 2001), we tag its domain as "Traffic". The context directly refers to the future, hence the context source is tagged as "Future information". Finally, since the model is expected to obey the constraints in the context, we tag the evaluated capability as "Instruction following".

Since the context contains constraints, the Region of Interest CRPS metric that we introduce (Sec. 4) heavily penalizes forecasts that exceed these constraints: models that do not incorporate the information about bounds in the context, such as quantitative forecasting models, would not be able to predict the ground truth (orange line) because its lower bound is much higher than that of the history. In this case, the region of interest for the metric is the entire forecast horizon because the context applies everywhere. Although statistical forecasters may pick up on the seasonality present in the history (black line), they would obtain worse scores than models capable of processing the context and adjusting the lower bound of their predictions.

## B.2 TASK: ELECTRICAL CONSUMPTION INCREASE

> **Domain:** *Energy*
> **Context sources:** *Future information, Covariate information*
> **Capabilities:** *Instruction following, Retrieval from context*
>
> **Context:** *"This is the electricity consumption recorded in Kilowatt (kW) in city A. A heatwave struck the city, which began on 2012-10-09 18:00:00 and lasted for approximately 3 hours, saw temperatures soar to unprecedented levels. According to the city's electricity provider, power consumption during the peak of the heatwave reached approximately 5 times the typical usage for this time of year."*

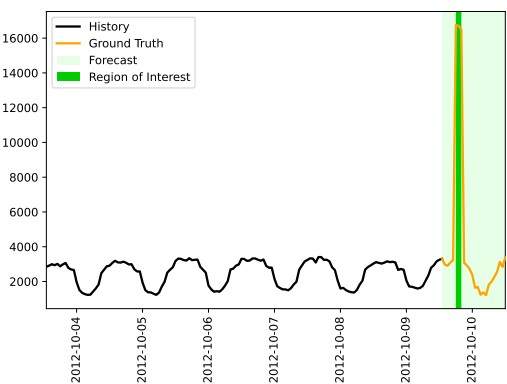

The "Short News Electricity Increase" task introduces a large shock in the forecast horizon that is only referred to in the context. Hence, the model must interpret the context appropriately to forecast the spike.

Since this series represents electricity consumption (Sec. 3.1), we tag it a coming from the "Energy" domain. The context sources for this task are twofold: the first context source is "Future information", which represents knowledge of the five-fold increase in typical usage during the shock. The second source of context, "Covariate information", represents the occurrence of a heatwave, which coincides with the timing and duration of the shock. The model must therefore interpret both the information on the magnitude of the shock from the future information, as well as the timing and duration of the sock from the covariate information. Together, these pieces of information enable an accurate forecast despite the lack of information about the shock in the task's numerical history.

The skills for this task are tagged as "Instruction following" and "Retrieval from context". While instruction following involves interpreting the context to include the shock in the prediction, the model must also retrieve from the context the relevant information, as there is unneeded information in the context as well: an accurate forecast does not require knowing that the temperature has reached unprecedented levels.

In this task, we also see a "Region of Interest" (RoI), characterized by a darker region of the forecast horizon. This RoI represents the region of the forecast horizon for which the context is relevant, i.e. the period during which the increased power consumption occurred. As detailed in Sec. 4, this region of interest is taking into account in the RCRPS metric.

B.3 TASK: ATM MAINTENANCE

***Domain:*** *Retail*
***Context sources:*** *Intemporal information, Covariate information*
***Capabilities:*** *Instruction following, Deductive reasoning*

***Context:*** *"This is the number of cash withdrawals from an automated teller machine (ATM) in an arbitrary location in England. The ATM was under maintenance for 7 days, periodically every 14 days, starting from 1996-11-30 00:00:00. Assume that the ATM will not be in maintenance in the future."*

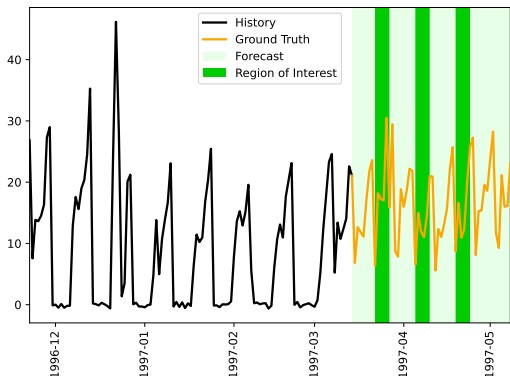

The "Automated Teller Machine (ATM) Under Period Maintenance" task represents the history of withdrawals from an ATM that undergoes regular maintenance. This maintenance introduces a periodic, easily forecastable signal into the history. However, the context explicitly states that the forecast should assume the ATM will not be in maintenance during the forecast. Therefore, forecasting models are expected to ignore this signal.

Since this series represents ATM withdrawals, we tag it as "Retail". The context includes information such as the location of the ATM, and therefore provides "Intemporal information". As the maintenance frequency and duration is also described, the context sources include "Covariate information".

This task is tagged with two capabilities. "Instruction following" is necessary because the model must assume that the ATM will not be in maintenance in the future. However, the model must use "Deductive reasoning" to determine what and when the impact of the maintenance was – reducing the number of withdrawals to 0 every 14 days –, and avoid including that pattern in the forecast. The RoI represents when the maintenance periods would have occurred in the forecast horizon, which is likely where forecasting models that do not leverage the context will forecast 0. While a quantitative forecasting model would find such a signal irresistible, context-aware models should avoid repeating the pattern in the forecast.

We also note that the series is not quite 0 during the maintenance periods. This is a consequence of using one of our memorization mitigation schemes (Appendix A.1, paragraph "Memorization mitigation").

### B.4 Task: Montreal Fire High Season

> **Domain:** *Public Safety*
> **Context sources:** *Intemporal information, Historical information*
> **Capabilities:** *Deductive reasoning, Mathematical reasoning, Retrieval from memory*
>
> **Context:** *"The Montreal Fire Department is in charge of responding to various kind of public safety incidents. This is the number of field fire incidents responded to by Montreal firefighters in the borough of Rivière-des-Prairies-Pointe-aux-Trembles. In other years, the yearly average number of incidents was 106 with the busiest month being June."*

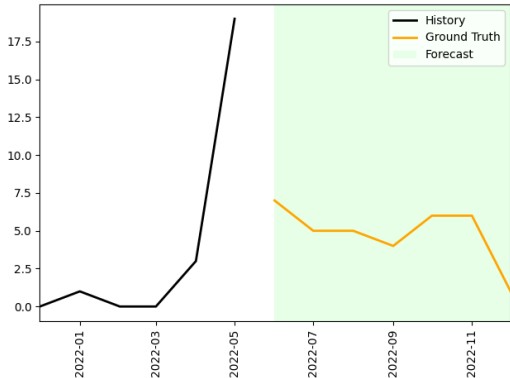

The "Montreal Field Fire With Explicit Short History" task requires predicting the number of field fire incidents during the summer, so we tag it as being part of the "Public Safety" domain.

The context contains information from two different sources: it contains "Intemporal information", such as the location and nature of the incidents. However, it also contains "Historical information", which verbalizes statistics about past values of the series, beyond the numerical data. That is, the yearly average number of incidents, along with the knowledge that June is the month with the most incidents.

This task is tagged with many skills and involves several steps of interpretation to arrive at a reasonable forecast. We first note that the task requires "Retrieval from memory": an important piece of information for this prediction is that winters in Montreal, a city in the northern hemisphere, are long and harsh, with temperatures reaching $-40°C$. Secondly, the task requires the model to use "Deductive reasoning" to deduce that, since temperatures are so cold during the winter months, fields are likely covered in snow and are rather unlikely to catch fire. Finally, the model can employ "Mathematical reasoning" to determine how many field fires are likely to occur on average in the forecast horizon, given the total number of field fires that have already blazed during the history.

Note that "Retrieval from memory" tasks do not explicitly ask the model to return information retrieved from memory; rather, we tag tasks as such because they cannot be solved without key information that is not present in the history or the context.

## B.5   TASK: SOLAR PREDICTION

> **Domain:** *Climatology*
> **Context sources:** *Intemporal information*
> **Capabilities:** *Analogical reasoning, Deductive reasoning, Retrieval from memory*
>
> **Context:** *"This series estimates the power production for a given day of a new solar power plant located in the state of Georgia, which has a climate similar to Alabama's."*

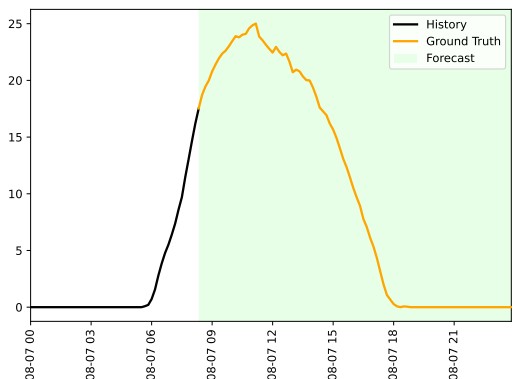

The "Explicit Similar Location and Day Solar Forecast" task requires forecasting the power production of a solar power plant based on a very short history and information about the similarity between its climate and that of an adjacent location. We therefore tag the domain of this series as "Climatology".

Without the "Intemporal information" that the context provides, it is quite possibly impossible to accurately forecast the parabola-like shape of the ground truth: the history contains very few defining characteristics, which makes it interchangeable with that of many potential processes and therefore many possible forecasts. The model must use "Deductive reasoning" to foresee this reversion to zero based on the fact that solar panels do not produce electricity at night.

However, the information in the context alone is not sufficient to provide an accurate forecast: nothing indicates the time at which production should peak. It must therefore rely on "Retrieval from memory" to retrieve information about Alabama's climate and then "Analogical reasoning" to apply it to the present problem.

## B.6 TASK: SPEED FROM LOAD

> **Domain:** *Mechanics*
> **Context sources:** *Causal information, Intemporal information, Covariate information*
> **Capabilities:** *Causal reasoning, Mathematical reasoning, Instruction following*
>
> **Context:** *"The wind tunnel is a chamber with one controllable fan that pushes air through it. We can control the load of the fan (corresponding to the duty cycle of the pulse-width-modulation signal) and measure its speed (in revolutions per minute). The fan is designed so its steady-state speed scales broadly linearly with the load. Unless completely powered off, the fan never operates below a certain speed, corresponding to a minimum effective load between 0.1 and 0.2. The task is to forecast the speed of the fan. The load is between 0 and 1. At full load (=1), the fan turns at a maximum speed of 3000 rpm. The load is set to: 0.0 until 05:47:09, 0.1 from 05:47:09 until 05:47:29, 0.0 from 05:47:29 until 05:48:01, 0.2 from 05:48:01 until 05:48:27, 0.1 from 05:48:27 until 05:48:49, 0.0 from 05:48:49 until 05:49:00."*

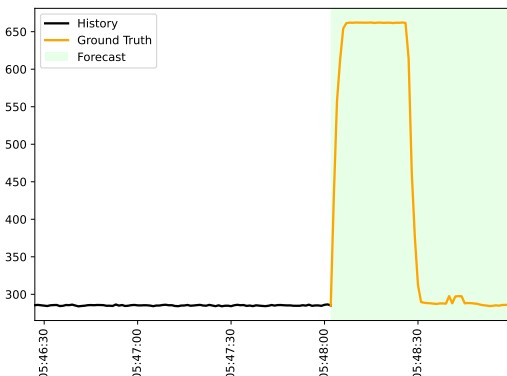

The "Speed From Load" task combines many different context sources and capabilities to produce a forecast of the revolutions per minute (RPM) of a fan in a wind tunnel based on its load. This task, based on the Causal Chambers dataset (Gamella et al., 2024), is therefore tagged as part of the "Mechanics" domain.

As the plot shows, producing an accurate forecast of the ground truth (orange line) from the numerical history alone (black line) is essentially impossible. However, the context of the task is quite rich: it provides "Intemporal information" on the nature of the task, such as the limits of the load and of the fan, "Covariate information" that describes the load during the history and future, as well as "Causal information" on the control that the load exerts on the fan, as well as the proportionality of their relationship.

To leverage the context requires multiple skills: firstly, "Instruction following" is necessary to understand that the task is to forecast the speed of the fan (as opposed to e.g. the load) and to apply the correct loads at the right moments. Secondly, the model must use "Causal reasoning" to understand that the changes in the load will directly impact the speed of the fan. Finally, the model must leverage "Mathematical reasoning" to calculate the speed of the fan as a function of the load.

## C    ADDITIONAL RESULTS

### C.1    FULL RESULTS PARTITIONED BY MODEL CAPABILITIES

Tab. 2 provides the results of all tested models, partitioned by model capabilities.

**Table 2:** *Results on the CiK benchmark. Starting from the left, the first column shows the RCRPS averaged over all tasks. The second column shows the rank of each method w.r.t. other baselines, averaged over all tasks. The remaining columns show the average RCRPS stratified by model capabilities (Sec. 3.3). All averages are weighted according to the scheme described in Sec. 5.1 and accompanied by standard errors. Lower is better and the best averages are in bold. An asterisk (\*) denotes models that do not use natural language context.*

| Model | Average RCRPS | Average Rank | Instruction Following | Retrieval | | Reasoning | | | |
|---|---|---|---|---|---|---|---|---|---|
| | | | | From Context | From Memory | Deductive | Analogical | Mathematical | Causal |
| **Direct Prompt (ours)** | | | | | | | | | |
| Llama-3.1-405B-Inst | **0.159 ± 0.008** | **4.905 ± 0.254** | **0.140 ± 0.013** | 0.109 ± 0.002 | **0.191 ± 0.006** | 0.133 ± 0.001 | **0.167 ± 0.008** | 0.316 ± 0.028 | 0.376 ± 0.039 |
| Llama-3-70B-Inst | 0.518 ± 0.030 | 12.030 ± 0.246 | 0.504 ± 0.038 | 0.371 ± 0.071 | 0.523 ± 0.048 | 0.461 ± 0.048 | 0.694 ± 0.117 | 0.573 ± 0.044 | 0.643 ± 0.049 |
| Llama-3-8B-Inst | 1.647 ± 0.069 | 18.786 ± 0.235 | 1.604 ± 0.131 | 0.199 ± 0.010 | 1.568 ± 0.067 | 2.133 ± 0.082 | 1.555 ± 0.008 | 1.589 ± 0.177 | 1.840 ± 0.238 |
| Mixtral-8x7B-Inst | 1.061 ± 0.058 | 15.813 ± 0.296 | 0.857 ± 0.077 | 0.296 ± 0.049 | 1.077 ± 0.078 | 1.352 ± 0.117 | 1.145 ± 0.144 | 1.000 ± 0.086 | 1.096 ± 0.106 |
| GPT-4o | 0.276 ± 0.010 | **5.021 ± 0.180** | 0.180 ± 0.004 | **0.087 ± 0.003** | 0.519 ± 0.029 | **0.113 ± 0.006** | 0.447 ± 0.029 | 0.590 ± 0.033 | 0.769 ± 0.046 |
| GPT-4o-mini | 0.353 ± 0.022 | 9.792 ± 0.243 | 0.296 ± 0.043 | 0.419 ± 0.014 | 0.471 ± 0.012 | 0.218 ± 0.005 | 1.024 ± 0.033 | 0.475 ± 0.080 | 0.578 ± 0.112 |
| Qwen-2.5-7B-Inst | 0.292 ± 0.032 | 11.810 ± 0.985 | 0.353 ± 0.062 | 0.141 ± 0.021 | 0.307 ± 0.019 | 0.206 ± 0.016 | 0.248 ± 0.032 | 0.399 ± 0.053 | 0.471 ± 0.073 |
| Mistral-7B-Inst | 1.943 ± 0.117 | 19.691 ± 0.843 | 2.255 ± 0.203 | 1.766 ± 0.174 | 1.171 ± 0.155 | 1.992 ± 0.142 | 0.874 ± 0.248 | 1.275 ± 0.223 | 0.952 ± 0.283 |
| **LLMP** | | | | | | | | | |
| Llama-3-70B-Inst | 0.550 ± 0.013 | 9.207 ± 0.254 | 0.645 ± 0.018 | 0.284 ± 0.015 | 0.392 ± 0.014 | 0.519 ± 0.026 | 0.312 ± 0.019 | 0.453 ± 0.020 | 0.495 ± 0.028 |
| Llama-3-70B | 0.237 ± 0.006 | 7.344 ± 0.290 | 0.310 ± 0.011 | 0.126 ± 0.009 | 0.217 ± 0.007 | 0.134 ± 0.003 | 0.241 ± 0.019 | **0.294 ± 0.008** | **0.329 ± 0.010** |
| Llama-3-8B-Inst | 0.484 ± 0.010 | 10.875 ± 0.204 | 0.345 ± 0.002 | 0.138 ± 0.004 | 0.910 ± 0.030 | 0.242 ± 0.008 | 1.278 ± 0.069 | 0.617 ± 0.022 | 0.787 ± 0.030 |
| Llama-3-8B | 0.313 ± 0.023 | 10.924 ± 0.393 | 0.404 ± 0.043 | 0.124 ± 0.003 | 0.280 ± 0.026 | 0.179 ± 0.014 | 0.267 ± 0.015 | 0.530 ± 0.084 | 0.661 ± 0.117 |
| Mixtral-8x7B-Inst | 0.264 ± 0.004 | 9.453 ± 0.289 | 0.344 ± 0.004 | 0.127 ± 0.003 | 0.224 ± 0.005 | 0.179 ± 0.010 | 0.173 ± 0.009 | 0.348 ± 0.005 | 0.405 ± 0.007 |
| Mixtral-8x7B | 0.262 ± 0.008 | 9.785 ± 0.239 | 0.348 ± 0.012 | 0.146 ± 0.022 | 0.230 ± 0.016 | 0.153 ± 0.002 | 0.230 ± 0.041 | 0.354 ± 0.007 | 0.414 ± 0.009 |
| Qwen-2.5-3B-Inst | 0.978 ± 0.042 | 23.506 ± 0.294 | 1.782 ± 0.045 | 1.791 ± 0.069 | 2.978 ± 0.054 | 2.863 ± 0.033 | 3.239 ± 0.120 | 2.795 ± 0.086 | 2.654 ± 0.115 |
| Qwen-2.5-3B | 1.351 ± 0.036 | 23.357 ± 0.325 | 1.947 ± 0.045 | 1.864 ± 0.080 | 3.007 ± 0.061 | 2.997 ± 0.023 | 2.999 ± 0.145 | 2.604 ± 0.085 | 2.234 ± 0.114 |
| Qwen-2.5-1.5B-Inst | 2.153 ± 0.027 | 22.767 ± 0.365 | 2.052 ± 0.046 | 1.566 ± 0.033 | 2.671 ± 0.038 | 2.156 ± 0.035 | 3.635 ± 0.053 | 2.480 ± 0.085 | 2.323 ± 0.113 |
| Qwen-2.5-1.5B | 1.731 ± 0.036 | 20.358 ± 0.247 | 1.343 ± 0.061 | 1.737 ± 0.074 | 2.594 ± 0.042 | 2.256 ± 0.042 | 3.275 ± 0.132 | 2.036 ± 0.083 | 1.526 ± 0.114 |
| Qwen-2.5-0.5B-Inst | 1.938 ± 0.024 | 22.739 ± 0.244 | 1.743 ± 0.043 | 1.800 ± 0.021 | 2.193 ± 0.025 | 2.303 ± 0.028 | 3.439 ± 0.004 | 1.685 ± 0.084 | 1.398 ± 0.114 |
| Qwen-2.5-0.5B | 1.991 ± 0.024 | 22.311 ± 0.335 | 1.827 ± 0.045 | 0.950 ± 0.025 | 1.967 ± 0.020 | 2.799 ± 0.022 | 1.804 ± 0.036 | 1.695 ± 0.085 | 1.443 ± 0.113 |
| **Multimodal Models** | | | | | | | | | |
| UniTime | 0.371 ± 0.002 | 16.002 ± 0.121 | 0.271 ± 0.003 | 0.179 ± 0.001 | 0.318 ± 0.001 | 0.510 ± 0.003 | 0.333 ± 0.001 | 0.332 ± 0.001 | 0.384 ± 0.001 |
| Time-LLM (ETTh1) | 0.476 ± 0.001 | 19.636 ± 0.101 | 0.448 ± 0.002 | 0.192 ± 0.000 | 0.373 ± 0.000 | 0.538 ± 0.003 | 0.397 ± 0.001 | 0.382 ± 0.001 | 0.440 ± 0.001 |
| **TS Foundation Models\*** | | | | | | | | | |
| Lag-Llama | 0.329 ± 0.004 | 15.222 ± 0.288 | 0.355 ± 0.007 | 0.181 ± 0.003 | 0.324 ± 0.003 | 0.272 ± 0.006 | 0.342 ± 0.006 | 0.386 ± 0.009 | 0.449 ± 0.012 |
| Chronos | 0.326 ± 0.002 | 13.789 ± 0.179 | 0.385 ± 0.002 | 0.138 ± 0.002 | 0.288 ± 0.002 | 0.249 ± 0.002 | 0.295 ± 0.003 | 0.362 ± 0.003 | 0.417 ± 0.004 |
| TimeGEN | 0.354 ± 0.000 | 16.624 ± 0.127 | 0.402 ± 0.000 | 0.176 ± 0.000 | 0.308 ± 0.000 | 0.279 ± 0.000 | 0.324 ± 0.000 | 0.431 ± 0.000 | 0.431 ± 0.000 |
| Moirai | 0.520 ± 0.006 | 14.551 ± 0.321 | 0.414 ± 0.000 | 0.155 ± 0.004 | 0.260 ± 0.003 | 0.751 ± 0.015 | 0.276 ± 0.008 | 0.337 ± 0.007 | 0.397 ± 0.010 |
| **Statistical Models\*** | | | | | | | | | |
| ARIMA | 0.480 ± 0.006 | 14.502 ± 0.213 | 0.399 ± 0.006 | 0.160 ± 0.002 | 0.517 ± 0.012 | 0.522 ± 0.013 | 0.706 ± 0.026 | 0.354 ± 0.007 | 0.403 ± 0.010 |
| ETS | 0.522 ± 0.009 | 16.760 ± 0.238 | 0.407 ± 0.009 | 0.228 ± 0.010 | 0.682 ± 0.018 | 0.571 ± 0.019 | 0.855 ± 0.035 | 0.453 ± 0.012 | 0.479 ± 0.015 |
| Exp-Smoothing | 0.603 ± 0.013 | 17.440 ± 0.182 | 0.571 ± 0.021 | 0.334 ± 0.013 | 0.743 ± 0.018 | 0.557 ± 0.019 | 0.899 ± 0.035 | 0.673 ± 0.038 | 0.782 ± 0.053 |

### C.2    RESULTS PARTITIONED BY TYPES OF CONTEXT

Table Tab. 3 provides a view of the results partitioned by the types of context. One can observe that Direct Prompt - Llama-3.1-405B-Instruct achieves the best performance at tasks where the context involves intemporal, future or covariate information, while GPT-4o has an upper hand at tasks involving historical context information. LLMP with Llama-3-70B-Instruct achieves the best performance in tasks that involve causal information in the context. This provides a view complementary to that of partitioning by model capabilities (as in Tab. 1), and emphasizes that no single model is the best at processing all types of context, leaving room for advancements in models in the future.

**Table 3:** *Results on the CiK benchmark aggregated over all tasks and kinds of context. The first column shows the RCRPS averaged over all tasks. The second column shows the rank of each method w.r.t. other baselines, averaged over all tasks. The remaining columns show the average RCRPS stratified by context source. All averages are weighted according to the scheme described in Sec. 5.1 and accompanied by standard errors. Lower is better and the best means are in bold. * denotes models that do not use natural language context.*

| Model | Average RCRPS | Average Rank | $c_I$ | $c_H$ | $c_F$ | $c_{cov}$ | $c_{causal}$ |
|---|---|---|---|---|---|---|---|
| **Direct Prompt (ours)** | | | | | | | |
| Llama-3.1-405B-Inst | **0.159 ± 0.008** | **4.905 ± 0.254** | **0.174 ± 0.010** | 0.146 ± 0.001 | **0.085 ± 0.003** | **0.169 ± 0.010** | **0.398 ± 0.045** |
| Llama-3-70B-Inst | 0.518 ± 0.030 | 12.030 ± 0.246 | 0.621 ± 0.042 | 0.308 ± 0.064 | 0.301 ± 0.033 | 0.452 ± 0.032 | 0.704 ± 0.056 |
| Llama-3-8B-Inst | 1.647 ± 0.069 | 18.786 ± 0.235 | 2.355 ± 0.100 | 0.813 ± 0.115 | 1.332 ± 0.094 | 1.185 ± 0.087 | 2.041 ± 0.271 |
| Mixtral-8x7B-Inst | 1.061 ± 0.058 | 15.813 ± 0.296 | 1.263 ± 0.082 | 0.561 ± 0.111 | 0.691 ± 0.094 | 0.724 ± 0.053 | 1.232 ± 0.121 |
| GPT-4o | 0.276 ± 0.010 | **5.021 ± 0.180** | 0.220 ± 0.007 | **0.118 ± 0.001** | 0.108 ± 0.001 | 0.265 ± 0.012 | 0.858 ± 0.053 |
| GPT-4o-mini | 0.353 ± 0.022 | 9.792 ± 0.243 | 0.474 ± 0.035 | 0.139 ± 0.002 | 0.141 ± 0.001 | 0.345 ± 0.030 | 0.644 ± 0.128 |
| Qwen-2.5-7B-Inst | 0.292 ± 0.032 | 11.810 ± 0.985 | 0.295 ± 0.031 | 0.196 ± 0.029 | 0.262 ± 0.058 | 0.252 ± 0.027 | 0.516 ± 0.083 |
| Mistral-7B-Inst | 1.943 ± 0.117 | 19.691 ± 0.843 | 1.892 ± 0.128 | 0.869 ± 0.145 | 2.576 ± 0.191 | 1.828 ± 0.155 | 1.042 ± 0.323 |
| **LLMP** | | | | | | | |
| Llama-3-70B-Inst | 0.550 ± 0.013 | 9.207 ± 0.254 | 0.455 ± 0.018 | 0.516 ± 0.028 | 0.690 ± 0.018 | 0.588 ± 0.018 | **0.392 ± 0.028** |
| Llama-3-70B | 0.237 ± 0.006 | 7.344 ± 0.290 | 0.213 ± 0.005 | 0.121 ± 0.008 | 0.233 ± 0.012 | 0.198 ± 0.004 | **0.360 ± 0.011** |
| Llama-3-8B-Inst | 0.484 ± 0.010 | 10.875 ± 0.204 | 0.477 ± 0.013 | 0.161 ± 0.006 | 0.264 ± 0.003 | 0.316 ± 0.008 | 0.878 ± 0.035 |
| Llama-3-8B | 0.313 ± 0.023 | 10.924 ± 0.393 | 0.334 ± 0.035 | 0.123 ± 0.004 | 0.232 ± 0.012 | 0.291 ± 0.031 | 0.739 ± 0.134 |
| Mixtral-8x7B-Inst | 0.264 ± 0.004 | 9.453 ± 0.289 | 0.242 ± 0.007 | 0.173 ± 0.004 | 0.268 ± 0.009 | 0.220 ± 0.002 | 0.437 ± 0.007 |
| Mixtral-8x7B | 0.262 ± 0.008 | 9.785 ± 0.239 | 0.250 ± 0.008 | 0.119 ± 0.003 | 0.254 ± 0.013 | 0.229 ± 0.007 | 0.457 ± 0.011 |
| Qwen-2.5-3B-Inst | 0.978 ± 0.042 | 23.506 ± 0.294 | 2.780 ± 0.046 | 2.718 ± 0.067 | 1.865 ± 0.023 | 2.088 ± 0.038 | 2.501 ± 0.130 |
| Qwen-2.5-3B | 1.351 ± 0.036 | 23.357 ± 0.325 | 2.962 ± 0.046 | 3.488 ± 0.057 | 2.163 ± 0.018 | 1.912 ± 0.039 | 1.897 ± 0.129 |
| Qwen-2.5-1.5B-Inst | 2.153 ± 0.027 | 22.767 ± 0.365 | 2.605 ± 0.041 | 1.672 ± 0.055 | 1.434 ± 0.026 | 2.024 ± 0.035 | 2.448 ± 0.128 |
| Qwen-2.5-1.5B | 1.731 ± 0.036 | 20.358 ± 0.247 | 2.337 ± 0.049 | 2.982 ± 0.052 | 1.109 ± 0.052 | 1.457 ± 0.047 | 1.304 ± 0.129 |
| Qwen-2.5-0.5B-Inst | 1.938 ± 0.024 | 22.739 ± 0.244 | 2.445 ± 0.038 | 1.960 ± 0.063 | 1.616 ± 0.012 | 1.715 ± 0.032 | 1.199 ± 0.129 |
| Qwen-2.5-0.5B | 1.991 ± 0.024 | 22.311 ± 0.335 | 2.539 ± 0.039 | 2.083 ± 0.052 | 1.743 ± 0.012 | 1.721 ± 0.032 | 1.225 ± 0.128 |
| **Multimodal Models** | | | | | | | |
| UniTime | 0.371 ± 0.002 | 16.002 ± 0.121 | 0.455 ± 0.002 | 0.154 ± 0.000 | 0.226 ± 0.003 | 0.396 ± 0.001 | 0.422 ± 0.001 |
| TimeLLM (ETTh1) | 0.476 ± 0.001 | 19.636 ± 0.101 | 0.517 ± 0.002 | 0.183 ± 0.000 | 0.376 ± 0.002 | 0.446 ± 0.001 | 0.482 ± 0.001 |
| **TS Foundation Models*** | | | | | | | |
| Lag-Llama | 0.329 ± 0.004 | 15.222 ± 0.288 | 0.333 ± 0.005 | 0.167 ± 0.005 | 0.277 ± 0.006 | 0.301 ± 0.004 | 0.495 ± 0.014 |
| Chronos | 0.326 ± 0.002 | 13.789 ± 0.179 | 0.314 ± 0.002 | 0.179 ± 0.003 | 0.316 ± 0.002 | 0.252 ± 0.002 | 0.460 ± 0.004 |
| TimeGEN | 0.354 ± 0.000 | 16.624 ± 0.127 | 0.333 ± 0.000 | 0.177 ± 0.000 | 0.348 ± 0.000 | 0.291 ± 0.000 | 0.474 ± 0.000 |
| Moirai | 0.520 ± 0.006 | 14.551 ± 0.321 | 0.596 ± 0.009 | 0.140 ± 0.001 | 0.364 ± 0.002 | 0.510 ± 0.008 | 0.438 ± 0.011 |
| **Statistical Models*** | | | | | | | |
| ARIMA | 0.480 ± 0.006 | 14.502 ± 0.213 | 0.565 ± 0.010 | 0.200 ± 0.007 | 0.307 ± 0.003 | 0.390 ± 0.006 | 0.440 ± 0.011 |
| ETS | 0.522 ± 0.009 | 16.760 ± 0.238 | 0.627 ± 0.014 | 0.362 ± 0.014 | 0.323 ± 0.008 | 0.401 ± 0.010 | 0.508 ± 0.017 |
| Exp-Smoothing | 0.603 ± 0.013 | 17.440 ± 0.182 | 0.700 ± 0.020 | 0.493 ± 0.016 | 0.438 ± 0.009 | 0.492 ± 0.017 | 0.827 ± 0.060 |

## C.3 EXTENDED RESULTS ON ALL MODELS

**Table 4:** *Extended results on the CiK benchmark aggregated over all tasks. The first column shows the RCRPS averaged over all tasks. The second column shows the rank of each method w.r.t. other baselines, averaged over all tasks. All averages are weighted according to the scheme described in Sec. 5.1 and accompanied by standard errors. Lower is better and the best means are in bold.*

| Model | Average RCRPS | Average Rank |
|---|---|---|
| **With Context** | | |
| Direct Prompt (ours) | | |
| Llama-3.1-405B-Inst | **0.159 ± 0.008** | **7.337 ± 0.524** |
| Llama-3-70B-Inst | 0.518 ± 0.030 | 20.916 ± 0.497 |
| Llama-3-8B-Inst | 1.647 ± 0.069 | 35.232 ± 0.497 |
| Mixtral-8x7B-Inst | 1.061 ± 0.058 | 29.273 ± 0.628 |
| GPT-4o | 0.276 ± 0.010 | **8.425 ± 0.377** |
| GPT-4o-mini | 0.353 ± 0.022 | 15.699 ± 0.450 |
| Qwen-2.5-7B-Inst | 0.292 ± 0.032 | 20.167 ± 2.124 |
| Mistral-7B-Inst | 1.943 ± 0.117 | 38.038 ± 1.755 |
| LLMP | | |
| Llama-3-70B-Inst | 0.550 ± 0.013 | 16.226 ± 0.484 |
| Llama-3-70B | 0.237 ± 0.006 | 11.473 ± 0.614 |
| Llama-3-8B-Inst | 0.484 ± 0.010 | 17.519 ± 0.431 |
| Llama-3-8B | 0.313 ± 0.023 | 17.529 ± 0.825 |
| Mixtral-8x7B-Inst | 0.264 ± 0.004 | 14.645 ± 0.534 |
| Mixtral-8x7B | 0.262 ± 0.008 | 15.233 ± 0.447 |
| Qwen-2.5-3B-Inst | 0.978 ± 0.042 | 45.344 ± 0.682 |
| Qwen-2.5-3B | 1.351 ± 0.036 | 45.157 ± 0.755 |
| Qwen-2.5-1.5B-Inst | 2.153 ± 0.027 | 44.344 ± 0.791 |
| Qwen-2.5-1.5B | 1.731 ± 0.036 | 38.889 ± 0.487 |
| Qwen-2.5-0.5B-Inst | 1.938 ± 0.024 | 44.018 ± 0.552 |
| Qwen-2.5-0.5B | 1.991 ± 0.024 | 42.701 ± 0.768 |
| Multimodal Models | | |
| UniTime | 0.371 ± 0.002 | 30.402 ± 0.181 |
| Time-LLM (ETTh1) | 0.476 ± 0.001 | 38.066 ± 0.162 |
| **Without Context** | | |
| Direct Prompt (ours) | | |
| Llama-3.1-405B-Inst | 0.473 ± 0.005 | 30.266 ± 0.286 |
| Llama-3-70B-Inst | 0.714 ± 0.035 | 34.375 ± 0.520 |
| Llama-3-8B-Inst | 1.900 ± 0.059 | 44.040 ± 0.366 |
| Mixtral-8x7B-Inst | 0.847 ± 0.045 | 32.912 ± 0.693 |
| GPT-4o | 0.441 ± 0.008 | 27.886 ± 0.357 |
| GPT-4o-mini | 0.423 ± 0.006 | 31.602 ± 0.265 |
| Qwen-2.5-7B-Inst | 0.377 ± 0.034 | 27.707 ± 2.272 |
| Mistral-7B-Inst | 1.752 ± 0.094 | 38.969 ± 1.923 |
| LLMP | | |
| Llama-3-70B-Inst | 0.378 ± 0.004 | 23.404 ± 0.430 |
| Llama-3-70B | 0.312 ± 0.006 | 19.951 ± 0.445 |
| Llama-3-8B-Inst | 0.503 ± 0.009 | 27.800 ± 0.406 |
| Llama-3-8B | 0.345 ± 0.003 | 22.766 ± 0.358 |
| Mixtral-8x7B-Inst | 0.383 ± 0.015 | 22.097 ± 0.424 |
| Mixtral-8x7B | 0.306 ± 0.007 | 20.539 ± 0.456 |
| Qwen-2.5-3B-Inst | 2.356 ± 0.029 | 32.875 ± 0.910 |
| Qwen-2.5-3B | 2.315 ± 0.029 | 36.915 ± 0.887 |
| Qwen-2.5-1.5B-Inst | 1.515 ± 0.033 | 40.771 ± 0.960 |
| Qwen-2.5-1.5B | 1.069 ± 0.028 | 35.309 ± 0.961 |
| Qwen-2.5-0.5B-Inst | 1.318 ± 0.037 | 38.513 ± 0.676 |
| Qwen-2.5-0.5B | 1.819 ± 0.027 | 42.033 ± 0.674 |
| Multimodal Models | | |
| UniTime | 0.405 ± 0.002 | 32.199 ± 0.183 |
| Time-LLM (ETTh1) | 0.454 ± 0.002 | 36.339 ± 0.168 |
| TS Foundation Models | | |
| Lag-Llama | 0.329 ± 0.004 | 27.480 ± 0.715 |
| Chronos-Tiny | 0.328 ± 0.001 | 24.606 ± 0.411 |
| Chronos-Mini | 0.341 ± 0.001 | 25.776 ± 0.397 |
| Chronos-Small | 0.328 ± 0.002 | 23.594 ± 0.339 |
| Chronos-Base | 0.672 ± 0.003 | 27.366 ± 0.344 |
| Chronos-Large | 0.326 ± 0.002 | 22.871 ± 0.378 |
| TimeGEN | 0.354 ± 0.000 | 31.949 ± 0.183 |
| Moirai-Small | 0.565 ± 0.031 | 31.616 ± 0.399 |
| Moirai-Base | 0.624 ± 0.013 | 31.112 ± 0.329 |
| Moirai-Large | 0.520 ± 0.006 | 25.428 ± 0.824 |
| Statistical Models | | |
| ARIMA | 0.480 ± 0.006 | 24.232 ± 0.446 |
| ETS | 0.522 ± 0.009 | 29.589 ± 0.552 |
| Exp-Smoothing | 0.603 ± 0.013 | 31.480 ± 0.323 |

## C.4 SIGNIFICANT FAILURES PER MODEL

We observe that in a few instances in the benchmark, some models tend to obtain significantly worse performance when evaluated with context. In our evaluation, we term all instances where the RCRPS value of a model is greater than 5, as significant failures of the model on those instances. We found 5 as a suitable value for analyzing such failures, as it intuitively represents the value a forecast would get if the distance between the forecast and the ground-truth was 5 times bigger than the range of the ground-truth for the task. When we aggregate the RCRPS of instances in the benchmark (such as in Tab. 1), we cap the RCRPS of such significant failures to 5, to avoid outliers with a much higher RCRPS affecting the aggregate score. In Tab. 5, we show the number of such instances in our evaluation of the benchmark where we found models to have significant failures (out of a total of 355 evaluated instances). Interestingly, some models such as Direct Prompt with Llama-3.1-405B-Instruct and LLMP with Llama-3-70B and Llama-3-8B are more robust to such significant failures, and do not incur such failures. On the other hand, models such as Qwen family of models (that are notably significantly smaller than the rest) with LLMP achieve the most significant failures, followed by Llama-3-70B-Inst and Llama-3-8B-Inst with LLMP. We postulate that this is because of models misinterpreting context. It is still an open question as to how to increase the robustness of models to prevent or reduce such significant failures. We visualize such significant failures in Appendix C.6.

**Table 5:** *Number of instances with significant failures in models that support context*

| Model | Number of instances with significant failures |
|---|:---:|
| Direct Prompt (ours) | |
| Llama-3.1-405B-Inst | 0 |
| Llama-3-70B-Inst | 1 |
| Llama-3-8B-Inst | 3 |
| Mixtral-8x7B-Inst | 5 |
| GPT-4o | 5 |
| GPT-4o-mini | 2 |
| Qwen-2.5-7B-Inst | 1 |
| Mistral-7B-Inst | 2 |
| LLMP | |
| Llama-3-70B-Inst | 18 |
| Llama-3-70B | 0 |
| Llama-3-8B-Inst | 12 |
| Llama-3-8B | 0 |
| Mixtral-8x7B-Inst | 1 |
| Mixtral-8x7B | 1 |
| Qwen-2.5-3B-Inst | 115 |
| Qwen-2.5-3B | 150 |
| Qwen-2.5-1.5B-Inst | 95 |
| Qwen-2.5-1.5B | 102 |
| Qwen-2.5-0.5B-Inst | 102 |
| Qwen-2.5-0.5B | 111 |
| Multimodal Models | |
| UniTime | 0 |
| Time-LLM (ETTh1) | 2 |

## C.5 VISUALIZATIONS OF SUCCESSFUL CONTEXT-AWARE FORECASTS

> **Context:** " This series represents the occupancy rate (%) captured by a highway sensor.
> Consider that the meter will be offline for maintenance between 2024-04-11 13:00:00 and 2024-04-11
> 15:00:00, which results in zero readings. "

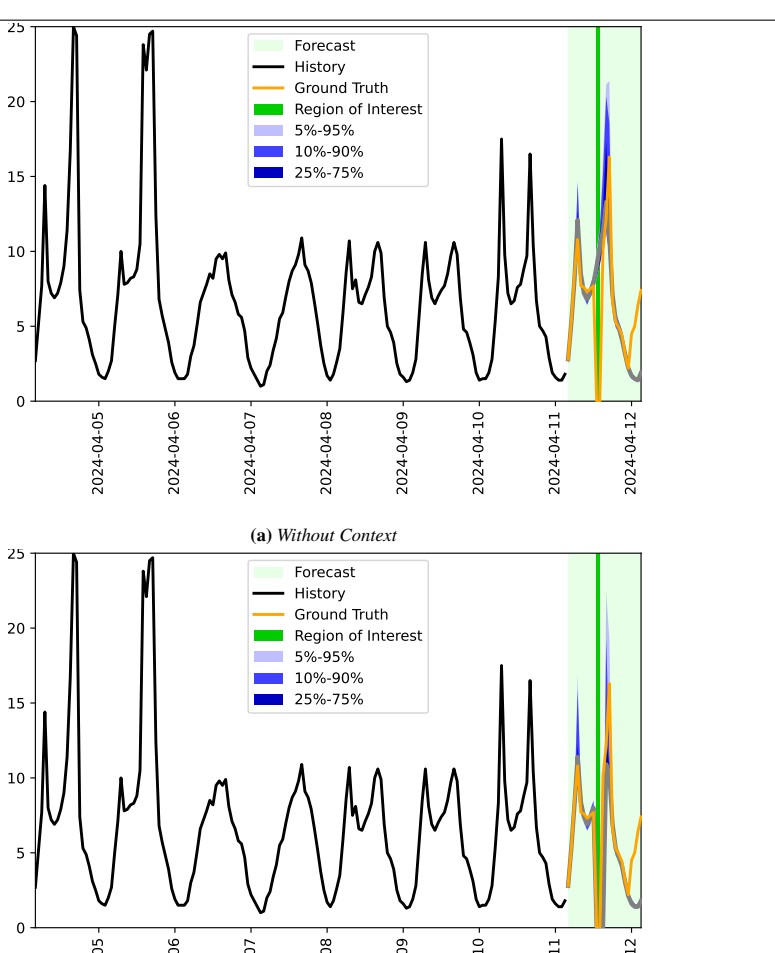

**(a)** *Without Context*

**(b)** *With Context*

**Figure 9:** *Example of successful context-aware forecasting by Direct Prompt with Llama-3.1-405B-Instruct*

1620
1621
1622
1623
1624
1625
1626
1627
1628
1629
1630
1631
1632
1633
1634
1635
1636
1637

**Context:** *" This series contains Diffuse Horizontal Irradiance for a location in Sinaloa, Mexico. The Diffuse Horizontal Irradiance is the total amount of sun energy (in Watts per squared meter) arriving indirectly on a horizontal surface, ignoring the direct sunlight. Even when there are no clouds to scatter the sun light, there will still be some Diffuse Horizontal Irradiance, since clouds are not the only cause of light scattering. When there are no clouds, the Diffuse Horizontal Irradiance is mostly a function of the position of the sun in the sky, with only small variations from factors such as water vapour and dust particles levels. If the cloud cover is light, the Diffuse Horizontal Irradiance will increase due to the increase scattering of sun light, but heavy cloud cover will decrease it due to some sun light no longer being able to reach the ground.*
*At the beginning of the series, the weather was cloudy.*
*At 2022-07-12 11:00:00, the weather became clear.*
*At 2022-07-12 19:00:00, the weather became cloudy.*
*At 2022-07-13 12:00:00, the weather became clear.*
*At 2022-07-13 13:00:00, the weather became cloudy.*
*At 2022-07-14 06:00:00, we expect that the weather will become clear.*
*At 2022-07-14 07:00:00, we expect that the weather will become cloudy.*
*At 2022-07-14 10:00:00, we expect that the weather will become clear.*
*At 2022-07-14 18:00:00, we expect that the weather will become cloudy. "*

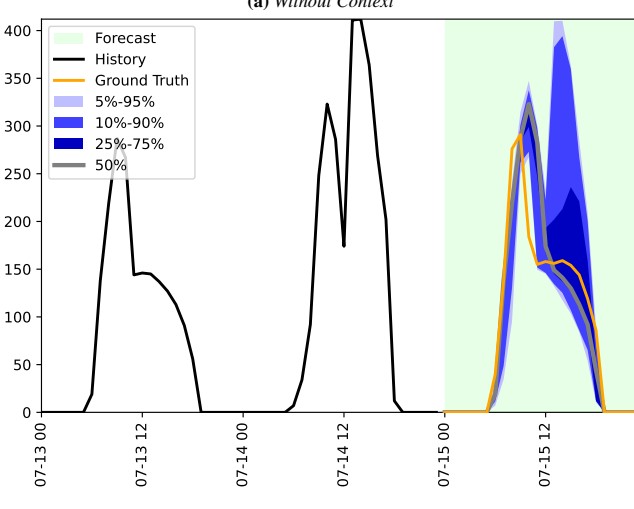

**(a)** *Without Context*

**(b)** *With Context*

**Figure 10:** *Example of successful context-aware forecasting by Direct Prompt with Llama-3.1-405B-Instruct*

**Context:** *" This is the number of cash withdrawals from an automated teller machine (ATM) in an arbitrary location in England.*
*Consider that the building which contains the ATM is closed from 1997-09-05 00:00:00, for 8 days. "*

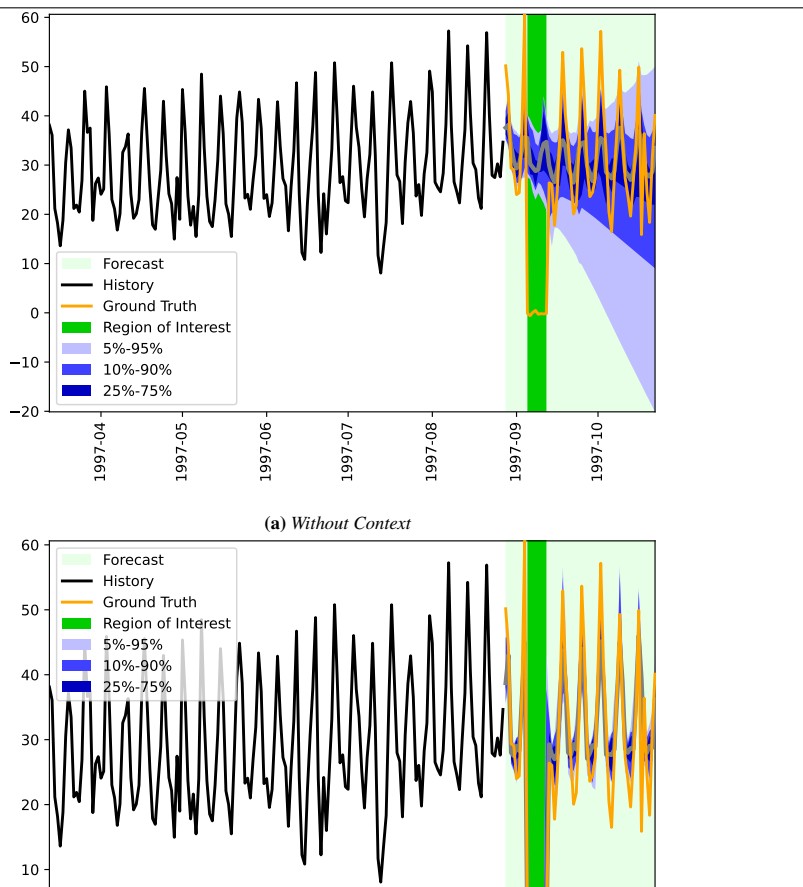

**(a)** *Without Context*

**(b)** *With Context*

**Figure 11:** *Example of successful context-aware forecasts by Direct Prompt with GPT-4o*

*Context:* " *The Montreal Fire Department is in charge of responding to various kind of public safety incidents. This is the number of field fire incidents responded to by Montreal firefighters in the Rivière-des-Prairies-Pointe-aux-Trembles borough. In other years, the yearly average number of incidents was 106 with the busiest month being June.*
*The Mayor is determined to completely eradicate this kind of incident. Fortunately, the city's public safety research group identified that field fires and trash fires tend to co-occur. When the amount of field fires increases, the amount of trash fires also tends to increase. The same holds when they decrease.*
*The Mayor has a plan: they will implement daily spraying of all piles of trash with water starting on 2022-06.* "

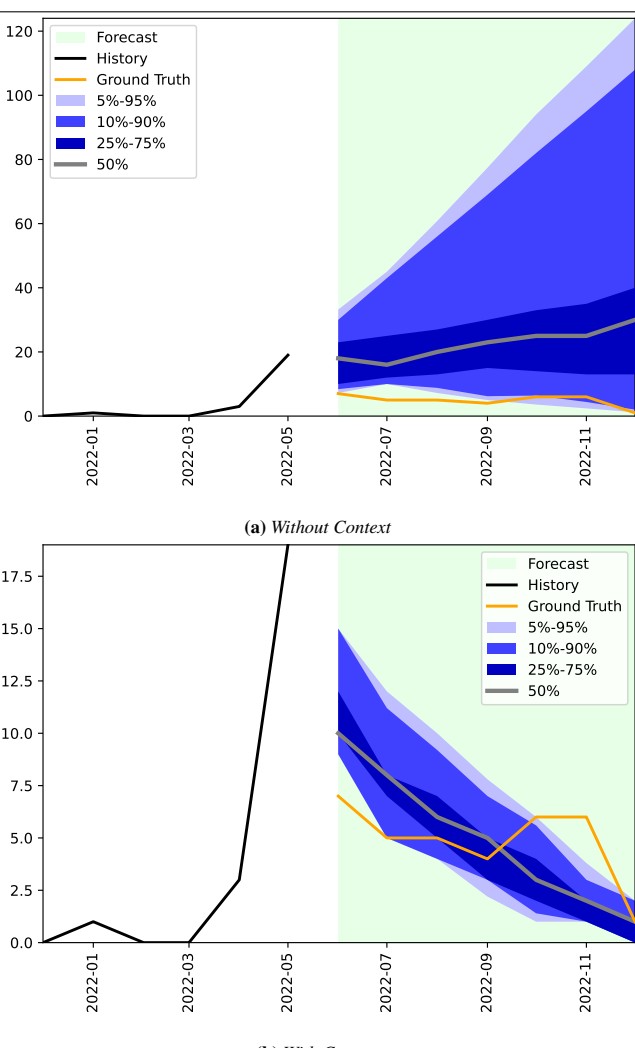

**(a)** *Without Context*

**(b)** *With Context*

**Figure 12:** *Example of successful context-aware forecasts by Direct Prompt with GPT-4o*

**Context:** " *This is the Unemployment Rate for Okaloosa County, in Florida.*
*For reference, here is the Unemployment Rate for a few American states during the same period:*
*Pennsylvania*
———————

*(2023-08-01 00:00:00, 4.2)*
*(2023-09-01 00:00:00, 3.0)*
*(2023-10-01 00:00:00, 3.1)*
*(2023-11-01 00:00:00, 2.9)*
*(2023-12-01 00:00:00, 2.9)*
*(2024-01-01 00:00:00, 3.5)*
*(2024-02-01 00:00:00, 3.7)*
*(2024-03-01 00:00:00, 3.4)*
*(2024-04-01 00:00:00, 2.9)*
*(2024-05-01 00:00:00, 3.2)*
*(2024-06-01 00:00:00, 3.7)*
*(2024-07-01 00:00:00, 4.0)*

*Florida*
———————

*(2023-08-01 00:00:00, 3.3)*
*(2023-09-01 00:00:00, 3.1)*
*(2023-10-01 00:00:00, 3.1)*
*(2023-11-01 00:00:00, 3.0)*
*(2023-12-01 00:00:00, 2.9)*
*(2024-01-01 00:00:00, 3.1)*
*(2024-02-01 00:00:00, 3.1)*
*(2024-03-01 00:00:00, 3.3)*
*(2024-04-01 00:00:00, 3.1)*
*(2024-05-01 00:00:00, 2.9)*
*(2024-06-01 00:00:00, 3.5)*
*(2024-07-01 00:00:00, 3.8)*

*Wisconsin*
———————

*(2023-08-01 00:00:00, 3.4)*
*(2023-09-01 00:00:00, 2.9)*
*(2023-10-01 00:00:00, 2.8)*
*(2023-11-01 00:00:00, 2.7)*
*(2023-12-01 00:00:00, 2.9)*
*(2024-01-01 00:00:00, 2.8)*
*(2024-02-01 00:00:00, 3.3)*
*(2024-03-01 00:00:00, 3.5)*
*(2024-04-01 00:00:00, 3.0)*
*(2024-05-01 00:00:00, 3.0)*
*(2024-06-01 00:00:00, 3.3)*
*(2024-07-01 00:00:00, 3.3)* "

**(a)** *Without Context*

**(b)** *With Context*

**Figure 13:** *Example of successful context-aware forecasts by LLMP with Mixtral-8x7B-Instruct*

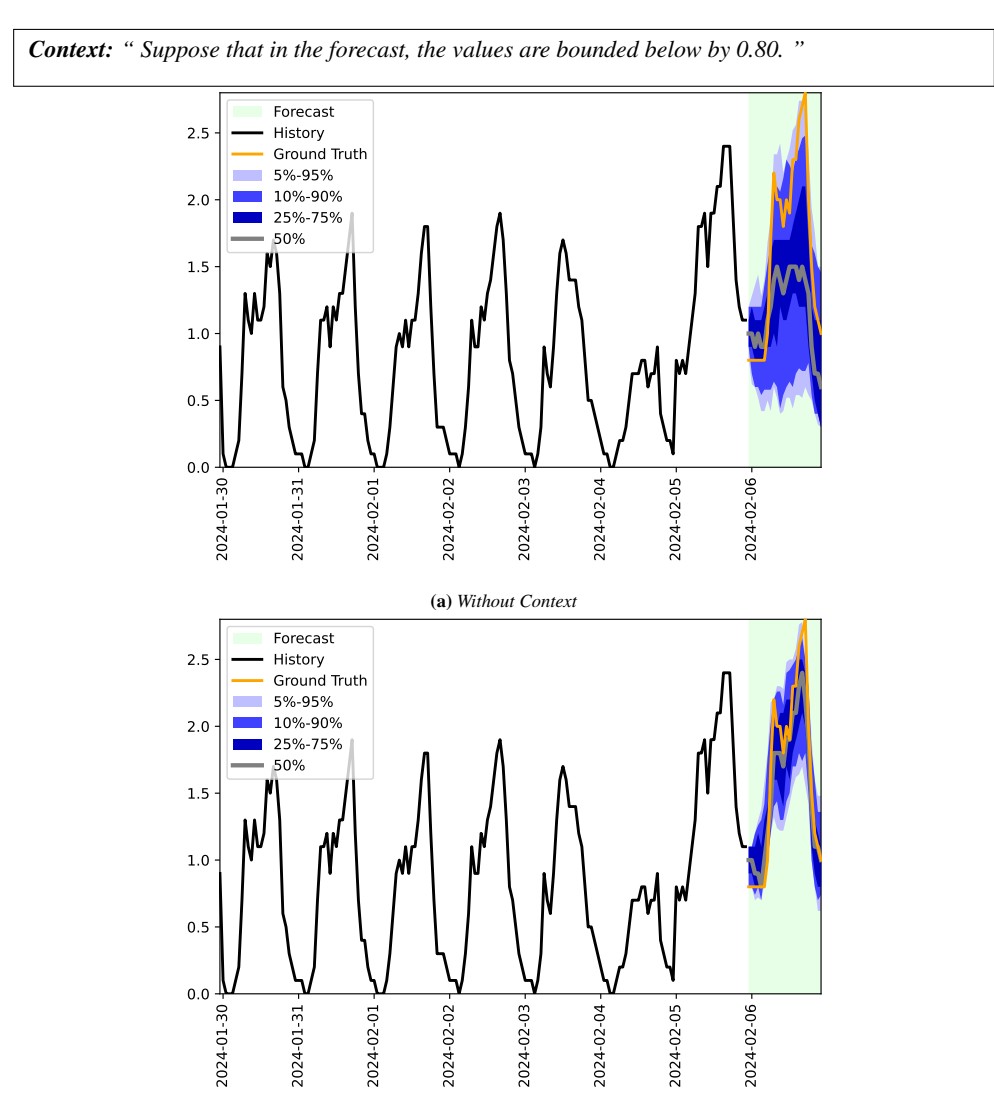

**Context:** " Suppose that in the forecast, the values are bounded below by 0.80. "

**(a)** *Without Context*

**(b)** *With Context*

**Figure 14:** *Example of successful context-aware forecasts by LLMP with Mixtral-8x7B-Instruct*

**Context:** *" This series contains the amount of sunlight (in Watts per squared meter) arriving on a horizontal surface, for a location in Alaska, United States. "*

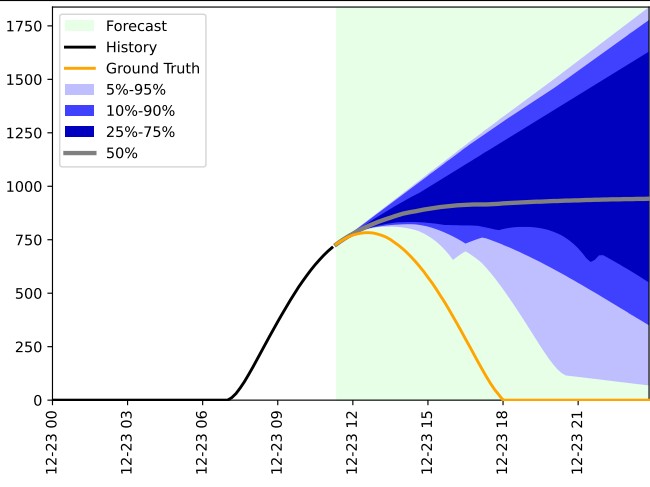

**(a)** *Without Context*

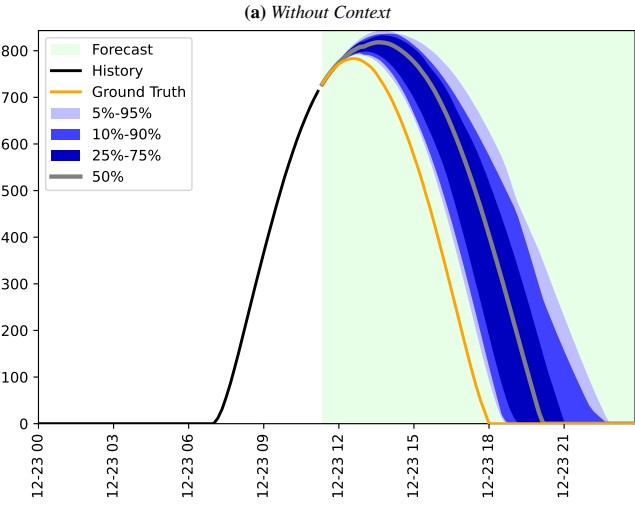

**(b)** *With Context*

**Figure 15:** *Example of successful context-aware forecasts by LLMP with Llama-70B*

*Context:* " *The Montreal Fire Department is in charge of responding to various kind of public safety incidents. This series contains the number of field fire incidents responded to by the Montreal Fire Department in the Rosemont-La Petite-Patrie borough. On average, they respond to 58 incidents per year and the month with the most incidents was June.*

*The Mayor is determined to completely eradicate this kind of incident. Fortunately, the city's public safety research group, a team of highly qualified experts, identified that field fires and gas leaks tend to co-occur. When the amount of field fires increases, the amount of gas leaks also tends to increase. The same holds when they decrease.*

*The Mayor has a plan: they will implement a strict prohibition of using any form of combustible gas in the city starting on 2023-06. In a recent interview, they claimed, "This is a bulletproof plan, and I am certain it will immediately put an end to field fires." "*

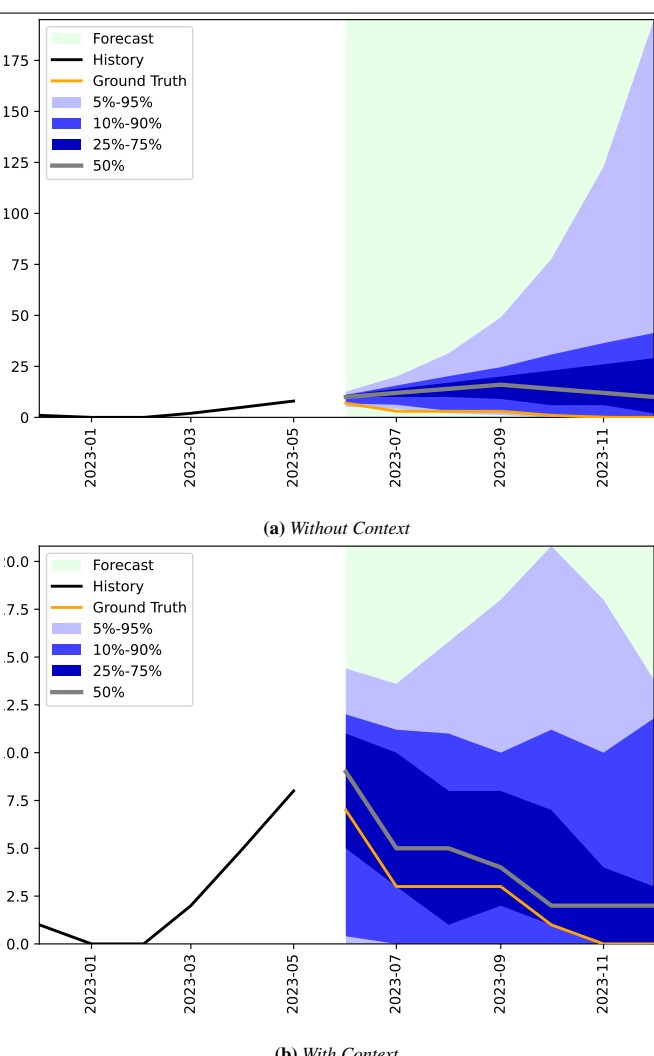

**(a)** *Without Context*

**(b)** *With Context*

**Figure 16:** *Example of successful context-aware forecasts by LLMP with Llama-70B*

## C.6 Visualizations of Significant Failures

> ***Context:*** *" Given are variables X_0 and X_1, where X_0 is a covariate and X_1 is the variable to forecast. Variables are generated from a linear Structural Vector Autoregressive (SVAR) model with additive gauss noise and a noise scale of 1.487e-03, with lag = 3.*
> *The task is to forecast the value of the variable X_1 at time t, given the values of the covariate X_0 and the variable X_1 itself at times t-1, ... t-3. For the first 128 days, the covariate X_0 takes a value of 8 from 2024-02-21 to 2024-03-11, 12 from 2024-03-12 to 2024-05-06, 12 from 2024-05-07 to 2024-06-27. For the next 32 days, the covariate X_0 takes a value of 30 from 2024-06-28 to 2024-07-13, 60 from 2024-07-14 to 2024-07-14, 60 from 2024-07-15 to 2024-07-29. Each day can be treated as a timestep for the forecasting task. The causal parents affect the child variables at different lags.*
> *The causal parents for each variable is given below:*
> *No parents for X_0 at any lag.*
> *Parents for X_1 at lag 1: ['X_0', 'X_1'] affect the forecast variable as 0.527 \* X_0 + -0.895 \* X_1.*
> *Parents for X_1 at lag 2: ['X_0', 'X_1'] affect the forecast variable as 1.380 \* X_0 + -0.758 \* X_1.*
> *Parents for X_1 at lag 3: ['X_0', 'X_1'] affect the forecast variable as -0.661 \* X_0 + -0.793 \* X_1. "*

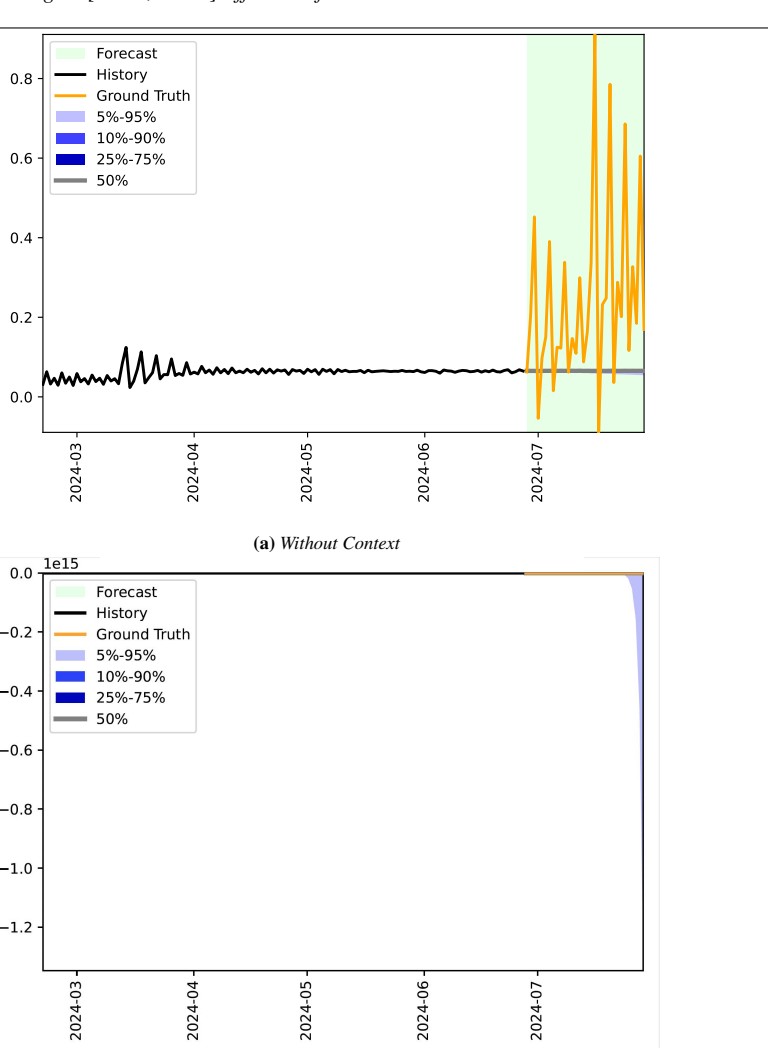

**(a)** *Without Context*

**(b)** *With Context*

**Figure 17:** *Example to show a significant failure case of Direct Prompt with GPT-4o where its performance worsens with context*

*Context:* " *This series contains the road occupancy rates on a freeway in the San Francisco Bay area. The days for which the forecast is required are Thursday 2024-07-04, Friday 2024-07-05, Saturday 2024-07-06. Note that 2024-07-04 is a holiday due to Independence Day. Note that traffic on this freeway typically reduces on holidays.* "

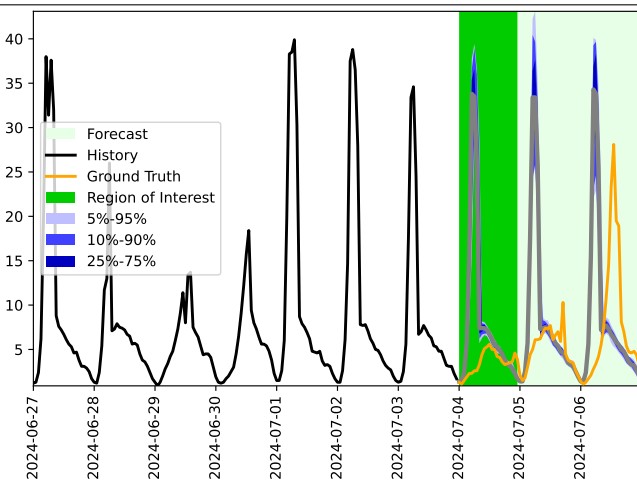

**(a)** *Without Context*

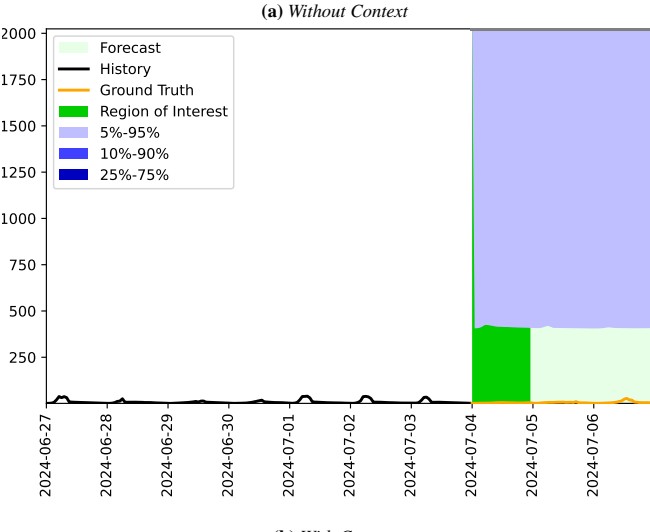

**(b)** *With Context*

**Figure 18:** *Example to show a significant failure case of LLMP with Llama-3-70B where its performance worsens with context*

**Context:** " *This series represents the occupancy rate (%) captured by a highway sensor. The sensor had a calibration problem starting from 2024-04-20 13:00:00 which resulted in an additive trend in the series that increases by 0.0072 at every hour. At timestep 2024-04-24 13:00:00, the sensor was repaired and this additive trend will disappear.* "

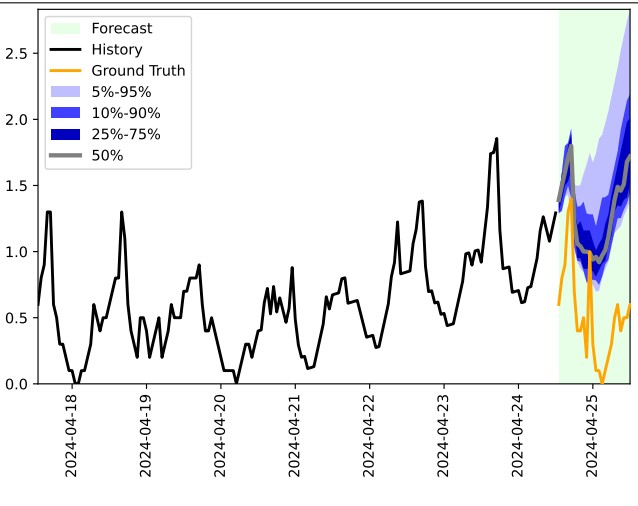

**(a)** *Without Context*

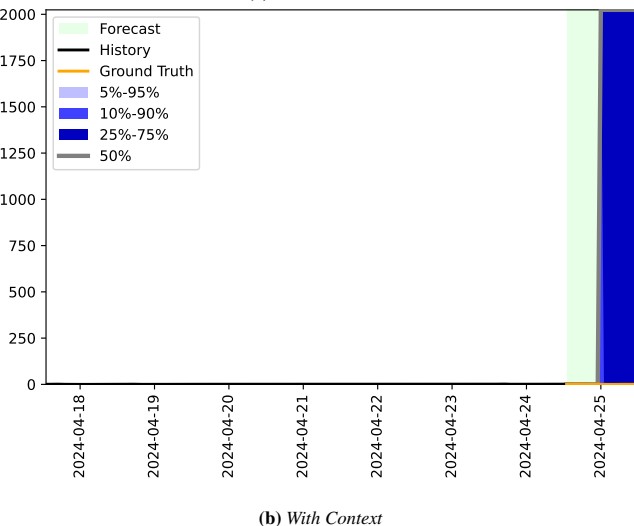

**(b)** *With Context*

**Figure 19:** *Example to show a significant failure case of LLMP with Llama-3-70B where its performance worsens with context*

***Context:*** *" The Montreal Fire Department is in charge of responding to various kind of public safety incidents. This series contains the number of field fire incidents responded to by the Montreal Fire Department in the L'Île-Bizard-Sainte-Geneviève borough. On average, they respond to 19 incidents per year with the busiest month being June.*

*The Mayor is determined to completely eradicate this kind of incident. Fortunately, the city's public safety research group, a team of highly qualified experts, identified that field fires and trash fires tend to co-occur. When the amount of field fires increases, the amount of trash fires also tends to increase. The same holds when they decrease.*

*The Mayor has a plan: they will implement daily spraying of all piles of trash with fire retardant foam starting on 2023-06. In a recent interview, they claimed, "This is a bulletproof plan, and I am certain it will immediately put an end to field fires." "*

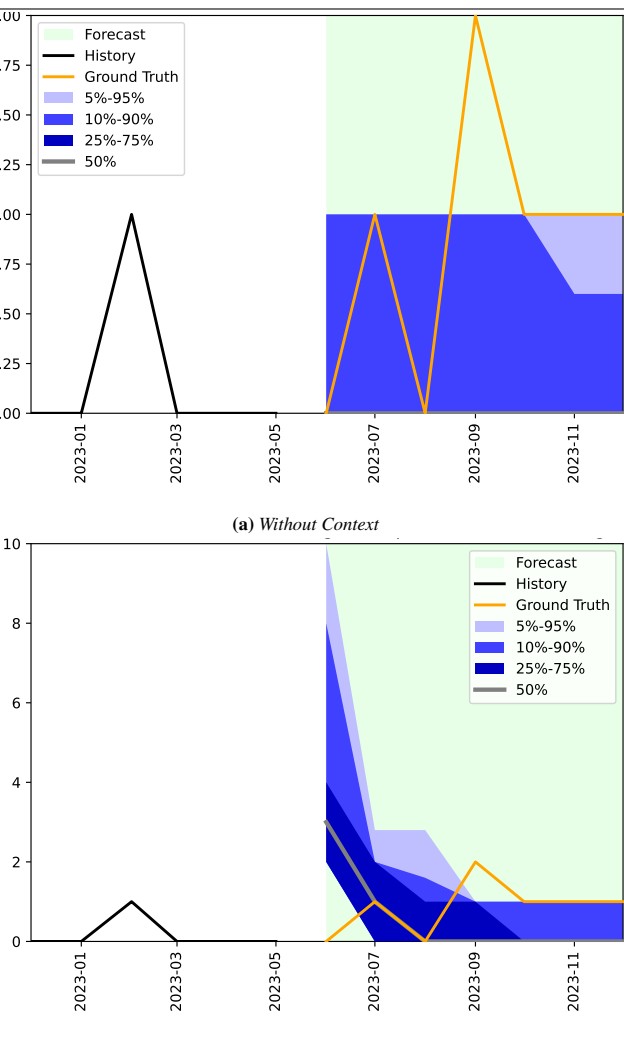

**Figure 20:** *Example to show a significant failure case of Direct Prompt with Llama-3-8B-Instruct where it misinterprets the context*

### C.7 COST OF API-BASED MODELS

Tab. 6 provides the cost incurred in evaluating GPT-4o (version gpt-4o-2024-05-13) and GPT-4o-mini (version gpt-4o-mini-2024-07-18) with the Direct Prompt method on CiK (as per the evaluation protocol used, described in Sec. 5.1).

**Table 6:** *Costs ($CAD) of evaluating the GPT-4o family of models on CiK. "Total" represents the total cost of evaluating each model on the CiK benchmark. The "Per-instance average" and the "Per-instance median" are the average and median cost of running a single instance for a given task, in other words the average and median cost of generating 25 sample trajectories for a given example of a task. As a reminder, each task in CiK is evaluated over 5 instances in our evaluation protocol.*

| Model | Total | Per-instance average | Per-instance median |
|---|---|---|---|
| GPT-4o | $143.83 | $0.288 | $0.170 |
| GPT-4o (no context) | $139.50 | $0.279 | $0.160 |
| GPT-4o-mini | $13.79 | $0.040 | $0.040 |
| GPT-4o-mini (no context) | $13.32 | $0.038 | $0.040 |

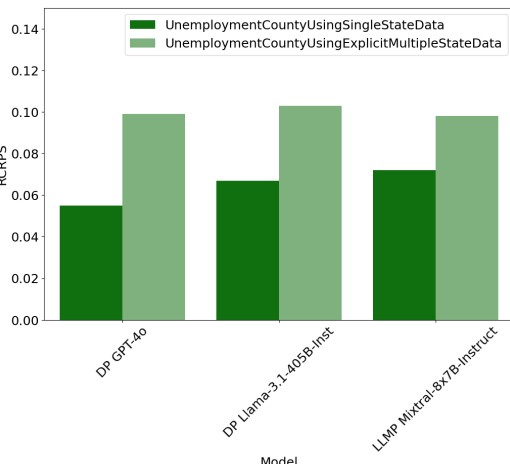

**Figure 21:** *A comparison of RCRPS (lower is better) for two tasks on predicting the Unemployment Rate of a county. Both contain the context needed to solve the task. However, the* `UnemploymentCountyUsingSingleStateData` *task (dark green) is filtered to only contain the relevant context. Other the other hand, the* `UnemploymentCountyUsingExpliciteMultipleStateData` *task (light green) also contains other unrelated context. We visualize three models here, all of which perform better when the context only includes the most relevant information.*

### C.8 IMPACT OF RELEVANT AND IRRELEVANT INFORMATION IN CONTEXT

We study here if models perform better on context that has already been filtered to only contain relevant information. To assess this, we compare two tasks on predicting the Unemployment Rate of a county.

1. For the `UnemploymentCountyUsingSingleStateData` task, the context contains the unemployment rate of the state which the county belongs to, tagged with the name of the state. This task can be visualized at `https://anon-forecast.github.io/benchmark_report_dev/UnemploymentCountyUsingSingleStateData.html`.

2. In the `UnemploymentCountyUsingExpliciteMultipleStateData` task, in addition to the unemployment rate of the parent state of the county, the context includes unemployment rates of 2 other randomly selected states, also tagged with the name of the states. This task can be visualized at `https://anon-forecast.github.io/benchmark_report_dev/UnemploymentCountyUsingExplicitMultipleStateData.html`.

Results of three randomly picked models from the benchmark is visualized in Fig. 21. We find that models perform much better when only the relevant state's data is provided, as opposed to the context also containing data from other states.

## C.9 Impact of Solely Irrelevant Information in Context

Many of our tasks include covariates in its context which are highly useful for the models to accurately predict the target time series. One question is: Do the LLM-based models perform well for such tasks due to correctly understanding that said covariates are helpful or because they blindly use the provided data without asking themselves if the data is actually relevant?

As a way to get some insight on this question, we took a task where the models have to forecast the unemployment data of an American county, given the unemployment data of the state the county is in (Task *UnemploymentCountyUsingSingleStateData*). We then modify this task by first trying to mislead the model by wrongly saying that the state-level data was from another state (without changing the data itself), then by giving the data from the other state (while explicitly telling the model that data is from said other state), before finally removing the state-level data altogether. The result for this experiment with 5 instances per task for Direct Prompt - GPT-4o is shown in Tab. 7, while the forecasts for a single instances are shown in Fig. 22. From these, we see that the model aggressively used data which is marked as being from an other state, even though if the data was actually from said other state, the performance would be closer to not having any state-level data. This shows that the model is liable to take any information provided as being useful, even though its usefulness is marginal.

**Table 7:** *Ability of the Direct Prompt - GPT-4o model to accurately predict the unemployement level of an American county, given various covariates. These results are averaged over 5 instances.*

| Available data | RCPRS |
| --- | --- |
| Data from the correct state, accurately tagged | 0.0583 |
| Data from the correct state, inaccurately tagged | 0.0557 |
| Data from an incorrect state, accurately tagged | 0.1966 |
| No state-level data | 0.2630 |

## C.10 The effect of significant failures on the aggregate performance of models

As discussed in Sec. 5.4, in a few instances from the benchmark, some models return forecasts that miss the ground truth by a large margin, which we term significant failures (detailed in Appendix C.4). We analyse the effect of such significant failures on the results here. We use the Direct Prompt - Mixtral 8x7B model as an example here, while the same phenomenon may apply to other models. In Fig. 4, we can find that the aggregate RCRPS of Direct Prompt - Mixtral 8x7B *worsens* when it uses context. However, in Fig. 5 (left), the win rate of the model vs quantitative baselines *improves* when it uses context. These two figures show results that seem contradictory, but are in fact compatible: adding context improves the model's RCRPS for most tasks, but greatly worsens it for a minority of tasks where the model achieves significant failures.

To further illustrate this effect, we visualize the task-wise RCRPS of the DP Mixtral-8x7B-Inst model, both with and without context, in Fig. 23. With context, the model gets an RCRPS close to zero in a large number of tasks. However, there is also a long tail of tasks with high RCRPS values with context, dominating and worsening the model's aggregate RCRPS.

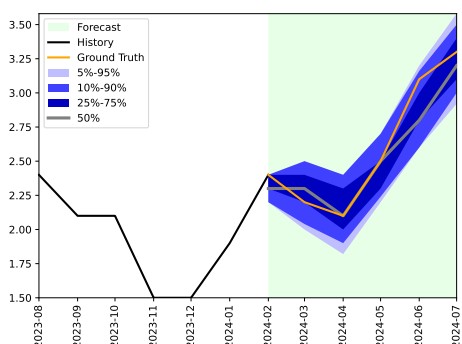

**(a)** *The task in our benchmark: the context contains the unemployment rate of the state the county is in, correctly tagged with the state name.*

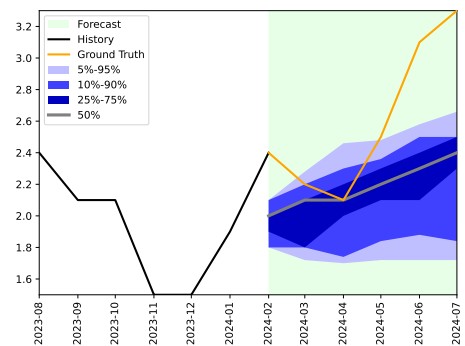

**(b)** *The context only mentions that this time series is an unemployment rate, and of which county it is. No state-level unemployement data is provided.*

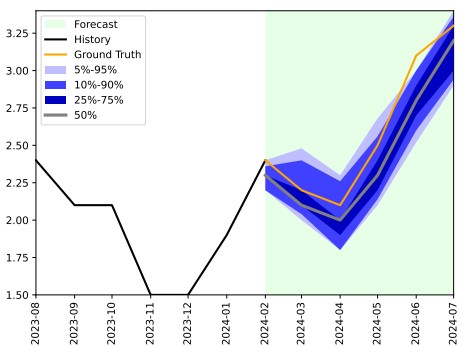

**(c)** *The state-level unemployment rate is incorrectly tagged as being from another state.*

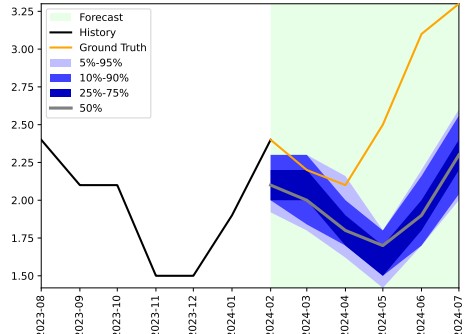

**(d)** *The context contains the unemployment rate of another state than the one the county is in, which is correctly tagged.*

**Figure 22:** *Forecasts done by Direct Prompt - GPT-4o, with varying information in the context. The task is to forecast the forecast the unemployment rate of an American county.*

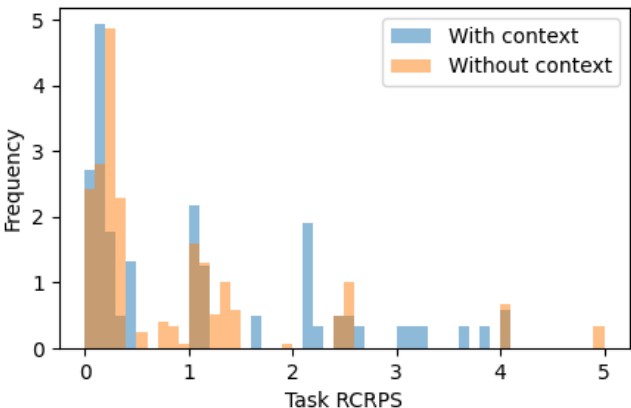

**Figure 23:** *Histogram of the RCPRS (lower is better) of the Direct Prompt Mixtral-8x7B-Inst model on each task, with and without context (with the weighting scheme detailed in Appendix A.4). With context, the model gets an RCRPS close to zero in a large number of tasks (also achieving a high win rate as seen in Fig. 5). However, there is also a long tail of tasks with high RCRPS values with context, dominating and worsening the model's aggregate RCRPS.*

## D  IMPLEMENTATION DETAILS OF MODELS

### D.1  DIRECT PROMPT

#### D.1.1  METHOD

For Direct Prompt, we propose to use a simple prompt template that we describe below, where ((**context**)) is replaced with the context of the respective task, ((**history**)) is replaced with the historical values in the given format, and ((**pred_time**)) is replaced with the prediction timesteps. The prompted model is expected to output predictions in the given template style (i.e. within the given forecast tags, in the given format) for all prediction timesteps in the prompt. Notably, unlike LLMP which consists of predicting the single next digit in a loop, Direct Prompt expects models to forecast in a single pass in a highly structured format, which requires models to understand and adhere to the template.

```
"
I have a time series forecasting task for you.

Here is some context about the task. Make sure to factor in any background knowledge,
satisfy any constraints, and respect any scenarios.
<context>
((context))
</context>

Here is a historical time series in (timestamp, value) format:
<history>
((history))
</history>

Now please predict the value at the following timestamps: ((pred_time)).

Return the forecast in (timestamp, value) format in between <forecast> and </forecast> tags.
Do not include any other information (e.g., comments) in the forecast.

Example:
<history>
(t1, v1)
(t2, v2)
(t3, v3)
</history>
<forecast>
(t4, v4)
(t5, v5)
</forecast>
"
```

We observe that models often produce samples that fail to adhere to the structure and are therefore rejected. When sampling 25 samples from the model, with Direct Prompt, we allow retrying for a maximum of $K$ times until we obtain 25 valid samples. If we do not have 25 valid samples from the model at the end of $K$ retries, we record a failure of the model and attribute the model the RCRPS upper bound of 5 for that task. In practice, we find that larger models (Llama-3.1-405B-Instruct, GPT-4o and GPT-4o) can produce 25 valid forecasts with 1 to 3 retries. However with the smaller models (such as Llama-3-70B-Instruct, Llama-3-8B-Instruct and Mixtral-8x7B-Instruct), up to 10 retries can be required to obtain 25 valid forecasts. Further, we found that without an explicit "Do not include any other information (e.g., comments) in the forecast.", models often included unwanted information along with the forecasts.

**Instruction-tuning is necessary for models to work with Direct Prompt**    Direct Prompting requires forecasts to be produced in a specific structure. To generate structured outputs, models need to be steerable (Dubey et al., 2024), a capability that is typically elicited from base models with post-training methods such as instruction tuning (Wei et al., 2021). We observe this in our evaluations as we find that several base models, including Llama-3-8B, Llama-3-70B, Mixtral-8x7B, and even

the biggest base model we tried, Llama-3.1-405B, are incapable of generating outputs adhering to the structure required for Direct Prompt, despite increasing the number of retries to as high as 50 retries. With Direct Prompt, these models often output irrelevant information, sometimes completing solely the context as a text completion task, and in other cases regurgitating forecasting datasets that they have memorized.

**Extensions of Direct Prompt** While very simple, such prompt templates can be powerful tools to understand how LLMs perform context-aided forecasting: as the prompt gives control over the structure and content of the output (particularly for instruction-tuned models), one may construct other, more involved template structures in the prompt. For instance, a prompt template could ask LLMs to explain the reasoning behind their (context-aided) forecasts, and more. We leave it to future work to understand how such prompt-based techniques can lead to more detailed evaluations and give us better insights into what the models are capable of.

### D.1.2 IMPLEMENTATION DETAILS

We used a single H100 GPU to run the Direct Prompt approach for Llama-3-8B-Instruct, and 2 H100 GPUs for Qwen-2.5-7B-Instruct, Mistral-7B-Inst, Llama-3-70B-Instruct and Mixtral-8x7B-Instruct. We queried Llama-3.1-405b-Instruct from an externally-hosted server running on 8 H100s. We use the OpenAI API to perform inference on the proprietary GPT-4o and GPT-4o-mini models. We provide the cost incurred in the inference of these models with the Direct Prompt method in Appendix C.7.

### D.1.3 EXAMPLE PROMPT

A prompt used in an example task from the benchmark is given below.

```
"
I have a time series forecasting task for you.

Here is some context about the task. Make sure to factor in any background knowledge,satisfy
    any constraints, and respect any scenarios.
<context>
Background: This is hourly traffic data.
Scenario: Suppose that there is an accident on the road and there is 40.0% of the usual
    traffic from 2024-04-24 17:00:00 for 6 hours.
</context>

Here is a historical time series in (timestamp, value) format:
<history>
(2024-04-23 00:00:00, 0.1)(2024-04-23 01:00:00, 0)(2024-04-23 02:00:00, 0)(2024-04-23
    03:00:00, 0)(2024-04-23 04:00:00, 0.1)(2024-04-23 05:00:00, 0.2)(2024-04-23 06:00:00,
    0.3)(2024-04-23 07:00:00, 0.5)(2024-04-23 08:00:00, 0.5)(2024-04-23 09:00:00, 0.4)
    (2024-04-23 10:00:00, 0.5)(2024-04-23 11:00:00, 0.5)(2024-04-23 12:00:00, 0.4)
    (2024-04-23 13:00:00, 0.6)(2024-04-23 14:00:00, 0.8)(2024-04-23 15:00:00, 1.2)
    (2024-04-23 16:00:00, 1.2)(2024-04-23 17:00:00, 1.3)(2024-04-23 18:00:00, 0.6)
    (2024-04-23 19:00:00, 0.3)(2024-04-23 20:00:00, 0.3)(2024-04-23 21:00:00, 0.3)
    (2024-04-23 22:00:00, 0.1)(2024-04-23 23:00:00, 0.1)(2024-04-24 00:00:00, 0.1)
    (2024-04-24 01:00:00, 0)(2024-04-24 02:00:00, 0)(2024-04-24 03:00:00, 0.1)(2024-04-24
    04:00:00, 0.1)(2024-04-24 05:00:00, 0.2)(2024-04-24 06:00:00, 0.3)(2024-04-24 07:00:00,
    0.5)(2024-04-24 08:00:00, 0.6)(2024-04-24 09:00:00, 0.5)(2024-04-24 10:00:00, 0.4)
    (2024-04-24 11:00:00, 0.5)(2024-04-24 12:00:00, 0.6)
</history>

Now please predict the value at the following timestamps: ['2024-04-24 13:00:00' '2024-04-24
    14:00:00' '2024-04-24 15:00:00' '2024-04-24 16:00:00' '2024-04-24 17:00:00'
    '2024-04-24 18:00:00' '2024-04-24 19:00:00' '2024-04-24 20:00:00' '2024-04-24 21:00:00'
    '2024-04-24 22:00:00' '2024-04-24 23:00:00' '2024-04-25 00:00:00' '2024-04-25
    01:00:00' '2024-04-25 02:00:00' '2024-04-25 03:00:00' '2024-04-25 04:00:00' '2024-04-25
    05:00:00' '2024-04-25 06:00:00' '2024-04-25 07:00:00' '2024-04-25 08:00:00'
    '2024-04-25 09:00:00' '2024-04-25 10:00:00' '2024-04-25 11:00:00' '2024-04-25
    12:00:00'].

Return the forecast in (timestamp, value) format in between <forecast> and </forecast> tags.
    Do not include any other information (e.g., comments) in the forecast.

Example:
<history>
(t1, v1)
(t2, v2)
(t3, v3)
</history>
<forecast>
(t4, v4)
(t5, v5)
</forecast>
"
```

## D.2 LLMP

In this section we outline LLM-processes (LLMP; Requeima et al. (2024)), one of the prompt-based
baselines evaluated in Sec. 5.3. Prompts are constructed by first providing textual information
followed by the numerical history. The context may include background knowledge, a scenario
description and task constraints, replaced by ((**background**)), ((**scenario**)) and ((**constraints**)),
respectively, in the prompt template below. The numerical history (((**history**))) is provided by
converting the numerical data to text where values are separated by commas (,) and tuples by newline
characters (\n). The LLM then outputs the continuation of the string prompt, forecasing the the value

for the next time index (((**next index**))). This forecast and the next time index is appended to the prompt allowing the LLM to autoregressively complete the entire forecast. Numerical samples are rejected if they do not adhere to a decimal representation format. See Requeima et al. (2024)) for full details.

The following is the prompt template used to construct prompts for the LLMP baseline:

```
"
Forecast the future values of this time series, while considering the following background
    knowledge, scenario, and constraints.

Background knowledge:
((background))

Scenario:
((scenario))

Constraints:
((constraints))

((history))
((next index))
"
```

A prompt used in an example task from the benchmark is given below:

```
"
Forecast the future values of this time series, while considering the following background
    knowledge, scenario, and constraints.

Background knowledge:
This is hourly traffic data.

Scenario:
Suppose that there is an accident on the road and there is 40.0% of the usual traffic from
    2024-04-24 17:00:00 for 6 hours.

Constraints:

2024-04-23 00:00:00,0.1\n2024-04-23 01:00:00,0\n2024-04-23 02:00:00,0\n2024-04-23 03:00:00,0
    \n2024-04-23 04:00:00,0.1\n2024-04-23 05:00:00,0.2\n2024-04-23 06:00:00,0.3\n2024-04-23
     07:00:00,0.5\n2024-04-23 08:00:00,0.5\n2024-04-23 09:00:00,0.4\n2024-04-23
    10:00:00,0.5\n2024-04-23 11:00:00,0.5\n2024-04-23 12:00:00,0.4\n2024-04-23 13:00:00,0.6
    \n2024-04-23 14:00:00,0.8\n2024-04-23 15:00:00,1.2\n2024-04-23 16:00:00,1.2\n2024-04-23
     17:00:00,1.3\n2024-04-23 18:00:00,0.6\n2024-04-23 19:00:00,0.3\n2024-04-23
    20:00:00,0.3\n2024-04-23 21:00:00,0.3\n2024-04-23 22:00:00,0.1\n2024-04-23 23:00:00,0.1
    \n2024-04-24 00:00:00,0.1\n2024-04-24 01:00:00,0\n2024-04-24 02:00:00,0\n2024-04-24
    03:00:00,0.1\n2024-04-24 04:00:00,0.1\n2024-04-24 05:00:00,0.2\n2024-04-24 06:00:00,0.3
    \n2024-04-24 07:00:00,0.5\n2024-04-24 08:00:00,0.6\n2024-04-24 09:00:00,0.5\n2024-04-24
     10:00:00,0.4\n2024-04-24 11:00:00,0.5\n2024-04-24 12:00:00,0.6\n2024-04-24 13:00:00,
"
```

We used a single H100 GPU to run the LLMP approach for the following models: Llama-3-8B, and Llama-3-8B-Instruct. We used 2 H100 GPUs for the Qwen-2.5 family of models, Mixtral-8x7B, and Mixtral-8x7B-Instruct, and used used 8 H100 GPUs for the following models: Llama-3-70B, and Llama-3-70B-Instruct.

### D.3    UNITIME AND TIME-LLM

For multimodal models, we jointly train UniTime (Liu et al., 2024c) on its ensemble of datasets: ETTm1, ETTm2, ETTh1, ETTh2, Electricity, Weather, Exchange and Illness.

We also evaluate Time-LLM (Jin et al., 2024), another multimodal model built on top of the Llama architecture. We train Time-LLM on ETTh1 according to the authors' suggested specifications, and we compare the performance of both models with and without context.

**UniTime:** We train UniTime (Liu et al., 2024c) using a single seed on one AMD Instinct MI200 GPU for approximately 14 hours. It features a lightweight transformer with maximum context length of 210 and a pre-trained GPT2 language model as backbone, of which only the first half of the transformer layers are used. The time series baseline employs non-overlapping patch embeddings generated with a kernel size and stride of 16, and a maximum input sequence length of 96. When the total tokenized length exceeds the architecture's capacity, we truncate the context.

Unlike Time-LLM, UniTime is jointly trained on all datasets simultaneously. Batches were generated by first choosing a dataset uniformly at random then returning a batch from the associated data loader. To account for domain convergence speed imbalance, a mask rate of 0.5 is used and the training batch size is varied according to the dataset (details in the data config directory of the UniTime GitHub repository). Training was conducted for 10 epochs of the mixed dataset, with cosine decay from an initial learning rate of 1e-4 to a minimum of 1e-6 over a maximum period of 20 epochs. The results of our training on the original datasets are given in Tab. 8.

Finally, in order to accelerate training, we added BF16 automatic mixed precision training and gradient accumulation to the original training procedure.

**Time-LLM:** We train Time-LLM (Jin et al., 2024) on the ETTh1 dataset (Zhou et al., 2021) with a prediction length of 96. We train using a single seed on four AMD Instinct MI200 GPUs, with an average training time per run of approximately 13 hours. Training was conducted using a batch size of 8 per device and 4 gradient accumulation steps, along with a 1Cycle learning rate schedule with a maximum learning rate of 1e-3. In addition, runs were accelerated using DeepSpeed Stage 2 and BF16 automatic mixed precision.

Training was conducted over a maximum of 50 epochs with early stopping, and a time-based split of 70% for training, 10% for validation, and 20% for testing, where the most recent windows were reserved for the test set. All runs were trained with an input sequence length of 512, with overlapping patch embeddings generated with a kernel size of 16 and a stride of 8. The results on the ETTh1 test set are given in Tab. 9.

When evaluating on CiK tasks which do not conform to Time-LLM's requirements, we make the following modifications to the method:

- For short history tasks where the history length $|\mathbf{X_H}|$ is less than 5, we change the `topk` operator's $k$ value from 5 to $|\mathbf{X_H}|$ in the `calculate_lags()` function.

- For tasks where the length of the prediction window $|\mathbf{X_F}|$ exceeds the trained projection head's output dimension (in our case, 96), we repeat the last predicted value $|\mathbf{X_F}| - 96$ times. This occurs for very few tasks (3 tasks) with prediction windows of 97 or 98 steps depending on the sampled instance, which we assume leads to a negligible impact on evaluated results.

**Table 8:** *Evaluation results for UniTime on their test splits. Results are comparable to the original paper, although MSE on Illness is approximately 20% higher for prediction lengths 36,48,60.*

| Dataset | Mean Squared Error (MSE) | | | |
|---|---|---|---|---|
| Prediction Length | 96 | 192 | 336 | 720 |
| ETTh1 | 0.395 | 0.435 | 0.469 | 0.468 |
| ETTh2 | 0.291 | 0.368 | 0.413 | 0.422 |
| ETTm1 | 0.336 | 0.377 | 0.409 | 0.465 |
| ETTm2 | 0.181 | 0.248 | 0.315 | 0.417 |
| Exchange | 0.090 | 0.180 | 0.322 | 0.862 |
| Weather | 0.179 | 0.224 | 0.278 | 0.354 |
| Electricity | 0.198 | 0.202 | 0.217 | 0.257 |
| | 24 | 36 | 48 | 60 |
| Illness | 2.284 | 2.515 | 2.572 | 2.455 |

**Table 9:** *ETTh1 test set results for Time-LLM trained on ETTh1.*

| Time-LLM | MSE | MAE |
|---|---|---|
| ETTh1-pl96 | 0.3846123 | 0.4149854 |

**Why Do Time-LLM and UniTime Not Benefit (More) From Context?** Looking at table Appendix C.3, we see that context actually harms the performance of Time-LLM's forecasts. Two possible reasons for this are: 1) Time-LLM's adaptation procedure is unlikely to retain the backbone LLM's language-processing capabilities, and 2) Time-LLM's single-dataset training procedure is unlikely to generalize to unseen time series patterns. Part of Time-LLM's model adaptation involves training linear layers at the input and output of the language model. Although the backbone LLM remains frozen, these linear layers must be trained, and Time-LLM opts for a highly structured prompting format which involves domain knowledge, task instructions and input statistics. Since the training data for the linear layers consists of output representations based on these highly structured prompts, it is not evident that the resulting architecture will generalize to more diverse contextual descriptions such as those found in CiK. Furthermore, although we have not conducted a formal analysis of the diversity of the ETTh1 dataset, it is not a priori obvious that such a dataset would have a sufficient diversity of patterns to train a time series foundation model.

Interestingly, UniTime's performance does benefit from context for some tasks (see Fig. 24). However, the aggregate RCRPS and rank of UniTime with respect to other models indicate that it still struggles to produce forecasts competitive with even quantitative forecasting methods.

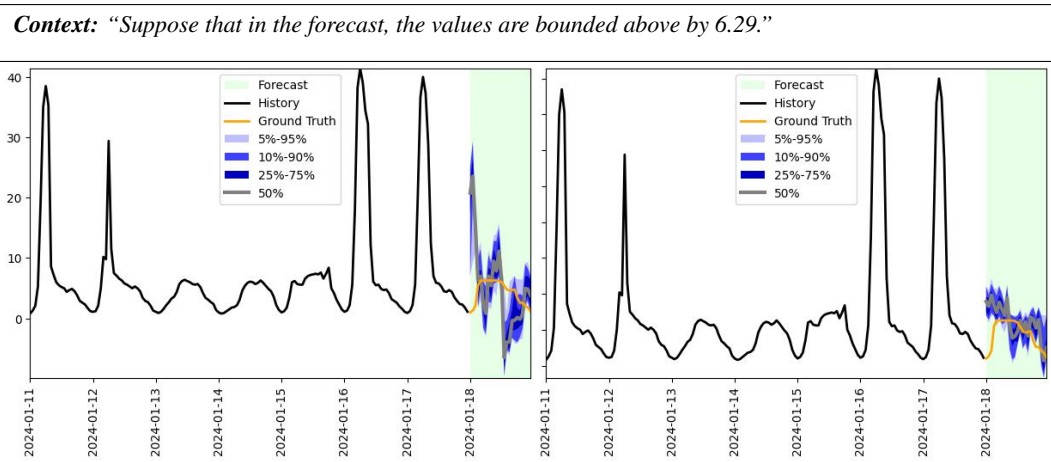

**Figure 24:** *A comparison of forecasts from UniTime without context (left) and with context (right). On average across 5 instances, UniTime's RCRPS is 64% better with context than without on the "Bounded Prediction Constraint Based On Prediction Quantiles" task.*

### D.4 LAG-LLAMA

We use the publicly available implementation of Lag-Llama (Rasul et al., 2023) located at `https://github.com/time-series-foundation-models/`, and its associated pre-trained weights. The model inference was done on a single H100 GPU.

### D.5 CHRONOS

We use the publicly available implementation of Chronos (Ansari et al., 2024) located at `https://github.com/amazon-science/chronos-forecasting`. We evaluated (see Appendix C.3) our tasks on all 5 available models: chronos-tiny, chronos-mini, chronos-small, chronos-base and chronos-large, and reported the results of the best performing model, chronos-large in Tab. 1. The model inference was done on a single H100 GPU.

### D.6 MOIRAI

We use the publicly available implementation of Moirai (Woo et al., 2024) located at `https://github.com/SalesforceAIResearch/uni2ts`. We evaluated (see Appendix C.3) our tasks on the 3 following models: moirai-1.0-R-small (located at `https://huggingface.co/Salesforce/moirai-1.0-R-small`), moirai-1.0-R-base (located at `https://huggingface.co/Salesforce/moirai-1.0-R-base`) and moirai-1.0-R-large (located at `https://huggingface.co/Salesforce/moirai-1.0-R-large`) and reported the results of the best performing model, moirai-1.0-R-large in Tab. 1. The model inference was done on a single H100 GPU.

### D.7 TIMEGEN

We access TimeGEN-1, an optimization of the TimeGPT model (Garza et al., 2023), using the API made available through the `nixtla` Python package. Unlike all other baselines, we only generate point forecasts with TimeGEN due to its probabilistic mode requiring much longer historical data than is available in instances evaluated in the benchmark. This is the reason the RCPRS values for TimeGEN have zero standard error.

### D.8 EXPONENTIAL SMOOTHING

We used the Exponential Smoothing implementation from the `statsmodels` Python package, namely the `statsmodels.tsa.holtwinters.ExponentialSmoothing` class. Both trend and seasonal components of the models are set to be additive. The seasonal period length is either set manually for tasks where the simple guess using the time series frequency is incorrect. If there is not at least two full seasonal periods in the history window of the time series, we disable the seasonal component of the model. Since some of the benchmark tasks can have as few as 3 time steps in the history window, we also disable the trend component if we have less than 5 time steps in said window.

### D.9 ETS AND ARIMA

We used the implementations of ETS and ARIMA from the `forecast` R package, using `rpy2` for compatibility with Python. For ETS, we use the `ets` method, which we call with automatic error, trend, and seasonality components. In the rare cases where the ETS forecast contains NaN values, we manually switch off the trend component and rerun the forecast. The ARIMA results are computed using the `auto.arima` method. If the ARIMA fits fail, we rerun it with restricted parameter and disabled seasonality.

## E DETAILS OF THE PROPOSED METRIC

The CiK benchmark is designed to determine whether models can improve their probabilistic forecasts by leveraging associated textual information (see Sec. 2). To support this goal, the evaluation metric:

1. should be a **proper scoring rule**, such that a model who perfectly knows what the correct forecast is should have no reason to favor another prediction;

2. must be **easy to compute** using a finite sample from the forecast distribution, since many models do not provide a functional form for their forecasts.

To account for the importance of leveraging relevant context, the metric should also:

1. **penalize obviously impossible forecasts**, i.e. that can be inferred as implausible from the contextual information;

2. **take a similar range of values across different tasks**, to prevent some tasks to dominate the score as we average the results across tasks;

3. **prioritize forecast quality for timesteps with relevant context**, even if these timesteps are a small portion of the forecast horizon.

To satisfy the first two properties, we start with the Continuous Ranked Probability Score (CRPS) (Gneiting & Raftery, 2007), a reliable strictly proper scoring rule for univariate probability distribution, and take its mean over all time steps. To compute the CRPS from a finite number of samples, we use the estimator based on its probability weighted moment form (Taillardat et al., 2016), since it is unbiased (Zamo & Naveau, 2018). See Appendix E.3 for more details about this estimator.

Many of our tasks are built to include information about a hard constraint on $\mathbf{X}_F$ in their $\mathcal{C}$, which can be written as $v_{\mathcal{C}}(\mathbf{x}_F) = 0$. If we were only interested to measure by how much a forecast breaks the constraint, we could take inspiration from the threshold-weighted CRPS (Gneiting & Ranjan, 2011) by using $v_{\mathcal{C}}$ as its chaining function (Allen et al., 2023):

$$\text{twCRPS}_{v_{\mathcal{C}}}(\widetilde{\mathbf{X}}_F, \mathbf{x}_F) \equiv \text{CRPS}\left(v_{\mathcal{C}}(\widetilde{\mathbf{X}}_F), v_{\mathcal{C}}(\mathbf{x}_F)\right), \tag{1}$$

where $\widetilde{\mathbf{X}}_F$ is the forecast of $\mathbf{X}_F$ to be evaluated. Since, by construction, the ground-truth $\mathbf{x}_F$ always satisfy the constraints, we have $v_{\mathcal{C}}(\mathbf{x}_F) = 0$. But since we do not care only about whether forecasts break constraints, we sum both the original CRPS and this twCRPS, but we weight the later by a factor of $\beta = 10$, to denote the additional interest we show to these errors. See Appendix E.4 for the various $v_{\mathcal{C}}$ used in the benchmark.

One common approach to normalize the CRPS to get similar ranges for multiple problems is to divide it by the mean absolute value of the target ground-truth of the forecasted series (Alexandrov et al., 2020). This has two issues: the metric is no longer proper, and it leads to much larger values for series close to zero than those far from it. To solve the first issue, we take advantage that we can generate many more instances from each of our tasks, by computing a normalization factor $\alpha$ from 25 instances not included in the benchmark. The details of this calculations are in Appendix E.1.

Many tasks in our benchmark contains contextual information which is highly relevant for a small fraction of the time steps in the forecasting window, while being only marginally relevant for the majority of the time steps. If we were to weight these two categories equally, then the score for a model which ignores the context would be hard to distinguish from the score of one who does not. We correct this issue by identifying the subset of time steps with relevant information, which we call the Region of Interest (RoI). We then weight the CRPS to give half weight to the RoI time steps and half weight to the non-RoI time steps. Therefore, we obtain our metric, which we call the Region-of-Interest CRPS (RCRPS):

$$\text{RCRPS}(\widetilde{\mathbf{X}}_F, \mathbf{x}_F) := \begin{cases} \alpha \cdot \left[\frac{1}{2|\mathcal{I}|} \cdot \sum\limits_{i \in \mathcal{I}} \text{CRPS}\left(\widetilde{X}_i, x_i\right) + \frac{1}{2|\neg\mathcal{I}|} \cdot \sum\limits_{i \in \neg\mathcal{I}} \text{CRPS}\left(\widetilde{X}_i, x_i\right) + \beta \cdot \text{CRPS}\left(v_{\mathbf{C}}(\widetilde{\mathbf{X}}_F), 0\right)\right] & \text{if } |\mathcal{I}| > 0 \\ \alpha \cdot \left[\frac{1}{|\neg\mathcal{I}|} \cdot \sum\limits_{i \in \neg\mathcal{I}} \text{CRPS}\left(\widetilde{X}_i, x_i\right) + \beta \cdot \text{CRPS}\left(v_{\mathbf{C}}(\widetilde{\mathbf{X}}_F), 0\right)\right], & \text{if } |\mathcal{I}| = 0 \end{cases}$$

where $\mathcal{I}$ is the set of time steps in the RoI, $\neg\mathcal{I}$ is the set of time steps in the forecast but not in the RoI, $\alpha$ is the aforementioned scaling factor, and we drop the factor of two and the first sum for tasks where there is no meaningful RoI.

### E.1 SCALING FOR CROSS-TASK AGGREGATION

The rationale behind scaling the RCPRS is to allow us to average its value from diverse tasks without the average being dominated by the forecast quality for tasks with time series with large values. An alternative argument is: all other conditions being equal, a forecaster that is wrong by 10 in its forecast for a time series which goes from 25 to 30 is worse than one that is wrong by 100 in its forecast for a time series which goes from 2500 to 3000. Furthermore, we have multiple tasks for which some instances have constant $\mathbf{x}_F$ or nearly so, often with values close to zero. Due to these tasks, we cannot simply use a scaling which only depends on said instances $\mathbf{x}_F$. Instead, we take advantage of our benchmark ability to generate a very large number of instances for each tasks by using $M = 25$ instances not included in our benchmark. Given the ground-truth future values $\mathbf{x}_F^m$ for these instance, the scaling factor $\beta$ for an individual task is as follow:

$$\alpha = \left[\frac{\sum_m \left(\max_i x_i^m - \min_i x_i^m\right)}{M}\right]^{-1}. \tag{2}$$

**Properness**   In an ideal scenario, all instances of a tasks would be fully independent. In that case then Eq. (2) would not contain any information about the target time series in the benchmark instances, making the RCPRS a proper scoring rule. However, due to possible overlaps in the time windows used when creating the instances and to auto-correlations, we cannot guarantee independence between instances, and thus we cannot guarantee that the RCPRS is actually a proper scoring rule. Note that this deviation from a proper scoring rule is minor, and has a much smaller effect than the one due to the common approach of normalizing the CRPS using the Mean Absolute Value of the ground-truth.

## E.2   CRPS AND TWCRPS

Given a univariate forecast $\widetilde{X}$ and a ground-truth realization $x$, the Continuous Ranked Probability Score (CRPS) can be defined in its integral as follow:

$$\mathrm{CRPS}(\widetilde{X}, x) = \int_{-\infty}^{\infty} dy \left[ \Phi_{\widetilde{X}}(y) - \mathbb{1}(y \geq x) \right]^2, \tag{3}$$

where $\Phi_{\widetilde{X}}(y)$ is the Cumulative Distribution Function of $\widetilde{X}$, and $\mathbb{1}$ is the indicator function.

There are multiple ways to compute the CRPS, but a particularly interesting one which showcases its link to the Mean Absolute Error is the energy form of the CRPS:

$$\mathrm{CRPS}(\widetilde{X}, x) = \mathbb{E}_{X \sim \widetilde{X}} |X - x| - \frac{1}{2} \mathbb{E}_{X, X' \sim \widetilde{X}} |X - X'|. \tag{4}$$

We get the threshold-weighted CRPS (twCRPS) from Eq. (4) by adding a weighting function $w(x)$ to it:

$$\mathrm{twCRPS}(\widetilde{X}, x) = \int_{-\infty}^{\infty} dy\, w(y) \left[ \Phi_{\widetilde{X}}(y) - \mathbb{1}(y \geq x) \right]^2. \tag{5}$$

To get the energy form of the twCRPS, we must compute the chaining function $v(x)$ from $w(x)$:

$$v(x) - v(x') = \int_{[x, x')} dy\, w(y). \tag{6}$$

Using $v(x)$, we can write the twCRPS as:

$$\mathrm{twCRPS}(\widetilde{X}, x) = \mathbb{E}_{X \sim \widetilde{X}} |v(X) - v(x)| - \frac{1}{2} \mathbb{E}_{X, X' \sim \widetilde{X}} |v(X) - v(X')|. \tag{7}$$

Eq. (7) can readily be generalized to a multivariate forecast, by using any $\mathbb{R}^d \to \mathbb{R}$ chaining function.

## E.3   ESTIMATING THE CRPS USING SAMPLES

Computing the CRPS using Eq. (3) or Eq. (4) directly would be extremely hard for most of the baselines included in our experiments. Instead, it is more computationally convenient to use an estimator of the CRPS which uses a finite number of samples $x_1, ..., x_M$ from the forecasting distribution. An unbiased estimator of the CRPS created from Eq. (4) is:

$$\mathrm{CRPS}(\widetilde{X}, x) \approx \frac{1}{M} \sum_{n=1}^{M} |x_n - x| - \frac{1}{2M(M-1)} \sum_{n=1}^{M} \sum_{n'=1}^{M} |x_n - x_{n'}|. \tag{8}$$

However, this estimator is relatively costly, having a $O(M^2)$ time complexity.

A faster estimator which gives the same result as Eq. (8) (up to numerical accuracy) is the one based on the probability weighted moment form of the CRPS (Taillardat et al., 2016; Zamo & Naveau, 2018):

$$\mathrm{CRPS}(\widetilde{X}, x) \approx \frac{1}{M} \sum_{n=1}^{M} |x_n - x| + \frac{1}{M} \sum_{n=1}^{M} x_n - \frac{2}{M(M-1)} \sum_{n=1}^{M} (n-1)x_n, \tag{9}$$

where the $x_n$ have been sorted in ascending order. We used Eq. (9) in our metric, since it is as accurate as Eq. (8), while only having a $O(M \log M)$ time complexity.

### E.4 Constraint-violation functions

In selecting constraint-violation functions $v_{\mathcal{C}}$ for our various tasks, we have the following requirements: it should be invariant to the number of timesteps in the forecasting window and it should be multiplied by $\alpha$ if all numerical data in a task is transformed using $x \to \alpha x + \beta$. Here are the $v_{\mathcal{C}}$ we use in some of our benchmark tasks:

- *Constant upper-bound constraint $x_i \leq \tau^+$:*

$$v_{\mathcal{C}}(\mathbf{x}_F) = \frac{1}{T-t} \sum_{i=t+1}^{T} \max(0, x_i - \tau^+),$$

- *Constant lower-bound constraint $x_i \geq \tau^-$:*

$$v_{\mathcal{C}}(\mathbf{x}_F) = \frac{1}{T-t} \sum_{i=t+1}^{T} \max(0, \tau^- - x_i),$$

- *Constant lower-bound and upper-bound constraints $\tau^- \leq x_i \leq \tau^+$:*

$$v_{\mathcal{C}}(\mathbf{x}_F) = \frac{1}{T-t} \sum_{i=t+1}^{T} \max(0, \tau^- - x_i) + \max(0, x_i - \tau^+),$$

- and *Variable upper-bound constraints, on a subset of time steps $x_i \leq \tau_i^+ \ \forall i \in C$:*

$$v_{\mathcal{C}}(\mathbf{x}_F) = \frac{1}{|C|} \sum_{i \in C} \max(0, x_i - \tau_i^+).$$

### E.5 Covariance of two CRPS estimators

One approach to compute standard error on the RCRPS is to compute the empirical standard deviation based on the 5 instances we use for each task. However, such a method would overestimate the standard error, since it would consider both the variance coming from the selection of instances of a given task, and the variance coming from the models sampling processes. Since all models are tested using the exact same instances, the variance coming from their selection is not relevant, and thus we need a way to ignore it.

To do so, we take advantage that the RCRPS is a weighted sum of multiple CRPS estimates. Since those estimates are not independent from one another, we can compute an estimate of the variance of the RCPRS under the sampling process by computing an estimate of the covariance matrix between the various CRPS estimates, followed by the appropriate weighted sum.

Let says we want to compute the covariance between the CRPS for variable $i$ and the CRPS for variable $j$, using $M$ independent and identically distributed samples from the joint distribution of $\widetilde{X}_i$ and $\widetilde{X}_j$.

$$\mathrm{Cov}\left(\mathrm{CRPS}\left(\widetilde{X}_i, x_i\right), \mathrm{CRPS}\left(\widetilde{X}_j, x_j\right)\right) =$$

$$\mathrm{Cov}\left(\frac{1}{M} \sum_n |\widetilde{X}_{i,n} - x_i| - \frac{1}{2M(M-1)} \sum_{n \neq n'} |\widetilde{X}_{i,n} - \widetilde{X}_{i,n'}|,\right.$$

$$\left.\frac{1}{M} \sum_n |\widetilde{X}_{j,n} - x_j| - \frac{1}{2M(M-1)} \sum_{n \neq n'} |\widetilde{X}_{j,n} - \widetilde{X}_{j,n'}|\right),$$

where the sums are over the various samples $n$ and $x_i$ and $x_j$ and the ground-truth values.

After some tedious algebraic manipulations, we obtain the final formula for the covariance of two CRPS estimates:

$$\mathrm{Cov}\left(\mathrm{CRPS}\left(\widetilde{X}_i, x_i\right), \mathrm{CRPS}\left(\widetilde{X}_j, x_j\right)\right) =$$

$$-\frac{1}{M} \mathop{\mathbf{E}}_{\widetilde{X}_i} |\widetilde{X}_i - x_i| \mathop{\mathbf{E}}_{\widetilde{X}'_j} |\widetilde{X}'_j - x_j|$$

$$+\frac{1}{M} \mathop{\mathbf{E}}_{\widetilde{X}_i} |\widetilde{X}_i - x_i| \mathop{\mathbf{E}}_{\widetilde{X}'_j} \mathop{\mathbf{E}}_{\widetilde{X}''_j} |\widetilde{X}'_j - \widetilde{X}''_j|$$

$$+\frac{1}{M} \mathop{\mathbf{E}}_{\widetilde{X}_i} \mathop{\mathbf{E}}_{\widetilde{X}'_i} |\widetilde{X}_i - \widetilde{X}'_i| \mathop{\mathbf{E}}_{\widetilde{X}''_j} |\widetilde{X}''_j - x_j|$$

$$-\frac{2M-3}{2M(M-1)} \mathop{\mathbf{E}}_{\widetilde{X}_i} \mathop{\mathbf{E}}_{\widetilde{X}'_i} |\widetilde{X}_i - \widetilde{X}'_i| \mathop{\mathbf{E}}_{\widetilde{X}''_j} \mathop{\mathbf{E}}_{\widetilde{X}'''_j} |\widetilde{X}''_j - \widetilde{X}'''_j|$$

$$+\frac{1}{M} \mathop{\mathbf{E}}_{(\widetilde{X}_i, \widetilde{X}_j)} |\widetilde{X}_i - x_i| \cdot |\widetilde{X}_j - x_j|$$

$$-\frac{1}{M} \mathop{\mathbf{E}}_{(\widetilde{X}_i, \widetilde{X}_j)} \mathop{\mathbf{E}}_{\widetilde{X}'_i} |\widetilde{X}_i - \widetilde{X}'_i| \cdot |\widetilde{X}_j - x_j|$$

$$-\frac{1}{M} \mathop{\mathbf{E}}_{(\widetilde{X}_i, \widetilde{X}_j)} \mathop{\mathbf{E}}_{\widetilde{X}'_j} |\widetilde{X}_i - x_i| \cdot |\widetilde{X}_j - \widetilde{X}'_j|$$

$$+\frac{1}{2M(M-1)} \mathop{\mathbf{E}}_{(\widetilde{X}_i, \widetilde{X}_j)} \mathop{\mathbf{E}}_{(\widetilde{X}'_i, \widetilde{X}'_j)} |\widetilde{X}_i - \widetilde{X}'_i| \cdot |\widetilde{X}_j - \widetilde{X}'_j|$$

$$+\frac{M-1}{M(M-1)} \mathop{\mathbf{E}}_{(\widetilde{X}_i, \widetilde{X}_j)} \mathop{\mathbf{E}}_{\widetilde{X}'_i} \mathop{\mathbf{E}}_{\widetilde{X}''_j} |\widetilde{X}_i - \widetilde{X}'_i| \cdot |\widetilde{X}_j - \widetilde{X}''_j|,$$

where variables with the same number of apostrophes ($'$) are drawn together and those with different number of apostrophes are independent variables.

To get an estimate of covariance using our $M$ samples, we can estimate each of these terms using their respective unbiased estimators. Once we have compute an estimate of the variance for a single task instance, the overall variance for a full task is computed using the formula for the variance of the average of multiple independent variables. One slight disadvantage of using this method, is that it offers now guarantee that the RCPRS variance estimate will be non-negative, so in the rare cases where the estimate for the variance of a full task is negative, we clip it to 0.

