# OpenReview forum: "Context is Key: A Benchmark for Forecasting with Essential Textual Information"
_ICLR.cc/2025/Conference — Submitted to ICLR 2025_

### Official Review · Reviewer_RToE · 2024-10-24

**Soundness:** 3
**Presentation:** 3
**Contribution:** 2
**Rating:** 5
**Confidence:** 4

**Summary:**

The paper introduces "Context is Key" (CiK), a benchmark designed to evaluate the ability of forecasting models to integrate numerical data with essential textual context (time-invariants, history, covariates, future and causals...). Recognizing that accurate forecasting often requires more than just numerical inputs, the authors aim to bridge this gap by creating a collection of tasks that necessitate the use of both modalities. Some key contributions:
1. a relatively complete benchmark named cik
2. analysis of different models
3. propose direct prompt, which is a simple strategy to prompt LLM to do time-serise prediction

**Strengths:**

I would say this is a 'huge' work, congratulations!
The following are some points I agree:
1. the benchmark is relatively complete and has a potential to have impact, which include textual context (time-invariants, history, covariates, future and causals).This work may lead to complex or graph modelling with these information.
2. the proposed strategy to evaluate is simple but useful.
3. the paper contains some use case and discussion of existed models

**Weaknesses:**

1. lack of discussion of noise in texts. The texts are complete but the quality filter is not well designed. In general, we need more well-designed texts which is really useful
2. what is the importance of texts? are they hidden in time-series itself? For me, time-series is the sampling result of a complex system, and even you have already done a lot and try to give more complete one, but the system is hard to define. As a result, the time-series itself may contain more information than the texts.
3. besides model evaluation, the benchmark is hard to use as in real world, people may prefer to use simple model and may lack of texts.
4. The cooperation between texts and numbers is not well-designed. Generally, LLMs including GPT are not good at dealing with numbers, and they are not sensitive to the symbols in numbers.

**Questions:**

1. can you just open-source all the related-materials and I think it's better to let all people to judge if it has enough value and easy to use.
2. how do you choose the best-quality text and how would you access this quality?
3. Is there any better way to cooperate the numbers in prompt to evaluate the effect of the benchmark?
5. text brings more calculation, but the effect of the texts is not well discussed. How to balance and choose the most effective one?

---

> ### Author Response · Authors · 2024-11-19
>
> Thank you very much for your thoughtful review. We are delighted that you consider our work to be 'huge'. We hope that the responses below will alleviate some of your concerns, and that you will consider raising your score.
>
>
> ---
>
> > W1: lack of discussion of noise in texts. The texts are complete but the quality filter is not well designed. In general, we need more well-designed texts which is really useful
>
> We would like to point out to the reviewer that all tasks in the benchmark were manually designed, from scratch, by the authors of this work (without resorting to external annotators or LLMs) , to ensure quality. Note that this is a key novelty of our work. Further, all tasks in the benchmark were filtered for quality through a rigorous peer review process  by a committee with significant expertise in time series forecasting (co-authors of this work).
>
> Additionally, please see point W1 of our response to reviewer kJ9d, where we describe a newly-performed LLM-based analysis of the relevance of the context for each task.
>
> We detail the full task creation process in the benchmark below, for your reference:
>
> First, we identified high-quality sources of public time series data from various application domains (see Section 3.1). Second, we established the categorization for sources of context (Section 3.2) and capabilities (Section 3.3) as a framework to guide the creation of new tasks and ensure their diversity. Third, team members separately created the tasks, each time selecting a data source, implementing a time series window selection strategy (e.g., short or long history), brainstorming about realistic sources of context and capabilities based on the forecasting task, writing code to generate such context by instantiating a template, and finally, if required, writing code to modify the time series data to reflect the context (e.g., some future spikes). Then, the tasks were peer-reviewed by a committee with significant expertise in time series forecasting (co-authors of this work). The creator of each task was not allowed to participate in its review. The review ensured that the text was of a high quality, that it without doubt enabled a better forecast, and that the context source and capability tags were well-assigned. If a task was deemed of not high enough quality, it was either returned for revisions, or rejected.
>
> The code for all tasks and experiments is available here: https://anonymous.4open.science/r/context-is-key-forecasting-E391/. An example task can be found here: https://anonymous.4open.science/r/context-is-key-forecasting-E391/cik_benchmark/tasks/montreal_fire/short_history.py, where the time series window selection occurs from L94-112 and the context generation occurs from L114-158.
>
> We will add the same discussion to our paper in the next revision.
>
>
> ---
>
> > W2: what is the importance of texts? are they hidden in time-series itself? For me, time-series is the sampling result of a complex system, and even you have already done a lot and try to give more complete one, but the system is hard to define. As a result, the time-series itself may contain more information than the texts.
>
> The benchmark is designed such that the provided numerical information alone is insufficient to forecast properly. Thereby, additional information in the form of text is required to solve the tasks, and models are expected to leverage both the history and the additional text information when forecasting. Note that this additional information is **not derived from the time series itself**, but rather from auxiliary information sources (in the case of intemporal context), or information on other variables (for covariate context), scenarios that condition the future states of the system on (for future context), statistics about the time series that are beyond the available short history (in case of historical context), and causal information about provided covariates  (in case of causal context). Thereby, the information conveyed in the text is not hidden in the time series, and cannot be derived from the time series.
>
> ---
>
> > W3: besides model evaluation, the benchmark is hard to use as in real world, people may prefer to use simple model and may lack of texts.
>
> The focus of our benchmark is context-aided forecasting, a very new problem setting. We agree with the reviewer that our benchmark can be seen as complementary to all other benchmarks such as that of Godahewa et al. (2024) and Qiu et al. (2024), which all focus on the canonical purely-numerical forecasting setting.
>
> **References**:
>
> 1. Rakshitha Godahewa, Christoph Bergmeir, Geoffrey I Webb, Rob J Hyndman, and Pablo Montero Manso. “Monash time series forecasting archive.” arXiv preprint arXiv:2105.06643, 2021.
>
> 2. Xiangfei Qiu, Jilin Hu, Lekui Zhou, Xingjian Wu, Junyang Du, Buang Zhang, Chenjuan Guo et al. "Tfb: Towards comprehensive and fair benchmarking of time series forecasting methods." arXiv preprint arXiv:2403.20150, 2024.

---

> ### Author Response · Authors · 2024-11-19
>
> ---
>
> > W4: The cooperation between texts and numbers is well-designed. Generally, LLMs including GPT are not good at dealing with numbers, and they are not sensitive to the symbols in numbers.
>
> There have been many recent works that show that LLMs can perform surprisingly well forecasting tasks zero-shot. For instance, Gruver et al. (2024)  and Requeima et al. (2024) showed that accurate forecasts could be obtained through zero-shot sequence completion. Our work extends such evaluations of LLMs to the relatively new problem setting of context-aided forecasting, and shows that they are especially useful in this problem setting. Moreover, as we point out in Section 5.3, LLM baselines are good forecasters also when compared to quantitative forecasting models in a purely numerical, no-context setting. Our results confirm that LLMs are surprisingly strong at forecasting, but further research would be needed to understand the surprising performance of LLMs in forecasting, in efforts to improve these models.
>
> **References**:
>
> Nate Gruver, Marc Finzi, Shikai Qiu, and Andrew G Wilson. Large language models are zero-shot time series forecasters. Advances in Neural Information Processing Systems, 36, 2024.
>
> James Requeima, John Bronskill, Dami Choi, Richard E Turner, and David Duvenaud. LLM processes: Numerical predictive distributions conditioned on natural language. arXiv preprint arXiv:2405.12856, 2024.
>
> ---
>
> **Questions**:
>
> ---
>
> > Q1: can you just open-source all the related-materials and I think it's better to let all people to judge if it has enough value and easy to use.
>
> We plan to release the code and datasets used for the benchmark and all experiments in the paper, under an Apache-2.0 license upon acceptance. Anonymized code for all tasks and experiments is available here: https://anonymous.4open.science/r/context-is-key-forecasting-E391/.
>
> ---
>
> > Q2: how do you choose the best-quality text and how would you access this quality?
>
> We wrote the text of all the tasks ourselves. To ensure their quality, we used peer review to make sure that the texts are high-quality and also important for the forecasting task. Please see our answer to W1 for the full task creation process.
>
>
> ---
>
> > Q3: Is there any better way to cooperate the numbers in prompt to evaluate the effect of the benchmark?
>
> Thank you for your interesting question. The scope of our work is to establish a high-quality benchmark to evaluate models that jointly use text and numerical history for forecasting. Exploring how to best leverage numerical information and text is an interesting research direction that we leave for future work, that this benchmark enables quantifying progress on.
>
> ---
>
> > Q4: text brings more calculation, but the effect of the texts is well discussed. How to balance and choose the most effective one?
>
> All tasks in the benchmark are designed such that the text is absolutely necessary to be taken into account. We write the context ourselves to ensure its relevance (see our answer to W1 for the full task creation process). There are tasks that require the model to choose the most relevant text from the given textual context, as opposed to providing the filtered, relevant text directly. These tasks are tagged under the Retrieval from Context, (see tasks under “Retrieval:context” in [the benchmark visualization website](https://anon-forecast.github.io/benchmark_report_dev/)). Rightly, we find that when contexts are already filtered to only have the relevant context, models perform better at the task.
>
> For example, we compare two tasks on predicting the Unemployment Rate of a county. For the UnemploymentCountyUsingSingleStateData task, we give context relevant to the variable (Unemployment Rate of the State in which the county is) - this task is visualized [here](https://anon-forecast.github.io/benchmark_report_dev/UnemploymentCountyUsingSingleStateData.html). In the UnemploymentCountyUsingExpliciteMultipleStateData task, in addition to the context about the relevant state, the context includes unemployment rates of 2 other states - task visualized [here](https://anon-forecast.github.io/benchmark_report_dev/UnemploymentCountyUsingExplicitMultipleStateData.html). [This figure](https://raw.githubusercontent.com/anon-forecast/benchmark_report_dev/refs/heads/main/iclr_rebuttal_resources/retrieval_difference_unemployment.png) visualizes the difference in performance in these tasks for 3 different models. We find that models perform much better when only the relevant county’s data is provided, as opposed to the data from multiple counties.

---

> > ### Author Response · Authors · 2024-11-22
> >
> > Dear reviewer, we would like to make sure your comments and concerns have been addressed. Please let us know if you need any additional clarifications.

---

> > > ### Author Response · Authors · 2024-11-25
> > >
> > > Dear reviewer,
> > >
> > > Thank you again for your feedback; the manuscript has undoubtedly improved based on your suggestions. Please find below all relevant changes, which are in blue in the paper for visibility.
> > >
> > > We have added a section on the task creation process in **appendix A.2.**, and added references to it in multiple parts of Section 3.
> > >
> > > We have further added a section in **appendix A.3.**, on the LLM-based critique on the relevance of context, to further validate the quality of the benchmark, and added an reference to it in Section 3.
> > >
> > > We have added a section in **appendix C.7.** on the impact of relevant and irrelevant information in the context, to study if models perform better on context that has already been filtered to only contain relevant information. A reference to appendix C has been added in Section 5.3.
> > >
> > > Finally, we have added a reference to the open-source code for all tasks and experiments available here: https://anonymous.4open.science/r/context-is-key-forecasting-E391/ in **Section 3**.
> > >
> > > ---
> > >
> > > We would also like to point out that the paper has several other new additions, such as the inclusion of TimeGEN (a foundation model), ETS and ARIMA (statistical models) in the benchmark, an overview of the distribution of the lengths of the natural language context, numerical history and prediction horizon in appendix A.7. etc.
> > >
> > > ---
> > >
> > > We would like to make sure your comments and concerns have been addressed in the updated manuscript. Please let us know if you need any additional clarifications.

---

> ### Author Response · Authors · 2024-11-29
>
> Dear Reviewer RToE,
>
> We want to thank you for the time you took to review our work. We appreciate the suggestions and the detailed feedback, and we have carefully worked to address any concerns.
>
> As the discussion period comes to a close, if you feel that your concerns have been addressed, we would greatly appreciate any change to your score. If you require any additional clarifications, we are happy to provide them.
>
> Thank you again for your time and for providing thoughtful comments.
>
> Kind regards,
>
> The Authors.

---

> > ### Author Response · Authors · 2024-12-03
> >
> > Dear Reviewer RToE,
> >
> > Again, thank you for your time and efforts in the review process. As it stands, the paper has two scores = 5 and one = 6 which, as far as we can tell based on the public comments, have not been revised following the significant revisions made to the paper and experiments. We believe that major outstanding concerns, such as those on the task creation process, have been addressed.
> >
> > As the reviewer response period ends **today AoE**, we urge you to please reconsider your score, as **your opinion may be key in the present context**. We respect your judgement and would appreciate a signal, even if you choose to maintain your score.
> >
> > Kind regards,
> >
> > The Authors

---

### Official Review · Reviewer_kJ9d · 2024-10-29

**Soundness:** 3
**Presentation:** 4
**Contribution:** 3
**Rating:** 6
**Confidence:** 4

**Summary:**

The paper addresses a pertinent issue: the scarcity of benchmarks for context-enhanced forecasting. Although many recent studies focus on predicting future values using textual cues, there is limited data available for training and evaluating such models. This paper introduces a manually curated benchmark specifically for text-aided time-series forecasting, featuring numerical time series data paired with contextual text. The benchmark is extensive in its selection of domains and tasks, providing a comprehensive resource. It is also thoroughly evaluated across a wide range of forecasting models. Furthermore, the paper introduces a novel metric for forecast evaluation.

**Strengths:**

This work is original and highly relevant to the current trend of LLM-based forecasting. The benchmark is well-designed, offering broad coverage across various domains and tasks, with results for multiple forecasters included to showcase its capabilities. The analysis and results are clearly presented, with examples and figures that effectively illustrate key benchmark characteristics and greatly enhance comprehension. Additionally, the paper introduces an intriguing new metric for evaluating forecast quality within relevant regions, which adds depth to the evaluation framework.

**Weaknesses:**

While this work is both novel and relevant, it lacks analytical rigor in its benchmark evaluation. The reader is supposed to assume that all provided textual information contributes meaningfully to the forecasts, with minimal evidence beyond the overall performance improvements seen for the entire dataset. Evaluating the relevance of the textual context—perhaps through methods such as LLM-based or human assessments—would strengthen the claim that these textual data are correct and relevant descriptions for the time series. Additionally, some covariate information appears to include future events, which a causal forecaster would not typically access (e.g., “Speed From Load” in Appendix B.2). This raises concerns about causal consistency, as there is no mechanism for systematically separating different types of contextual data other than through manual or LLM editing. Such limitations could present challenges for users who want to avoid incorporating future or irrelevant covariate information in their experiments.

The paper also lacks clarity regarding the historical context length and forecasting horizon—key details that should be specified. Furthermore, the reliability of the benchmark results hinges on the sample size, yet no information about the number of samples for the datasets is provided.

Perhaps the most notable contribution of the paper is its manual curation of the dataset. However, this process remains underexplained. Details such as the curation methodology, sources of textual data, and the criteria used for selecting relevant data are absent, which limits the transparency of this work.  A more comprehensive discussion of these aspects would significantly enhance the credibility and utility of the dataset for future research.

**Questions:**

Questions:
1) Does the covariate information always imply the availability of the future values of the covariates, or are there examples with covariate information provided only for the history time series?
2) While I believe it would be handled in a discussion of the manual curation process, I wanted to know if the entire manual curation for so many datasets was done by the authors who would be credited for the paper or if any form of crowdsourcing was utilized for the manual tasks. Were the annotators paid fairly for their efforts if any crowdsourcing was utilized? Was any LLM used during the manual curation process?

Suggestions for the authors:
1) Include a discussion on the manual curation process with information on the data sources and the selection of relevant context from them.
2) Include benchmark descriptions mentioning the sequence lengths, prediction horizon, and number of sequences present in each benchmark to build confidence in the robustness of the work.
3) Provide some ideas for separating different contextual information in the text.
4) Highlight the efforts taken to ensure that any contextual information paired with the time series is actually correct/relevant. Do a similar task to highlight the relevance of the said "region of interest."

**Details Of Ethics Concerns:**

The paper does not provide any details about the manual curation process involved in creating the benchmark. Given the scale of data curation implied, it seems unlikely that this task could have been completed by a small group of authors without support from crowdsourcing, LLMs, or other manual annotators. The lack of discussion regarding these aspects raises questions about the claim of "careful manual curation." If crowdsourcing or external labor was utilized, the absence of a description of the tasks, associated costs, and acknowledgment of contributors may hint towards uncredited or underpaid labor.

---

> ### Author Response · Authors · 2024-11-19
>
> Thank you for your feedback and for emphasizing the originality, relevance, and clarity of our work. We appreciate the depth of your review. Please find answers to your comments and questions below, and do let us know if you would like further clarification on anything else. We hope that these explanations will resolve your concerns, especially regarding soundness and transparency, and that you will consider increasing your score.
>
> We first respond to W4 and Q2 together, and then move to addressing the other weaknesses and questions thereafter.
>
> ---
>
> > W4: Perhaps the most notable contribution of the paper is its manual curation of the dataset. However, this process remains underexplained.
>
> We apologize for the lack of clarity here. The word “curate” was poorly chosen in the contribution section. We hereby clarify the dataset creation process.
>
> All tasks were manually designed, from scratch, by the authors of this work **without resorting to external annotators or LLMs**. First, we identified high-quality sources of public time series data from various application domains (see Section 3.1). Special care was taken to find data sources that are continuously updated to facilitate future benchmark updates. Second, we established the categorization for sources of context (Section 3.2) and capabilities (Section 3.3) as a framework to guide the creation of new tasks and ensure their diversity. Third, team members created the tasks, each time selecting a data source, implementing a time series window selection strategy (e.g., short or long history), brainstorming about context types and capabilities required to solve the forecasting problem, writing a code to generate the context (e.g., calculating statistics of the series beyond the observed numerical history), and finally, if required, writing code to modify the time series data to reflect the context (e.g., introducing some spikes in future values).
>
> Then, the tasks were peer-reviewed by a committee with significant expertise in time series forecasting (co-authors of this work). The creator of each task was not allowed to participate in the review. The review ensured that the text was of high quality, that it undoubtedly enabled a better forecast, and that the context source and capability tags were well-assigned. If a task was deemed of not high enough quality, it was either returned for revisions, or rejected.
>
> The code for all tasks and experiments is available here: https://anonymous.4open.science/r/context-is-key-forecasting-E391/. An example task can be found here: https://anonymous.4open.science/r/context-is-key-forecasting-E391/cik_benchmark/tasks/montreal_fire/short_history.py, where the time series window selection occurs from L94-112 and the context generation occurs from L114-158.
>
> We will add the same discussion to our paper in the next revision.
>
> ---
>
> > Q2: While I believe it would be handled in a discussion of the manual curation process, I wanted to know if the entire manual curation for so many datasets was done by the authors who would be credited for the paper or if any form of crowdsourcing was utilized for the manual tasks. Were the annotators paid fairly for their efforts if any crowdsourcing was utilized? Was any LLM used during the manual curation process?
>
> The above answer should clarify that no annotators, crowdsourcing, or LLMs were used to either delegate or automate the creation of tasks in this benchmark. It was manually created by time series experts to ensure its quality and relevance, and all are credited as authors of the paper.

---

> ### Author Response · Authors · 2024-11-19
>
> > W1: The reader is supposed to assume that all provided textual information contributes meaningfully to the forecasts [...] evaluating the relevance of the textual context [...] through methods such as LLM-based or human assessments would strengthen the claims
>
> Indeed, it is crucial for readers to be able to assess the quality of the tasks. This was the motivation for providing the [website](https://anon-forecast.github.io/benchmark_report_dev/) alongside the paper, and examples in Appendix B.
> Nevertheless, we appreciate your suggestion and,  following your recommendation, we further assessed the quality of the context using an LLM-based critique. The critique was built by prompting GPT-4o with the historical and future numerical data, as well as the context, and asking it to assess if its estimation of future values would either improve or degrade based on the context (see this [file](https://github.com/anon-forecast/benchmark_report_dev/blob/main/iclr_rebuttal_resources/llm_validation.py) for the code; the prompt used is given in lines 18-66). We ran this critique on 5 instances from each of the 71 tasks and report results in [this figure](https://raw.githubusercontent.com/anon-forecast/benchmark_report_dev/refs/heads/main/iclr_rebuttal_resources/gpt_analysis.png). In short, for all tasks, the critique believes that the context enables better forecasts (significantly better for most tasks).
>
> The output of the critique, including justifications of its score for each task instance, is attached to our response as a CSV [here](https://raw.githubusercontent.com/anon-forecast/benchmark_report_dev/refs/heads/main/iclr_rebuttal_resources/llm_validation.csv). A detailed presentation of the critique and these new results will be included in the appendix of the revised paper.
>
> We hope this resolves your concerns regarding analytical rigor in evaluation. Please let us know if you would like us to deepen the analysis.
>
> ---
>
> > W2: some covariate information appears to include future events, which a causal forecaster would not typically access (e.g., “Speed From Load” in Appendix B.2). This raises concerns about causal consistency, as there is no mechanism for systematically separating different types of contextual data other than through manual or LLM editing. Such limitations could present challenges for users who want to avoid incorporating future or irrelevant covariate information in their experiments.
>
> The mentioned future events are either scenarios a user would like to simulate (e.g., in UnemploymentCounty tasks such as [this task](https://anon-forecast.github.io/benchmark_report_dev/UnemploymentCountyUsingMultipleStateData.html), or [this task](https://anon-forecast.github.io/benchmark_report_dev/UnemploymentCountyUsingSingleStateData.html)) or control variates that do not break causal consistency (e.g., in [SpeedFromLoad](https://anon-forecast.github.io/benchmark_report_dev/SpeedFromLoadTask.html), knowing that the load is set to a certain value in the future does not leak useful information for predicting the current speed). In either case, tasks containing this type of context are clearly tagged as containing ‘Future information’ or 'Causal information’, and these components of the context are defined in separate variables in the code (see e.g. line 75 [here](https://anonymous.4open.science/r/context-is-key-forecasting-E391/cik_benchmark/tasks/causal_chambers.py)). Thereby, in these cases, a user can easily remove them from the context for evaluation for their experiments.
>
> ---
>
> > W3.1: Lacks clarity regarding the historical context length and forecasting horizon
>
> Please see [this image](https://raw.githubusercontent.com/anon-forecast/benchmark_report_dev/refs/heads/main/iclr_rebuttal_resources/context_history_pred_lengths.png), for histograms depicting the distribution of lengths for the context, numerical history and target length of a set of five instances for each task in CiK.  We measure the length of the natural language context in characters, and the numerical sequences in floats.
>
> We will add this information to the paper in the next revision.
>
> > W3.2: The reliability of the benchmark results hinges on the sample size, yet no information about the number of samples for the datasets is provided.
>
> This information is available in Section 5.1 (Evaluation Protocol) of the main text. Forecast distributions are estimated with 25 samples from each model and, for each of the 71 tasks, we consider 5 random instances to evaluate the models. This was done in order to make the evaluation of the benchmark sufficiently accurate, while being reproducible and affordable.

---

> ### Author Response · Authors · 2024-11-19
>
> > Q1: Does the covariate information always imply the availability of the future values of the covariates, or are there examples with covariate information provided only for the history time series?
>
> There are tasks where the covariate information is only available for the history. Please see [this example](https://anon-forecast.github.io/benchmark_report_dev/MontrealFireNauticalRescueAnalogyFullLocalizationMaybeWaterTask.html) taken from the benchmark report. In this task, the covariates: the presence of water bodies and the number of incidents in the city's other boroughs, are only provided for the history.
>
> **Suggestions for the authors**:
>
> Again, thank you for your efforts in reviewing our work and for these suggestions which will definitely improve the quality of our contribution. We implement each of these and comment on them below:
>
> > 1) Include a discussion on the manual curation process with information on the data sources and the selection of relevant context from them.
>
> Please see our response to W4.
>
> > 2) Include benchmark descriptions mentioning the sequence lengths, prediction horizon, and number of sequences present in each benchmark to build confidence in the robustness of the work.
>
> Please see our response to W3.
>
> > 3) Provide some ideas for separating different contextual information in the text.
>
> The various sources of contextual information are exemplified in Fig. 3 and defined in the text in Section 3.2. Moreover, each task is tagged with its context sources in the [visualization website](https://anon-forecast.github.io/benchmark_report_dev) that accompanies the benchmark. We also give concrete examples of domain, context source,  and capability assignments in Appendix B. Further, the various components of the context are defined in separate variables in the code (see e.g. line 75 [here](https://anonymous.4open.science/r/context-is-key-forecasting-E391/cik_benchmark/tasks/causal_chambers.py) for an example) for clarity. Please let us know if this provides clarification. We are more than happy to discuss this further.
>
> > 4) Highlight the efforts taken to ensure that any contextual information paired with the time series is actually correct/relevant. Do a similar task to highlight the relevance of the said "region of interest."
>
> We include a new LLM-based analysis of the relevance of the context for each task. Please see our response to W1. The regions of interest are highlighted only for tasks where context refers to a set of timesteps, for which we thoroughly verify the code for each task manually to ensure a consistency between the highlighted timesteps and the timesteps mentioned in the context text.  For examples, see lines 70 to 84 in [this task in the code](https://anonymous.4open.science/r/context-is-key-forecasting-E391/cik_benchmark/tasks/electricity_tasks.py) where the exact same variables are used in the context and to consider the region of interest during evaluation.

---

> > ### Author Response · Authors · 2024-11-22
> >
> > Dear reviewer, we would like to make sure your comments and concerns have been addressed. Please let us know if you need any additional clarifications.

---

> > > ### Comment · Reviewer_kJ9d · 2024-11-23
> > >
> > > Dear authors, thank you for the clarifications, especially on the manual curation process. I understand that no external annotators or LLMs were used, and therefore would like to commend the authors for the efforts taken to manually create such a massive benchmark.
> > >
> > > I believe the manuscript has not yet been revised following the reviews. I would be open to updating my score after reviewing the revised version.

---

> ### Author Response · Authors · 2024-11-25
>
> Dear reviewer,
>
> Thank you again for your feedback; the manuscript has undoubtedly improved based on your suggestions. Please find below all relevant changes, which are in blue in the paper for visibility.
>
> We have added a section on the task creation process in **appendix A.2.**, and added references to it in multiple parts of Section 3.
>
> We have added a section in **appendix A.3.**, on the LLM-based critique on the relevance of context, to further validate the quality of the benchmark, and added an reference to it in Section 3.
>
> We have added an overview of the distribution of the lengths of the natural language context, numerical history and prediction horizon in **appendix A.7**, and added an reference to it in Section 3.
>
> Finally, we have added a reference to the open-source code for all tasks and experiments available here: https://anonymous.4open.science/r/context-is-key-forecasting-E391/ in **Section 3**.
>
> ---
>
> We would also like to point out that the paper has several other new additions, such as the inclusion of TimeGEN (a foundation model), ETS and ARIMA (statistical models) in the benchmark, a study on the impact of having both relevant and irrelevant information in the context in Appendix C.7. etc.
>
> ---
>
> We would like to make sure your comments and concerns have been addressed in the updated manuscript. Please let us know if you need any additional clarifications.

---

> ### Author Response · Authors · 2024-11-29
> **Thank you**
>
> Dear Reviewer kJ9d,
>
> We want to thank you for the time you took to review our work. We appreciate the suggestions and the detailed feedback, and we have carefully worked to address any concerns.
>
> As the discussion period comes to a close, if you feel that your concerns have been addressed, we would greatly appreciate any change to your score. If you require any additional clarifications, we are happy to provide them.
>
> Thank you again for your time and for providing thoughtful comments.
>
> Kind regards,
>
> The Authors.

---

> > ### Author Response · Authors · 2024-12-03
> >
> > Dear Reviewer kJ9d,
> >
> > Again, thank you for your time and efforts in the review process. As it stands, the paper has two scores = 5 and one = 6 which, as far as we can tell based on the public comments, have not been revised following the significant revisions made to the paper and experiments. We believe that major outstanding concerns, such as those on the task creation process, have been addressed.
> >
> > As the reviewer response period ends **today AoE**, we urge you to please reconsider your score, as **your opinion may be key in the present context**. We respect your judgement and would appreciate a signal, even if you choose to maintain your score.
> >
> > Kind regards,
> >
> > The Authors

---

> > > ### Comment · Reviewer_kJ9d · 2024-12-03
> > >
> > > Having gone through the revised manuscript and the other comments, I have decided to update the score to 6.

---

> > > > ### Author Response · Authors · 2024-12-04
> > > >
> > > > Thank you, we appreciate your time and effort.
> > > >
> > > > Sincerely,
> > > > The Authors

---

### Official Review · Reviewer_6GZP · 2024-11-02

**Soundness:** 3
**Presentation:** 3
**Contribution:** 3
**Rating:** 6
**Confidence:** 2

**Summary:**

This paper explores the integration of contextual textual data with numerical time series to improve time series forecasting. It introduces the CiK benchmark, consisting of 71 diverse forecasting tasks across multiple domains. Unlike existing benchmarks, CiK requires models to process both numerical data and associated textual context, reflecting real-world complexities such as seasonal trends or future constraints (e.g., maintenance periods). The authors also propose a novel metric, the Region of Interest Continuous Ranked Probability Score (RCRPS), which weights context-sensitive time windows and penalizes constraint violations.

**Strengths:**

The paper provides rigorous evaluation, testing various model types and prompting techniques. The introduction of the RCRPS metric enhances assessment accuracy by factoring in context relevance and constraint adherence. The combination of text and time series has always been something that researchers in the field want to try, and this benchmark provides a good research foundation. The writing structure of the paper is very clear

**Weaknesses:**

From the experimental results, we can see that text provides a good auxiliary role, but this method should be limited to models with LLM as the backbone.

**Questions:**

1. Are all the time series is univariate?
2. In this benchmark construction, have you tried very refined text rather than information at a specific time step or overall? How did it perform?
3. Regarding retrieval, if I use the time series segment corresponding to a specific text as the retrieval "text", will the performance be better? Because this is more direct.

---

> ### Author Response · Authors · 2024-11-19
>
> We thank the reviewer for the positive evaluation of the work, and for highlighting the clear presentation, the diversity of models and prompting techniques, the importance of the introduced RCRPS metric and the rigorous evaluation. We are pleased that you consider that our work provides a good foundation for an important question in the research field. We are glad to clarify the points brought up in the review.
>
> ---
>
> > W1: From the experimental results, we can see that text provides a good auxiliary role, but this method should be limited to models with LLM as the backbone.
>
> Indeed, for a model to perform well on the CiK benchmark, it must effectively process natural language context. Currently, the top-performing models are large language models (LLMs) that are directly prompted to generate forecasts. This surprising result may stem from their ability to understand context, their enhanced reasoning capabilities, and their training for sequence completion—a task similar to time series forecasting.
>
> However, future methodologies need not be restricted to models with LLM backbones. Several time series models can utilize context provided as numerical covariates. If there is an effective way to convert natural language into numerical representations (such as text embeddings), it could be possible to leverage these models to generate accurate context-aware forecasts, even without an LLM backbone. While it remains to be seen how the field will address this challenge, one certainty is that the CiK benchmark will provide the community with a reliable means to measure progress toward this goal.
>
> ---
>
> >Q1: Are all the time series is univariate?
>
> Yes, all tasks in the benchmark require forecasting univariate time series. This choice was made for simplicity and, since we wanted to focus on evaluating the ability to use natural language context to improve forecasts. Introducing additional challenges, such as modeling multivariate dependencies, would have complexified the result analysis, by entangling failure to use context with failure to model multivariate dependencies or challenges in assessing the quality of multivariate forecast distributions (see Marcotte et al., 2023 for a discussion of pitfalls of multivariate metrics). Moreover, existing benchmarks, such as the datasets in the Monash Time Series Forecasting Archive (Godahewa et al. 2021) are well-suited to evaluate multivariate forecasting and can be viewed as being complementary to CiK. Nevertheless, we agree that including multivariate tasks is a natural extension to CiK and mention it in the “Future Work” section of the discussion.
>
> References:
>
> 1. Étienne Marcotte, Valentina Zantedeschi, Alexandre Drouin, and Nicolas Chapados. "Regions of reliability in the evaluation of multivariate probabilistic forecasts." In International Conference on Machine Learning, pp. 23958-24004. PMLR, 2023.
>
> 2. Godahewa, Rakshitha, Christoph Bergmeir, Geoffrey I. Webb, Rob J. Hyndman, and Pablo Montero-Manso. "Monash time series forecasting archive." arXiv preprint arXiv:2105.06643 (2021).
>
> ---

---

> ### Author Response · Authors · 2024-11-19
>
> > Q2: In this benchmark construction, have you tried very refined text rather than information at a specific time step or overall? How did it perform?
>
> For some tasks in the benchmark, the context only applies to specific timesteps in the prediction region, and the context explicitly refers to these regions. Two such tasks are the Electricity Consumption Increase task (shown in App. B.2; visualized [here](https://anon-forecast.github.io/benchmark_report_dev/ElectricityIncreaseInPredictionTask.html)) and the ATM Maintenance task (shown in App. B.3; visualized [here](https://anon-forecast.github.io/benchmark_report_dev/ATMUnderPeriodicMaintenanceTaskWithConclusion.html)). In both these tasks, the context was manually crafted and the time series was modified at the specific timesteps where the context would apply. In such tasks, we highlight the region where context is necessary, as the region of interest (as shown in dark green shade in the figures) which is taken into consideration in the evaluation metric (as explained in the Region of interest (RoI) paragraph of Section 4). More example tasks from the benchmark report: [Example 1](https://anon-forecast.github.io/benchmark_report_dev/ExplicitWithDatesAndDaysTrafficForecastTaskwithHolidaysInPredictionWindow.html), [Example 2](https://anon-forecast.github.io/benchmark_report_dev/IncreasedWithdrawalScenario.html), [Example 3](https://anon-forecast.github.io/benchmark_report_dev/SensorSpikeTask.html).
>
> The benchmark also contains tasks where the context refers to all timesteps, such as containing constraints that apply to all timesteps (example in App. B.1; visualized [here](https://anon-forecast.github.io/benchmark_report_dev/BoundedPredConstraintsBasedOnPredQuantilesTask.html)), is descriptive and contains intemporal information about the variable (examples in App. B.4 and visualized [here](https://anon-forecast.github.io/benchmark_report_dev/MontrealFireFieldFireExplicitShortHistoryTask.html); another example in App. B.5 and visualized [here](https://anon-forecast.github.io/benchmark_report_dev/SimilarLocationDaySolarForecastTask.html)) or containing causal information (referring to dependencies between variables and covariates). These tasks ensure that the context is necessary for accurate predictions, by limiting the history to be insufficient for the prediction horizon. More example tasks taken from the benchmark report: [Example 1](https://anon-forecast.github.io/benchmark_report_dev/MontrealFireFieldAndTrashNeutralToneExplicitCausalConfoundingTask.html), [Example 2](https://anon-forecast.github.io/benchmark_report_dev/MontrealFireFieldFireExplicitShortHistoryTask.html).
>
> ---
>
> > Q3: Regarding retrieval, if I use the time series segment corresponding to a specific text as the retrieval "text", will the performance be better? Because this is more direct.
>
> All tasks in the benchmark are designed such that the text is absolutely necessary to be taken into account. We write the context ourselves to ensure its relevance (see our answer to W1 for the full task creation process). There are tasks that require the model to choose the most relevant text from the given textual context, as opposed to providing the filtered, relevant text directly. These tasks are tagged under the Retrieval from Context, (see tasks under “Retrieval:context” in [the benchmark visualization website](https://anon-forecast.github.io/benchmark_report_dev/)). Rightly, we find that when contexts are already filtered to only have the relevant context, models perform better at the task.
>
> For example, we compare two tasks on  predicting the Unemployment Rate of a county. For the UnemploymentCountyUsingSingleStateData task, we give context relevant to the variable (Unemployment Rate of the State in which the county is) - this task is visualized [here](https://anon-forecast.github.io/benchmark_report_dev/UnemploymentCountyUsingSingleStateData.html). In the UnemploymentCountyUsingExpliciteMultipleStateData task, in addition to the context about the relevant state, the context includes unemployment rates of 2 other states - task visualized [here](https://anon-forecast.github.io/benchmark_report_dev/UnemploymentCountyUsingExplicitMultipleStateData.html). [This figure](https://raw.githubusercontent.com/anon-forecast/benchmark_report_dev/refs/heads/main/iclr_rebuttal_resources/retrieval_difference_unemployment.png) visualizes the difference in performance in these tasks for 3 different models. We find that models perform much better when only the relevant state's data is provided, as opposed to the context also containing data from other states.

---

> > ### Author Response · Authors · 2024-11-22
> >
> > Dear reviewer, we would like to make sure your comments and concerns have been addressed. Please let us know if you need any additional clarifications.

---

> > > ### Comment · Reviewer_6GZP · 2024-11-25
> > >
> > > Dear Author,
> > >
> > > I would like to thank the authors for addressing most of my concerns. I hope they incorporate these changes in their revised version of the paper. After reviewing other comments, I believe it would be helpful if the authors could provide a revised paper for further clarification. Until then, I have decided to maintain my score.

---

> ### Author Response · Authors · 2024-11-25
>
> Dear reviewer,
>
> Thank you again for your feedback; the manuscript has undoubtedly improved based on your suggestions.
>
> We have added an explicit note on the univariate nature of tasks in our benchmark,  and suggest multivariate tasks as a natural extension in **Section 7**.
>
> We have added a section in **appendix C.7.** on the impact of relevant and irrelevant information in the context, to study if models perform better on context that has already been filtered to only contain relevant information. A reference to appendix C has been added in Section 5.3.
>
> ---
>
> We would also like to point out all other major changes in the paper below.
>
> We have added a section on the task creation process in **appendix A.2.**, and a section on the LLM-based critique on the relevance of context in **appendix A.3.**, to further validate the quality of the benchmark.
>
> The paper has several other new additions, such as the inclusion of **new models**: TimeGEN (a foundation model), ETS and ARIMA (statistical models) in the benchmark, an overview of the distribution of the lengths of the natural language context, numerical history and prediction horizon in **appendix A.7.** etc.
>
> Finally, we have added a reference to the open-source code for all tasks and experiments available here: https://anonymous.4open.science/r/context-is-key-forecasting-E391/ in **Section 3**.
>
> ---
>
> We would like to make sure your comments and concerns have been addressed in the updated manuscript. Please let us know if you need any additional clarifications.

---

> ### Author Response · Authors · 2024-11-29
>
> Dear Reviewer 6GZP,
>
> We want to thank you for the time you took to review our work. We appreciate the suggestions and the detailed feedback, and we have carefully worked to address any concerns.
>
> As the discussion period comes to a close, if you feel that your concerns have been addressed, we would greatly appreciate any change to your score. If you require any additional clarifications, we are happy to provide them.
>
> Thank you again for your time and for providing thoughtful comments.
>
> Kind regards,
>
> The Authors.

---

> > ### Author Response · Authors · 2024-12-03
> >
> > Dear Reviewer 6GZP,
> >
> > Again, thank you for your time and efforts in the review process. As it stands, the paper has two scores = 5 and one = 6 which, as far as we can tell based on the public comments, have not been revised following the significant revisions made to the paper and experiments. We believe that major outstanding concerns, such as those on the task creation process, have been addressed.
> >
> > As the reviewer response period ends today AoE, we urge you to please reconsider your score, as your opinion may be key in the present context. We respect your judgement and would appreciate a signal, even if you choose to maintain your score.
> >
> > Kind regards,
> >
> > The Authors

---

### Official Review · Reviewer_7GBq · 2024-11-04

**Soundness:** 1
**Presentation:** 2
**Contribution:** 2
**Rating:** 3
**Confidence:** 4

**Summary:**

This paper introduces a new benchmark, CiK, to evaluate how well forecasting models can use essential textual context alongside numerical data to improve prediction accuracy. The benchmark includes 71 tasks across various fields, where models need to integrate natural language information—like historical trends or future events—with time series data for accurate forecasts. To assess performance, the authors develop the Region of Interest CRPS (RCRPS) metric, which emphasizes context-sensitive parts of the forecast and accounts for constraints stated in the text. Through experiments, they show that a simple prompting method for large language models (LLMs) outperforms traditional forecasting methods, underscoring the importance of context for improved predictions.

**Strengths:**

1. Good writing and easy to follow
2. CiK uniquely requires essential textual context for forecasting, marking a new direction in multimodal prediction.
3. The benchmark is robust, with real-world tasks and a novel, context-focused RCRPS metric.

**Weaknesses:**

1. Missing Information on Context Annotations:
The paper relies on carefully crafted textual contexts but omits crucial details about the annotation process, such as the guidelines provided to annotators, the number and qualifications of annotators, methods used to resolve disagreements, and quantitative measures of inter-annotator agreement (IAA). This lack of information raises questions about the consistency and reliability of the annotations. Including examples or a sample of the annotation guidelines, a description of annotator expertise, and the process for calculating IAA would strengthen the benchmark’s credibility and demonstrate rigorous annotation practices.

2. Limited Benchmark Novelty:
While the CiK benchmark combines existing time-series datasets with manually created textual contexts, its contribution to multimodal benchmarks is limited in novelty. The approach resembles prior work that integrates time-series data with textual sources like news or social media.[1][2] To clarify its uniqueness, the authors could provide comparisons to specific existing work and clearly articulate the novel contributions or improvements over these prior works. Additionally, the manual creation of contexts raises concerns about scalability; introducing semi-automated methods or leveraging AI to generate contexts could make the benchmark more practical for real-world applications and future expansions.

3. Ambiguous Task Type Annotations:
The paper lacks clarity in task categorization, with no explicit definitions provided for each model capability category. For instance, “instruction following” is inconsistently applied, leaving tasks like “Public Safety” uncategorized, despite requiring instruction interpretation. It would be helpful if the authors included definitions for each capability category, specified criteria for categorizing tasks, and offered examples illustrating why certain tasks fall into each category. This additional information would clarify the task taxonomy and improve the interpretability of the benchmark structure.

4. Unexplained Results Discrepancies:
Certain performance discrepancies raise concerns about the validity of the benchmark’s metrics. For example, LLMP Mixtral-8x7B shows lower CRPS performance with context compared to without in Figures 4 and 5, yet it still appears to outperform traditional quantitative models when using context. This inconsistency suggests that CRPS may not fully capture the forecast quality in multimodal contexts. The authors could benefit from including a discussion on why CRPS was chosen, exploring alternative or complementary metrics, or providing a deeper analysis of the observed discrepancies to enhance the reliability of the reported results.

5. Limited Model Variety:
The benchmark’s experimental setup primarily includes larger models like Llama-3 series, limiting the variety across model sizes and architectures, as well as smaller models like Mistral, Qwen, and Falcon. A more diverse set of models, including smaller or less resource-intensive models, could offer broader insights and improve the benchmark’s generalizability. Explaining any practical or strategic reasons for the current model selection would provide additional context. Exploration of smaller models or discussing plans for future testing would also enhance the paper’s impact.

[1] Sawhney, Ramit, Arnav Wadhwa and Shivam Agarwal. “FAST: Financial News and Tweet Based Time Aware Network for Stock Trading.” Conference of the European Chapter of the Association for Computational Linguistics (2021).

[2] Liu, Mengpu, Mengying Zhu, Xiuyuan Wang, Guofang Ma, Jianwei Yin and Xiaolin Zheng. “ECHO-GL: Earnings Calls-Driven Heterogeneous Graph Learning for Stock Movement Prediction.” AAAI Conference on Artificial Intelligence (2024).

**Questions:**

1. How the textual contexts were annotated, including any guidelines, annotator expertise, and inter-annotator agreement (IAA) metrics?

2. Do the authors envision methods to automate or partially automate this process, such as using existing NLP techniques to generate context?

3. Could the authors provide explicit definitions for each capability and clarify the criteria used to categorize tasks?

4. Could the authors clarify whether this discrepancy suggests limitations of the CRPS metric or other factors in the benchmark design?

---

> ### Author Response · Authors · 2024-11-19
>
> Thank you for your thorough evaluation, which recognizes our manuscript’s clarity, the novel direction that our benchmark opens for multimodal prediction, the new RCRPS metric, and the robustness and real-world relevance of our benchmark. We hope our responses clarify your concerns, especially regarding soundness, and that you will consider revising your score. If not, please let us know and we will do our best to clarify.
>
> ---
>
> > W1: Missing Information on Context Annotations
>
> We apologize for the lack of clarity in describing how tasks were created. In particular, the term “curation” used in the contributions section was poorly chosen. We clarify the dataset creation process, including the generation of textual contexts, below.
>
> We, the authors, designed each task ourselves by handwriting context to accompany manually chosen sources of time series data. In particular, we did not use any external annotators for this work. First, we sought publicly available time series data across various domains. Then, we developed the framework around sources of context (Sec 3.2) and capabilities (Sec 3.3) to encourage the creation of a diverse, comprehensive set of forecasting tasks. Each task was created separately by team members, who would 1) select a time series data source and brainstorm about a context-aided forecasting task pertinent to that data source, 2) implement a selection strategy for a window according to the task idea, 3) brainstorm about context types and capabilities that could help address the particular forecasting problem, 4) write a template for the context and any accompanying code to instantiate the template (e.g. statistics on past values to be included in the text), and finally, if the task required it, 5) modify the time series to reflect the context (e.g. adding a spike to the ground truth in the forecast region that would be only predictable by the textual context).
>
> Tasks, when created, were peer-reviewed by a committee of time series forecasting experts who are co-authors of this work. To ensure the validity of this peer-review, the original creator of a given task did not participate in the committee. This review process ensured that forecasting experts, when faced with a task for the first time, could validate whether the context 1) could improve the forecast accuracy if used properly, 2) was accurate and comprehensible, and 3) that the context sources and capabilities assigned to a given task were appropriate. If the task was deemed of not high enough quality, it was either returned for revisions or rejected.
>
> To further illustrate how task instances are generated, the reviewer can inspect the code for all tasks and experiments here: https://anonymous.4open.science/r/context-is-key-forecasting-E391. An example task can be found here: https://anonymous.4open.science/r/context-is-key-forecasting-E391/cik_benchmark/tasks/montreal_fire/short_history.py, where the time series window selection occurs from L94-112 and the context generation occurs from L114-158. Additionally, please see point W1 of our response to reviewer kJ9d, where we describe a newly performed LLM-based analysis of the relevance of the context for each task, which indicates that the context enables better forecasts (significantly better for most tasks).
>
> We will add the same discussion to our paper in the next revision.
>
> We hope that this resolves any outstanding concerns. Otherwise, please let us know and we are happy to expand.

---

> > ### Comment · Reviewer_7GBq · 2024-11-25
> >
> > The authors’ response clarifies aspects of the dataset creation process but fails to adequately address concerns about the annotation quality and the lack of rigorous validation. While the authors explain that tasks were created by team members and reviewed by forecasting experts, they provide no evidence of a formalized guideline for task creation or evaluation. Without such a guideline, it is unclear how consistency and quality were maintained throughout the annotation and review process. The absence of clear instructions or criteria for generating textual contexts and designing tasks raises questions about the reproducibility and reliability of the work.
> >
> > Additionally, the lack of inter-annotator agreement (IAA) or equivalent metrics is a critical oversight. IAA is a standard approach in annotation projects to assess the consistency between multiple annotators or evaluators. The response does not indicate whether at least two independent annotators or reviewers evaluated each task, nor does it provide any quantitative measures of agreement. This omission undermines confidence in the reliability of both the human efforts and the peer-review process described.
> >
> > The authors’ reliance on an LLM-based analysis to assess the relevance of the context is also problematic. While such analysis can offer additional insights, it cannot substitute for rigorous human annotation validation. The quality and relevance of the human-generated annotations should first be assured through systematic processes, including clear guidelines and measurable inter-annotator agreement, before being supplemented or analyzed by automated methods.
> >
> > In conclusion, the response does not satisfactorily resolve the concerns about annotation quality and review. The absence of guidelines for task creation and review, along with the lack of inter-annotator agreement or evidence of independent validation, weakens the credibility of the dataset and its creation process. The authors need to provide clear documentation of their annotation and review protocols, along with quantitative metrics like IAA, to ensure the dataset’s reliability and reproducibility.

---

> > > ### Comment · Reviewer_7GBq · 2024-11-25
> > >
> > > The authors’ response highlights the novelty of their CiK benchmark by emphasizing the handcrafted nature of its tasks and the focus on high-quality contexts compared to existing work. However, this contribution appears incremental and raises concerns about scalability and practical relevance. While the authors argue that high-quality contexts are crucial for accurate forecasting, they fail to justify why this emphasis is more important than addressing the challenges posed by low-quality or noisy data, which are far more common in real-world applications. Furthermore, the reliance on manual efforts without a standardized annotation process or quality control measures, such as IAA, undermines the reliability and reproducibility of their approach. The validation of "high-quality" contexts is supported solely by forecasting results, which does not adequately address concerns about methodological rigor. Although the authors mention exploring scalable approaches in the future, the lack of a concrete plan for this limits the broader applicability of CiK. To strengthen their work, the authors need to better articulate the practical importance of high-quality contexts, account for noisy data scenarios, and establish robust annotation protocols to ensure consistency and scalability.

---

> > > > ### Comment · Reviewer_7GBq · 2024-11-25
> > > >
> > > > The authors' response provides some clarification regarding the categorization of tasks by domain, context sources, and capabilities, and highlights their role in guiding task design and analysis. However, critical concerns about the lack of a standard protocol and quality control in task definition and annotation remain unaddressed. While the authors describe their peer-review process and emphasize that the categorization is informative rather than prescriptive, they fail to present a formalized approach or rigorous validation for ensuring consistency and reliability in these categorizations. This omission raises concerns about the reproducibility and methodological soundness of their benchmark.
> > > >
> > > > Furthermore, the authors’ distinction between instruction-following tasks and descriptive contexts is neither theoretically supported nor sufficiently motivated. Their classification of public safety tasks as non-instructional appears to hinge on subjective interpretation rather than an objective framework, leaving significant room for ambiguity and misinterpretation. This is particularly concerning when considering related work, such as Zeng et al. [1], which offers a more structured approach to evaluating instruction-following capabilities in large language models. The lack of alignment with or acknowledgment of such theoretical foundations further undermines the credibility of their categorization.
> > > >
> > > > [1] Zeng, Zhiyuan, et al. "Evaluating large language models at evaluating instruction following." arXiv preprint arXiv:2310.07641 (2023).

---

> > > > > ### Comment · Reviewer_7GBq · 2024-11-25
> > > > >
> > > > > The authors’ response provides an explanation for the seemingly contradictory results in Figures 4 and 5, attributing the differences to the distributional effects of adding context. While this explanation clarifies the phenomenon, it raises concerns about the appropriateness and reliability of the metrics chosen for evaluation. Specifically, the reliance on aggregated RCRPS and win rate metrics warrants further scrutiny.
> > > > >
> > > > > The authors acknowledge that the aggregated RCRPS metric does not fully reflect the distribution of task performance, as it is dominated by significant failures in a minority of tasks. This limitation undermines the utility of RCRPS as a comprehensive evaluation metric, as it may misguide users by masking important nuances in the model’s performance. If the aggregated RCRPS fails to capture the variability across tasks effectively, it is unclear why the authors have not explored or proposed a more robust aggregation method that better accounts for such distributional effects.
> > > > >
> > > > > Additionally, the authors suggest using the win rate across tasks as a complementary metric. However, win rate metrics are often criticized for being difficult to reproduce and unstable, particularly when task distributions are uneven or context-dependent. Without a clear framework for standardizing win rate calculations, this metric could introduce additional uncertainty and variability into the evaluation process.

---

> > > > > > ### Comment · Reviewer_7GBq · 2024-11-25
> > > > > >
> > > > > > Thank you for conducting additional experiments and sharing the updated results. However, these findings further reinforce critical concerns about the paper's methodology and evaluation framework. First, the new results demonstrate that instruction-following ability plays a significant role in the success of models, as some models fail to adhere to the required output template when using the Direct Prompt method. This aligns with concerns raised in the previous response to the evaluation categorization, where the lack of alignment with theoretical motivation or prior work, was highlighted. The task definitions in the paper appear to overlook the importance of instruction-following capabilities, which are central to the effective use of context-aided forecasting tasks.
> > > > > >
> > > > > > Second, the poor performance of Mistral-7B-Instruct and the degradation of results with LLMP prompts raise questions about the robustness of the proposed metrics. The observed failures, particularly when context worsens performance, could stem from limitations in the model family or the aggregated RCRPS metric itself. Aggregated metrics may fail to capture critical nuances in task performance, especially when a minority of tasks experience significant failures that dominate the overall score. This reinforces the need for more robust, context-sensitive evaluation metrics that account for distributional effects and better reflect model capabilities.
> > > > > >
> > > > > > Lastly, while compute constraints limited the testing of larger models in time for the rebuttal, this omission weakens the comprehensiveness of the evaluation. Addressing these gaps in future work will be crucial to provide a complete analysis. Overall, these results highlight the need to reconsider both the task definitions and the evaluation framework to ensure the benchmark effectively captures the key capabilities required for context-aided forecasting.

---

> ### Author Response · Authors · 2024-11-19
>
> > W2: Limited benchmark novelty
>
> Thank you for sharing these additional references, we will revise the related work section to include them. While we do agree that these works are related to CiK, we reiterate a key difference that supports the novelty of this work: CiK is the first benchmark where the tasks are carefully designed such that the text context contains information that is crucial to accurate forecasts. Previous works assembled large collections of time series and text, but the crucial nature of the text is not guaranteed. We detail the exact differences between CiK and these works below:
>
> For example, [1][2] include 1) contextual information that is programmatically scraped and filtered through, e.g., regex, and 2) a large volume of possibly irrelevant contextual information, such as the majority of earnings calls audio and transcripts. While such automated approaches to dataset construction are useful for generating a large dataset in a scalable manner (e.g., for training purposes), the added value of the contextual information is uncertain, hence one cannot reliably evaluate context-aided forecasting with it.
>
> This is the issue that we aimed to avoid when creating CiK, where all tasks are meticulously handcrafted using real-world data to ensure reliability. Hence, we believe our work is complementary to related work, and addresses an important quality gap in existing benchmarks, introducing novel elements such as a rigorous framework for manually generating context-aided forecasting tasks with various sources of context and capabilities required to leverage it, broad domain diversity (e.g., compared to the single domain focus in [1, 2]), and a new metric specifically designed for context-aided forecasting whose relevance has been emphasized by other reviewers.
>
> In our next revision of the paper, we will extend the related work section by including and discussing the differences between CiK and the two additional references you provided. We will also include the exploration of more scalable approaches to high-quality benchmark design for context-aided forecasting in the Future Work section.
>
> ---
>
>
> > W3: Ambiguous Task Type Annotations
>
> Tasks are categorized according to three characteristics: domain, context sources, and capabilities. Examples of tasks and justifications for their domain, context source, and capability categorizations are given in Appendix B to help the reader's intuition.
>
> Domains are fully determined based on data provenance and are outlined in Appendix A.1. Context sources, defined in Section 3.2 and exemplified in Fig. 3, refer to various aspects of the underlying temporal process from which the context draws information. Finally, capabilities, defined in Section 3.3, reflect high-level capabilities that are required to translate the context into correct forecasts. **Exact definitions of the capabilities can be found in Appendix A.4**. It is true that capabilities are subject to debate and that achieving a perfect categorization is perhaps not possible. We used the peer review process previously described to arrive at these conclusions (which will be clarified in the paper). However, we view these as being informative as opposed to prescriptive and they serve two purposes: 1) they provide us with a conceptual scaffolding for designing new tasks by highlighting cognitive capabilities to test and 2) they offer structured dimensions for analysis, allowing us to evaluate model performance and identify limitations across different aspects of a forecaster's capability.
>
> As for your question regarding the public safety (or MontrealFire) tasks, these were not classified as instruction following since the context is purely descriptive of the situation, rather than giving explicit instructions like “suppose that” as in the [ElectricityIncreaseInPredictionTask](https://anon-forecast.github.io/benchmark_report_dev/ElectricityIncreaseInPredictionTask.html) or “consider that” in the [CashDepletedinATMScenarioTask](https://anon-forecast.github.io/benchmark_report_dev/CashDepletedinATMScenarioTask.html). We hope that this explanation, combined with the definitions and examples in the appendix adds clarification and resolves your concerns. Please let us know.

---

> ### Author Response · Authors · 2024-11-19
>
> > W4: Unexplained Results Discrepancies
>
> Figures 4 and 5 show results that seem contradictory for Mixtral, but are compatible and due to a phenomenon we have observed: adding context improves the model’s RCRPS for most tasks but greatly worsens it for a minority of tasks (see Sec. 5.4, significant failures). The improvement on the majority of tasks is thus visible in Figure 5, with an increased number of tasks for which the model with context becomes better than all quantitative baselines. But the significant performance degradation (in absolute terms) on the few other tasks dominates the aggregated RCPRS value shown in Figure 4.
>
> To better illustrate this effect, we have generated a [figure](https://raw.githubusercontent.com/anon-forecast/benchmark_report_dev/refs/heads/main/iclr_rebuttal_resources/dp_mixtral_comparison.png) which shows the impact of adding context to Mixtral-8x7B-Inst on the RCRPS (the lower, the better). While adding context (blue) increases the frequency of RCRPS values close to 0, it also adds a fat tail of RCRPS higher than 2. The former is why adding context improves the model win rate, while the later explains why adding context worsens its aggregate RCRPS.
>
> Thank you for pointing out this pair of results. We believe it motivates reporting both the aggregate RCRPS metric and the win rate across tasks, which offer complementary perspectives on the results. It also strengthens our claim for the need for future work on more robust models for context-aided forecasting (Sec. 5.4). We will also include a discussion on this set of results in the appendix in the next revision of the paper.
>
> ---
>
> > W5: Limited model variety
>
> We acknowledge that evaluating on a diversity of model sizes is important, which is why we evaluated multiple sizes of models (8B, 70B, 405B) and reported an efficiency-performance analysis.
>
> Following the reviewer’s suggestions, we experimented with the following models with Direct Prompt:
> Qwen-2.5-0.5B-Instruct, Qwen-2.5-1.5B-Instruct, Qwen-2.5-3B-Instruct, Qwen-2.5-7B-Instruct,
> Mistral-2.5-7B-Instruct, Falcon-2.5-7B-Instruct.
>
> and the following models with LLMP:
> Qwen-2.5-0.5B, Qwen-2.5-1.5B, Qwen-2.5-3B,
> Qwen-2.5-0.5B-Instruct, Qwen-2.5-1.5B-Instruct, Qwen-2.5-3B-Instruct.
>
> The updated results are attached in [this figure](https://raw.githubusercontent.com/anon-forecast/benchmark_report_dev/refs/heads/main/iclr_rebuttal_resources/results_with_new_models.jpg). We explain them below.
>
> **Results with the Direct Prompt method**:
>
> We find that with the Direct Prompt (DP) method, Qwen-2.5-7B-Instruct especially seem to outperform many other models as per the average RCRPS metric, and also obtain significant improvements in performance when provided with the context. The models achieves the third best score with the Direct Prompt method, significantly outperforming other, more expensive models. We thank the reviewer again for suggesting us to run this model.
>
> We found that Mistral-7B-Instruct does not output accurate forecasts and further suffers from significant failures when provided with the context (worsening its performance), similar to Mixtral-8X7B-Instruct. This indicates that the specific model family used indeed may have an impact on the robustness of the model for context-aided forecasting with the Direct Prompt method.
>
> All other models (Qwen-2.5-0.5B-Instruct, Qwen-2.5-1.5B-Instruct, Qwen-2.5-3B-Instruct, Falcon-2.5-7B-Instruct) fail to follow the template required in the output, which impedes proper evaluation. Instead, despite these models being instruction-tuned, the models produced unrelated information in response to the Direct Prompt prompt, for example, see the output of the Qwen-2.5-0.5B-Instruct model [here](https://raw.githubusercontent.com/anon-forecast/benchmark_report_dev/refs/heads/main/iclr_rebuttal_resources/screenshot_of_failure_qwen_0_5b.png). This may indicate that the Direct Prompt method would need models of a certain size to work with, to leverage their instruction-following capabilities to output valid forecasts.
>
> **Results with the LLMP method**:
>
> We found that all tested models (Qwen-2.5-0.5B, Qwen-2.5-1.5B, Qwen-2.5-3B, Qwen-2.5-0.5B-Instruct, Qwen-2.5-1.5B-Instruct, Qwen-2.5-3B-Instruct) achieve mediocre performance both with and without context, much worse than previously tested models. Notably, all these models except Qwen-2.5-3B and Qwen-2.5-3B-Instruct degrade with context and obtain significant failures, which may indicate that the LLMP method may need models of a certain size to work well for context-aided forecasting.
>
> We tried to run Qwen-2.5-7B, Qwen-2.5-7B-Instruct, Mistral-2.5-7B-Instruct and Falcon-2.5-7B-Instruct however these models took up a high amount of compute and time to run, therefore we could not finish them in time for the rebuttal. We will add them to the camera-ready and analyse them wherever required.
>
> We will add all these results to the next revision to the paper.

---

> ### Author Response · Authors · 2024-11-19
>
> > Q1: How the textual contexts were annotated, including any guidelines, annotator expertise, and inter-annotator agreement (IAA) metrics?
>
> Please see our response to W1 for the details.
>
> ---
>
> > Q2: Do the authors envision methods to automate or partially automate this process, such as using existing NLP techniques to generate context?
>
> The process was completely manual. We believe that the manual task creation process ensures a high quality benchmark. Please see our response to W2 for the details.
>
> ---
>
> > Q3: Could the authors provide explicit definitions for each capability and clarify the criteria used to categorize tasks?
>
> Please see our response to W3 for details on the task type annotations. For a full description of the task creation process, please see our response to W1.
>
> ---
>
> > Q4: Could the authors clarify whether this discrepancy suggests limitations of the CRPS metric or other factors in the benchmark design?
>
> Please see our response to W4 for more details. In short, we believe that this set of results actually credits the RCRPS as a metric that captures the quality of a forecast (how far the forecast is from the ground truth, in an absolute sense), as opposed to a task-wise win rate, which captures in how many tasks the model wins over the other models. Especially in the case where models fail significantly and these failures dominate the aggregate RCRPS (see Sec 5.4), the win rate provides a different view and shows the models still are useful and perform well across many tasks.

---

> > ### Author Response · Authors · 2024-11-22
> >
> > Dear reviewer, we would like to make sure your comments and concerns have been addressed. Please let us know if you need any additional clarifications.

---

> ### Author Response · Authors · 2024-11-27
>
> # Response to Reviewer 7GBq's Comments
>
> We respectfully disagree with the reviewer, as our benchmark and evaluation’s rigor, reliability, and reproducibility are supported by established practices in the related literature. More precisely:
>
> * The reliability of our benchmark is supported by a triplet of strategies:  the rigorous peer review process similar to that of the reviewer’s proposed reference [1] and described in Appendix A.2, the LLM-based critique that reviewer kJ9d requested, and the objective evaluation of forecasts against a numerical ground truth, including the additional results that the reviewer asked for.
> * The reproducibility of our evaluation is streamlined by the code and data releases, as is standard in the literature (https://anonymous.4open.science/r/context-is-key-forecasting-E391). We went a step further by providing a task visualization interface, to ease the task inspection by the community (https://anon-forecast.github.io/benchmark_report_dev/).
> * The rigor of our evaluation methodology exceeds or matches that of any previous work in multimodal forecasting [2-8]: we make use of a scoring rule that evaluates the entire predictive distribution rather than focusing solely on summary statistics (unlike MSE and MAE [2,3]) (Section 4), we discuss and motivate all aggregation and normalization choices (Section 5.1, Appendices A.4 and E.1), we report and analyze examples of model successes and failures (in appendix C.3, C.4 and C.5), and we provide different views for assessing model performance (notably reporting results by context type, capability, with/without context).
>
> In the following comment, we counterargue the reviewer’s points and stress that they do not constitute grounds for rejection:

---

> ### Author Response · Authors · 2024-11-27
>
> ## Annotation Quality
> * **Inter-annotator agreement (IAA) is not an established practice in multimodal forecasting**. IAA is used in only a single paper unrelated to forecasting [1], out of 7 discussed related works [2-8], including two papers put forward by the reviewer [7,8] and two papers on multimodal forecasting published at the NeurIPS [2] and EMNLP [3] conferences, both of which rely on manual curation. Furthermore, in [1] the reported IAA is motivated by finding agreement between inherently subjective author preferences between two answers that they manually designed for a task in a meta-evaluation benchmark on human preference datasets. However, no clear instructions or criteria for evaluation are reported, apart from objectivity. Therefore, our *formalized task creation and peer review validation process (Appendix A.2)* not only shares elements with  that put forward in the suggested paper that uses the IAA [1], but in effect   supersedes it because tasks were not admitted to the benchmark without the unanimous agreement of all co-authors (excluding the task creator).
> * **The use of IAA as in [1] is not scalable**, because it is human-based, despite the reviewer claiming that lack of scalability is a limitation of our work.
>
> ---
>
> ## Novelty and Relevance
> * It’s unclear why the reviewer lists two strengths of the benchmark as being  “uniquely requir[ing] essential textual context for forecasting, marking a new direction in multimodal prediction” and being “robust, with real-world tasks”, while subsequently insisting on the importance of “low-quality or noisy data, which are far more common in real-world applications [no citation provided]”. We welcome any constructive feedback with references to the appropriate literature.
> * “... the authors argue that high-quality contexts are crucial for accurate forecasting”. This is misreported. We argue that high-quality contexts are crucial for assessing how well models perform context-aided forecasting, to confidently attribute failures to models instead of poorly-defined tasks.
>
> ---
>
> ## Unfounded Instruction-Following Definition
> * Despite the reviewer claiming insufficient motivation for our purely informative definition of instruction following, the structured approach in [1] commended by the reviewer for its objectivity offers no support or motivation for their definition of instruction-following. Further, they use the IAA for establishing agreement on preferences, not for defining instruction-following.
> * **The validity of our benchmark is unaffected by this definition, which we use informatively for result interpretation**.
>
> ---
>
> ## Appropriateness of the RCRPS
> * As the RCRPS is derived from the CRPS metric, it accounts for distributional errors *for each task instance*. It is trivially true that the aggregate RCRPS misses distributional aspects by the definition of metric aggregation.  Such aggregations are standard in the literature [2,3], and are not aware of alternatives for the complementary averaging and the win-rate aggregations we present, to provide a more concise view of model performance.  We welcome constructive feedback and references in this direction.
> * “...RCRPS as a comprehensive evaluation metric…” This is misreported. We introduce the RCRPS as a novel metric based on the well-established CRPS metric, but augmented specifically for context-aided forecasting. We also report the average rank, win rate and visualizations of model failures, to provide complementary views into model performance.
> * Please provide a citation for the inappropriateness of the win rate.
>
> ---
>
> ## Larger Models
> * The reviewer’s subsequent criticism on the omission of larger models (after their initial request for smaller models) is unfounded. Not only do we evaluate models from diverse families up to 405B parameters, but we also evaluate a diversity of model sizes and families that goes well beyond the diversity exhibited by the related literature, including the small models specifically requested by the reviewer. Existing publications either do not evaluate open-source models exceeding 8B parameters [2] or do not run forecasting evaluations on models outside the closed-source GPT family [3].

---

> > ### Author Response · Authors · 2024-11-27
> >
> > **Overall, it is no longer clear to us what the reviewer considers a strength or a weakness of our submission, nor what the reviewer would like us to change.**
> > * Is the handcrafted context a strength due to its uniqueness, novelty and real-world tasks (strengths mentioned in initial review), or a weakness due to its incremental nature, lack of scalability and missing practical relevance (criticisms in response to rebuttal)?
> > * Does the reviewer want us to follow established review practices, such as the manual task creation as in [1] followed by a peer review process such as the one that we describe in Appendix A.2, or does the reviewer want us to alter the task creation and validation processes away from existing, peer reviewed approaches [2,3] to enable the calculation of the IAA used by a single paper on a benchmark dealing with subjective human preferences to evaluate LLM evaluators?
> > * Does the reviewer want us to create a theoretically-backed definition of instruction following, or an ad hoc one such as that in [1] which is cited as an example of an “objective framework”? If the reviewer acknowledges the proposed definition as informative rather than prescriptive, and the authors describe a peer review-based validation process on par with [1], why does this raise concerns on the soundness of the benchmark?
> > * How does the reviewer propose that we identify a (metric, aggregation strategy) pair that captures all possible nuances that may take place between distributions of tasks? Is there any such notion that is used in the existing literature? We would welcome any constructive feedback in this direction.
> > * Does the reviewer want smaller model evals or larger model evals? We are open to constructive suggestions backed by the literature that we can complete in the remaining time.
> >
> > ---
> >
> > [1] Zeng, Zhiyuan, et al. "Evaluating Large Language Models at Evaluating Instruction Following." Proceedings of the International Conference on Learning Representations, 2024, arXiv:2310.07641.
> >
> > [2] Liu, Haoxin, et al. "Time-MMD: Multi-Domain Multimodal Dataset for Time Series Analysis." The Thirty-Eighth Conference on Neural Information Processing Systems: Datasets and Benchmarks Track, 2024.
> >
> > [3] Mike A Merrill, Mingtian Tan, Vinayak Gupta, Thomas Hartvigsen, and Tim Althoff. 2024. “Language Models Still Struggle to Zero-shot Reason about Time Series”. In Findings of the Association for Computational Linguistics: EMNLP 2024, pages 3512–3533, Miami, Florida, USA. Association for Computational Linguistics.
> >
> > [4] Zhang, Yunkai, et al. "Insight Miner: A Large-scale Multimodal Model for Insight Mining from Time Series." NeurIPS 2023 AI for Science Workshop. 2023.
> >
> > [5] Xu, Zhijian, et al. "Beyond Trend and Periodicity: Guiding Time Series Forecasting with Textual Cues." arXiv preprint arXiv:2405.13522 (2024).
> >
> > [6] Emami, Patrick, et al. "SysCaps: Language Interfaces for Simulation Surrogates of Complex Systems." arXiv preprint arXiv:2405.19653 (2024).
> >
> > [7] Sawhney, Ramit, Arnav Wadhwa and Shivam Agarwal. “FAST: Financial News and Tweet Based Time Aware Network for Stock Trading.” Conference of the European Chapter of the Association for Computational Linguistics (2021).
> >
> > [8] Liu, Mengpu, Mengying Zhu, Xiuyuan Wang, Guofang Ma, Jianwei Yin and Xiaolin Zheng. “ECHO-GL: Earnings Calls-Driven Heterogeneous Graph Learning for Stock Movement Prediction.” AAAI Conference on Artificial Intelligence (2024).

---

> > > ### Comment · Reviewer_7GBq · 2024-11-27
> > >
> > > Thank you for your response. However, it appears that several of my concerns remain misunderstood or inadequately addressed. The authors’ rebuttal does not clarify or refine their methods but instead deflects important points. Below, I outline the unresolved issues:
> > >
> > > 1. While the hand-crafted high-quality context is a strength of the benchmark, why does this strength preclude acknowledging its weaknesses, particularly regarding scalability and connection to real-world noisy scenarios? It is a fact that extending the benchmark to include new tasks would involve significant human effort, and real-world scenarios often present noisier data. These points should be addressed constructively rather than dismissed. Instead, the response appears defensive, as though the benchmark should have no limitations.
> > >
> > > 2. The emphasis on IAA appears to misrepresent the core of my concern. The major issue is the lack of evidence supporting the claim of high-quality contexts, even though the authors describe a quality control process. What materials and measurements were used to ensure the quality of the benchmark during its construction? Do all the authors hold the same criteria for accepting the task?  Suggesting that reviewers and users verify quality by examining the code and data themselves shifts the burden onto external stakeholders rather than providing transparency and rigor upfront. Why not include metrics or materials to substantiate the quality claim?
> > >
> > > 3. The authors’ definition of instruction-following diverges from widely adopted definitions in prior work, yet this point is not acknowledged or justified adequately. I can list more literatures here if this is what authors require. Even though their results align with widely-adopted definitions of instruction-following, they do not incorporate this into their categorizations or evaluation framework. Many models exhibit low performance because they fail to follow instructions and provide expected results. Why is this critical aspect omitted from the evaluation, and why does the authors' framework insist on a unique definition rather than engaging with this aspect? This creates unnecessary confusion for users. By not addressing this directly, the authors miss an opportunity to strengthen their categorization and clarify their framework for users.
> > >
> > > 4. The authors acknowledge drawbacks in the aggregation of RCRPS but continue to use it as a major metric in the benchmark. Users are likely to rely on aggregated RCRPS for model comparisons, so ensuring its robustness is critical. While the authors suggest using both aggregated RCRPS and win rate for evaluation, they fail to propose any refinement or alternative aggregation strategy despite the evident limitations revealed by their own results. This oversight diminishes the utility of the metric. As for win rate, its limitations in model comparisons compared to statistical tests are well known. Do the authors genuinely require citations of foundational statistical tests to accept this critique?

---

> > > > ### Author Response · Authors · 2024-11-28
> > > >
> > > > We remind the reviewer that we responded to the raised concerns and revised the paper according to their initial review. In general, we do not understand why the remaining concerns are still standing nor why they are grounds for rejection.
> > > >
> > > > ---
> > > >
> > > > ## 1. Scalability and context quality
> > > >    1. In our initial response, we acknowledged that “the exploration of more scalable approaches to high-quality benchmark design for context-aided forecasting” is an interesting direction of future work and revised the discussion section of the paper accordingly.
> > > >    2. Although contexts in the wild will be noisy, we repeatedly stated that we must ensure the relevance of contexts in our benchmark to “reliably attribute failures to models and not to badly-defined tasks or contexts.”
> > > >
> > > > ---
> > > >
> > > > ## 2. Evidence to support high-quality contexts
> > > >    1. Materials and measurements to ensure benchmark quality during construction
> > > >       1. We thoroughly described the task validation process in our initial response and reported it in Appendix A.2. We repeat the criteria here, for the reviewer’s convenience:
> > > >          1. the text was of high quality, that it undoubtedly enabled a better forecast, and
> > > >          2. that the context source and capability tags were well-assigned
> > > >       2. Definitions of related concepts are in Section 3.2 and Appendix A.6.
> > > >       3. As for materials, the same [benchmark visualization website](https://anon-forecast.github.io/benchmark\_report\_dev/) was used to inspect the tasks by the committee during the task validation.
> > > >    2. Do the authors hold the same criteria for accepting the task?
> > > >       1. As previously stated, tasks were not admitted into the benchmark without **100% agreement** of the review committee based on the criteria put forward in the materials quoted above.
> > > >    3. “Shifting the burden” for examination
> > > >       1. As noted above and in our previous response, tasks were admitted only if there was unanimous agreement of the review committee based on the criteria. We further included a LLM-based critique to substantiate the quality claim (Appendix A.3), and the objective evaluation of forecast improvements with contexts w.r.t. without.
> > > >
> > > > ---
> > > >
> > > > ## 3. Instruction Following
> > > > 1. Our definition aligns with that of the instruction-following literature, such as that proposed by a recently published survey paper on Instruction Following: "\...understand various instructions and produce the corresponding responses"[1].
> > > > 2. We use the term instruction following only for task categorization in the benchmark. The fact that methods such as Direct Prompt fail to output in a parsable format is irrelevant to the task categorization or the benchmark, and is solely a limitation of the method itself. Furthermore, we do discuss how we handle such cases in evaluation (See Appendix D.1.1).
> > > >
> > > > ---
> > > >
> > > > ## 4.  RCRPS aggregation
> > > >
> > > > Any benchmark must report summary statistics and thus aggregation is inevitable. We summarize our previous response:
> > > >
> > > >    1. We normalized to enable cross-task aggregation “without the average being dominated by \[...\] tasks with time series with large values”. See Appendix Section E.1, “Scaling for cross-task aggregation.”
> > > >    2. Aggregate RCRPS, win-rate and examples of successful, and failed forecasts are used to provide complementary (although individually imperfect) views.
> > > >    3. We welcome any references in the relevant forecasting literature that provide superior aggregation methodologies.
> > > >
> > > > ---
> > > >
> > > > We hope that this response clarifies that there is no intention to deflect criticism, but rather that the reviewer’s concerns have previously been addressed.
> > > >
> > > > \[1\] Lou, Renze, Kai Zhang, and Wenpeng Yin. "Large Language Model Instruction Following: A Survey of Progresses and Challenges." Computational Linguistics (2024): 1-10.

---

> > > > > ### Comment · Reviewer_7GBq · 2024-12-02
> > > > >
> > > > > Thank you for the detailed response to the raised concerns. However, the revisions provided still fail to address critical issues in a convincing manner, and the inconsistencies, lack of rigor, and fundamental flaws in the methodology persist. Below, I outline the primary issues that remain unresolved:
> > > > >
> > > > > 1. Appreciate that authors acknowledge this drawback of their benchmarks.
> > > > > 2. While the authors explain their validation process and emphasize unanimous committee agreement, the mechanism for ensuring consistency and rigor in task categorization remains unclear. Unanimous agreement from reviewers is claimed. The response does not clarify the qualifications or expertise of the reviewers. How can readers trust that these reviewers were equipped to make consistent, unbiased judgments on task validity and categorization? Were reviewers provided with detailed and standardized instructions for evaluating tasks? The inconsistencies highlighted in the MontrealFire task suggest a lack of rigor. How can one confidently conclude that descriptive contexts (e.g., MontrealFire) are not instruction-following while other contexts with vague instruction-like phrasing are accepted? If reviewers disagreed during the task validation process, how were disagreements resolved? The response does not address whether such scenarios occurred or how they were handled. Were disagreements systematically reviewed or arbitrated by an external authority or set of rules? If so, what were those procedures?
> > > > > 3. The authors’ use of the definition from instruction-following literature is contradicted by their own framework and examples. The reliance on explicit linguistic cues such as "suppose that" or "consider that" as the basis for instruction-following categorization is overly simplistic and does not align with the cited definition. The claim that Direct Prompt’s failures are irrelevant to the benchmark further undermines the credibility of this response. It suggests a narrow, inflexible interpretation of instruction-following, which is inconsistent with the authors' own cited references. This demonstrates a failure to understand or properly implement the broader concept of instruction-following. The entire categorization framework requires significant rethinking to eliminate biases and inconsistencies.
> > > > > 4. The authors’ approach to aggregated RCRPS remains fundamentally flawed and undermines the reliability of their conclusions. The aggregated RCRPS metric is dominated by a minority of tasks where performance is "greatly worsened." This results in a skewed overall metric that does not accurately reflect the general trends in task performance. This issue is compounded by the authors’ own observation that adding context improves performance for the majority of tasks, yet the minority of tasks with severe degradation disproportionately affects the aggregated metric. Winning rate is merely a descriptive measure and cannot substitute for rigorous statistical evaluation. Any conclusions drawn from it are inherently superficial and lack scientific rigor. Winning rates fail to account for task variability or establish the statistical significance of observed differences. For example, while the authors assert that context improves performance in the majority of tasks, this claim is not substantiated by statistical evidence. Without proper pairwise comparisons or hypothesis testing, such conclusions are not credible. The authors’ dismissal of statistical pair tests is particularly concerning. Statistical tests are essential for validating claims about performance improvements, especially when metrics show conflicting trends. Robust analysis, such as paired t-tests or non-parametric tests, could demonstrate whether the performance differences with and without context are statistically significant. Ignoring this step leaves the conclusions speculative and unconvincing. Without rigorous statistical evaluation, the conclusions about the effectiveness of context remain unsupported and cannot be taken seriously.

---

### Comment · Area_Chair_7eMT · 2024-11-25

Dear reviewers,

As the deadline for discussion is ending soon. Please respond to the authors to indicate you have read their rebuttal. If you have more questions, now is the time to ask. This is important since the paper is currently undergoing extremely divergent scores.

AC

---

### Meta-Review · Area_Chair_7eMT · 2024-12-19

**Metareview:**

This work proposes Context is Key (CiK) benchmark to evaluate how well forecasting models can integrate textual context with numerical time series data, spanning 71 tasks across various domains. It is indeed a comprehensive coverage across multiple domains, and it further introduced a novel Region of Interest CRPS metric for context-sensitive evaluation. The benchmark provides a valuable research foundation for combining text and time series analysis, and the experimental evaluation includes testing various model types and prompting techniques.

However, several critical weaknesses lead to rejection. As pointed out by most reviewers, this workr lacks crucial information about the context annotation process, including guidelines, annotator expertise, and inter-annotator agreement metrics (Reviewer 7GBq, kJ9d). There are concerns about the benchmark's novelty, as it resembles prior work integrating time-series data with textual sources (Reviewer 7GBq). The analytical rigor is insufficient, with minimal evidence supporting the relevance of provided textual information to forecasts (Reviewer kJ9d). Important details about historical context length, forecasting horizon, and sample sizes are missing (Reviewer kJ9d). The work lacks proper discussion of text quality filtering and the relative importance of textual versus time-series information (Reviewer RToE). Additionally, there are unexplained performance discrepancies in the results (Reviewer 7GBq), and the benchmark's real-world applicability is limited due to its reliance on LLM-backbone models and the challenge of obtaining high-quality textual data (Reviewers 6GZP, RToE).

**Additional Comments On Reviewer Discussion:**

Reviewer 7GBq and RtoE are not satisfied with the rebuttal, typically: The lack of transparency regarding the qualifications of reviewers and the absence of standardized evaluation criteria leave the task validation process fundamentally unclear, undermining trust in the methodology. Moreover, the inconsistencies in task categorization, such as the MontrealFire example, highlight significant flaws in their framework, which relies on overly simplistic linguistic cues that contradict their cited definitions. The issues with the aggregated RCRPS metric, disproportionately influenced by a minority of tasks, and the authors’ dismissal of statistical pair tests further compound the methodological concerns. Without rigorous statistical validation and a more robust categorization framework, the conclusions drawn remain speculative and unconvincing, leaving the critical issues in the paper unaddressed.

---

### Decision · Program_Chairs · 2025-01-22

Reject